# Pessimism Principle Can Be Effective: Towards a Framework for Zero-Shot Transfer Reinforcement Learning

Chi Zhang [1]   Ziying Jia [2]   George K. Atia [1 3]   Sihong He [2]   Yue Wang [1 3]

## Abstract

Transfer reinforcement learning aims to derive a near-optimal policy for a target environment with limited data by leveraging abundant data from related source domains. However, it faces two key challenges: the lack of performance guarantees for the transferred policy, which can lead to undesired actions, and the risk of negative transfer when multiple source domains are involved. We propose a novel framework based on the pessimism principle, which constructs and optimizes a conservative estimation of the target domain's performance. Our framework effectively addresses the two challenges by providing an optimized lower bound on target performance, ensuring safe and reliable decisions, and by exhibiting monotonic improvement with respect to the quality of the source domains, thereby avoiding negative transfer. We construct two types of conservative estimations, rigorously characterize their effectiveness, and develop efficient distributed algorithms with convergence guarantees. Our framework provides a theoretically sound and practically robust solution for transfer learning in reinforcement learning.

## 1. Introduction

Reinforcement learning (RL) aims to learn a policy that optimizes an agent's performance when interacting with its environment. RL has achieved remarkable success in areas such as game playing (Silver et al., 2016; 2017; Mnih et al., 2013), transportation (He et al., 2023a;b; Wang et al., 2023a), robotics (Nguyen & La, 2019; Liu et al., 2021) or natural language processing (Sharma & Kaushik, 2017; Uc-Cetina et al., 2023; Kirk et al., 2023a). These successes, however, often depend on access to vast amounts of data, which are critical for enabling agents to understand their environments and develop effective policies. In domains with limited data—due to high exploration costs or other constraints—RL performance typically degrades. A promising solution is transfer learning (TL), which leverages more data from related environments (source domains) to train policies that can be transferred to target environments. We focus on **zero-shot transfer**, where no target domain data is available prior to deployment, and **multi-domain transfer**, where multiple source domains are utilized.

Despite its potential, TL is often hindered by the Sim-to-Real Gap (Salvato et al., 2021; Zhao et al., 2020), which refers to performance degradation when transferring learned policies from source to target domains. TL typically relies on constructing a proxy to approximate target domain performance and transferring the proxy's optimal policy. While successful when source domains closely resemble the target domain (e.g., high-fidelity simulations), significant divergence between the source and target domains—caused by modeling errors, non-stationarity, or perturbations—leads to inaccurate proxies and degraded transfer performance.

In multi-domain transfer, the presence of multiple source domains introduces additional challenges, particularly negative transfer. Source domains that differ significantly from the target domain can skew the proxy, resulting in overly pessimistic or suboptimal policies that hinder the transfer process if they cannot be identified and treated properly.

These challenges are critical in practical applications, where ineffective policies may have severe consequences, such as mechanical damage in robotics or traffic accidents in autonomous driving. Existing TL methods, such as domain randomization (DR) or imitation learning, lack systematic guarantees for transferred performance, as they fail to establish a robust connection between the proxy and target domain outcomes. To address these limitations, and inspired by the effectiveness of pessimism principle in various RL settings, particularly in scenarios where conservativeness is

[1]Department of Electrical and Computer Engineering, University of Central Florida, Orlando, FL 32816, USA [2]Department of Computer Science and Engineering, University of Texas at Arlington, Arlington, TX 76019, USA [3]Department of Computer Science, University of Central Florida, Orlando, FL 32816, USA. Correspondence to: Chi Zhang, Ziying Jia, George K. Atia, Sihong He, Yue Wang <chi.zhang@ucf.edu, zxj3060@mavs.uta.edu, george.atia@ucf.edu, sihong.he@uta.edu, yue.wang@ucf.edu>.

*Proceedings of the 42$^{nd}$ International Conference on Machine Learning*, Vancouver, Canada. PMLR 267, 2025. Copyright 2025 by the author(s).

advantageous, such as offline RL (Jin et al., 2020; Shi et al., 2022), we propose a novel framework that incorporates the **pessimism principle** in proxy construction for TL. By using a pessimistic estimation of target domain performance, our approach ensures that transferred policies achieve satisfactory outcomes while avoiding severe consequences from overly optimistic actions. Our framework also mitigates negative transfer in multi-domain settings. We rigorously characterize the effectiveness of this approach and design concrete algorithms with convergence guarantees. Our contributions are summarized as follows.

**Introducing pessimism principle in transfer learning.** We introduce the pessimism principle to transfer learning, enabling the transfer of conservative yet effective policies to the target domain. We characterize the framework's effectiveness, showing that the target domain's optimality gap depends on the degree of pessimism used in proxy construction. This motivates the design of proxies with minimal but sufficient pessimism, ensuring robust lower bounds on transferred performance and avoiding the severe consequences common in other TL methods.

**Construction of pessimistic proxy and design of concrete algorithms.** Leveraging the inherent pessimism in robust RL frameworks, we construct pessimistic proxies by incorporating prior knowledge of domain similarities, such as upper bounds on the distance between source and target domains. We develop several proxy types, demonstrate their effectiveness, and propose a distributed algorithm for multi-domain transfer. This algorithm ensures convergence and yields conservative but effective policies across diverse source domains.

**Design of proxy and algorithm to avoid negative transfer.** In multi-domain transfer, some source domains may significantly diverge from the target, leading to overly pessimistic policies if treated equally. We show that the pessimism principle inherently mitigates negative transfer by improving the proxy's value without overly optimistic assumptions. Building on this insight, we propose a refined proxy that avoids negative transfer and design a convergent algorithm to ensure effective policy transfer in such settings.

## 2. Preliminaries and Problem Formulation

**Markov Decision Process.** A Markov Decision Process (MDP) serves as the standard framework for formulating RL. An MDP can be represented by a tuple $\mathcal{M} = (\mathcal{S}, \mathcal{A}, P, r, \gamma)$, where $\mathcal{S}$ and $\mathcal{A}$ denote the state and action spaces, respectively, $P : \mathcal{S} \times \mathcal{A} \rightarrow \Delta(\mathcal{S})$[1] denotes the transition kernel, and we denote the probability vector under $(s, a)$-pair by $P_s^a$, $r : \mathcal{S} \times \mathcal{A} \rightarrow [0, 1]$ is the deterministic reward function, and $\gamma \in [0, 1]$ is the discount factor. At the $t$-th step, the

agent selects an action $a^t$ at state $s^t$, transitions to the next state $s^{t+1}$ according to the transition probability $P_{s^t}^{a^t}$, and receives a reward $r(s^t, a^t)$.

A stationary policy $\pi : \mathcal{S} \rightarrow \Delta(\mathcal{A})$ maps each state to a probability distribution over $\mathcal{A}$, and $\pi(a|s)$ gives the probability of selecting action $a$ at state $s$. The performance of the agent following a policy is measured by the expected cumulative reward, defined as the value function $V_P^\pi$ of policy $\pi$ with transition kernel $P$: $V_P^\pi(s) \triangleq \mathbb{E}_P \left[ \sum_{t=0}^\infty \gamma^t r(s^t, a^t) \big| s^0 = s, \pi \right]$, for all $s \in \mathcal{S}$. Alternatively, the cumulative reward can also be characterized by the Q-function $Q_P^\pi$ for all $(s, a) \in \mathcal{S} \times \mathcal{A}$ defined as $Q_P^\pi(s,a) \triangleq \mathbb{E}_P \left[ \sum_{t=0}^\infty \gamma^t r(s^t, a^t) \big| s^0 = s, a^0 = a, \pi \right]$.

The goal of an MDP is to learn the optimal policy $\pi^*$:

$$\pi^* \triangleq \arg\max_{\pi \in \Pi} V_P^\pi(s), \forall s \in \mathcal{S},$$

which exists and can be restricted to the set of deterministic policies (Puterman, 2014). The optimal value functions are $V_P^* \triangleq \max_\pi V_P^\pi = V_P^{\pi^*}$ and $Q_P^* \triangleq \max_\pi Q_P^\pi = Q_P^{\pi^*}$.

**Robust Markov Decision Process (RMDP).** Robust RL aims to optimize the performance under the worst case, when the environment is uncertain, and it is generally formulated as a robust MDP. The transition kernel in a robust MDP is not fixed but lies in an uncertainty set $\mathcal{P}$. In this paper, we focus on $(s, a)$-rectangular uncertainty sets (Nilim & El Ghaoui, 2004; Iyengar, 2005). Given a nominal kernel $P_0$, the uncertainty set $\mathcal{P}$ is defined as $\mathcal{P} \triangleq \otimes_{(s,a) \in \mathcal{S} \times \mathcal{A}} \mathcal{P}_s^a$, where $\mathcal{P}_s^a = \{P_s^a \in \Delta(\mathcal{S}) : D(P_s^a, (P_0)_s^a) \leq \Gamma_s^a\}$. Here, $D(\cdot, \cdot)$ is a distance metric between two probability distributions, and $\Gamma_s^a$ is the radius of the uncertainty set, measuring the level of environmental uncertainty.

The robust value function and robust Q-function of a policy in a RMDP measure the worst-case performance over all transition kernels in the uncertainty set:

$$V_\mathcal{P}^\pi(s) \triangleq \min_{P \in \mathcal{P}} V_P^\pi(s), Q_\mathcal{P}^\pi(s,a) \triangleq \min_{P \in \mathcal{P}} Q_P^\pi(s,a). \quad (1)$$

Then, the goal in the RMDP problem is to find the optimal robust policy $\pi^*$ that maximizes $V_\mathcal{P}^\pi$: $\pi^* \triangleq \arg\max_\pi V_\mathcal{P}^\pi$. Similarly, we denote $V_\mathcal{P}^{\pi^*}$ and $Q_\mathcal{P}^{\pi^*}$ by $V_\mathcal{P}^*$ and $Q_\mathcal{P}^*$.

The robust value function $Q_\mathcal{P}^\pi$ (and the optimal robust value function $Q_\mathcal{P}^*$, respectively) is the unique fixed point of the robust Bellman operator $\mathbf{T}^\pi$ (and the optimal robust Bellman operator $\mathbf{T}$, respectively) (Iyengar, 2005)[2]:

$$\mathbf{T}^\pi Q(s,a) = r(s,a) + \gamma \sigma_{\mathcal{P}_s^a} \left( \sum_{a \in \mathcal{A}} \pi(a|\cdot) Q(\cdot, a) \right), \quad (2)$$

$$\mathbf{T} Q(s,a) = r(s,a) + \gamma \sigma_{\mathcal{P}_s^a} (\max_{a \in \mathcal{A}} Q(\cdot, a)), \quad (3)$$

---

[1] $\Delta(\cdot)$ denotes the probability simplex defined on the space.

[2] $\sigma_\mathcal{P}(V) = \min_{p \in \mathcal{P}} pV$ is the support function of $V$ over $\mathcal{P}$.

## 2.1. Problem Formulation

The goal of zero-shot multi-domain transfer RL is to optimize the performance under a target MDP $\mathcal{M}_0 = (\mathcal{S}, \mathcal{A}, P_0, r, \gamma)$, from which no data is available. Instead, there are $K$ related source domains $\mathcal{M}_k = (\mathcal{S}, \mathcal{A}, P_k, r, \gamma)$, generating much more data. For simplicity, we consider the case of identical reward. The only assumption is that, there exists an upper bound $\Gamma \geq D(P_0, P_k), \forall k$, that is known by the learner. This assumption is reasonable, as such information is essential to achieve performance guarantee. And even if in the worst-case, we can set $\Gamma = 1$ (in total variation) to construct an (overly) conservative proxy, which yet still avoids decisions with severe consequences in target domain, preferred in transfer learning settings.

In this work, we consider a more challenging setting of distributed, decentralized transfer learning with partial information sharing. Specifically, we consider a framework in which a central server or global agent collects and aggregates partial information (instead of all raw data) from multiple source domains and distributes updated results back to these domains. Each source domain then performs local learning independently based on the received updates and subsequently returns refined outcomes to the global agent.

A key aspect of this framework is its privacy-preserving nature. The underlying environments or raw data from the local domains are not directly shared with the central learner. Instead, only aggregated or processed results are exchanged, safeguarding the confidentiality of local data. Additionally, there is no direct communication between local domains, further ensuring data isolation and reducing the risk of information leakage.

This setting is both practical and relevant to many real-world applications where data privacy is a critical concern. For example, in a ride-sharing platform like Uber, each vehicle and its operational environment can be treated as an independent source domain. A central server may aim to optimize a global dispatching or pricing policy for a new target region, such as a city where the platform is expanding. However, individual vehicles interact with their local environments independently and share only processed insights—such as aggregated trip statistics or anonymized performance metrics—with the server. This TL framework hence ensures effective policy transfer while maintaining the privacy of individual vehicle data.

## 3. Related Work

We briefly discuss the most commonly used transfer learning approaches (see Appendix A for additional discussions).

**Domain Randomization:** Domain randomization (DR) is widely used to reduce the Sim-to-Real Gap, by training agents on randomized versions of the simulated domain (Tobin et al., 2017; Vuong et al., 2019; Mehta et al., 2020; Tiboni et al., 2023; Tommasi, 2023). By introducing degrees of variability in domain properties such as physics, or object appearances, the agent aims to optimize the expected performance under a uniform distribution over these environments. DR is expected to enhance the robustness and generalizability of the learned policy so that it performs well when deployed in the target domain. Although DR has proven effective in many tasks (Shakerimov et al., 2023; Niu et al., 2021; Slaoui et al., 2019; Jiang et al., 2023), it has limitations when the domain shift is large or the target environment is significantly different from the simulated one, in which case the randomized environments do not provide enough information for the target domain. Furthermore, DR generally lacks theoretical guarantees and justifications for the performance in the target domain, except under some strict assumptions about the structures of the underlying MDPs (Chen et al., 2021; Jiang et al., 2022).

**Multi-task Learning:** Multi-task learning (MTL) is another method that aims to leverage knowledge from multiple related tasks to improve learning efficiency and generalization. Specifically, MTL aims to optimize a finite set of tasks simultaneously or to obtain a Pareto stationary point (Yang et al., 2020; Wilson et al., 2007; Vithayathil Varghese & Mahmoud, 2020; Teh et al., 2017). When the target domain shares common properties with these source tasks, this information will be maintained during the optimization of all source tasks and transferred. However, similar to DR, MTL also suffers from insufficient theoretical justification, requiring assumptions on the common representations of tasks. Moreover, MTL cannot avoid negative transfer, where some source tasks can be relatively distinct from the target task, thus slow down transfer learning. Also, solving an MTL problem can be difficult, due to the differences among the tasks known as the gradient conflict (Wang et al., 2024f).

**Representation Learning for Multitask RL:** Representation learning aims to extract the shared structure across multiple tasks, and utilize such a structure to reduce sample complexity in downstream tasks with similar features (Teh et al., 2017; Sodhani et al., 2021; Arulkumaran et al., 2022). Cheng et al. (2022); Agarwal et al. (2023); Sam et al. (2024) investigated multitask representation learning in an online RL setting, where the agent interacts with multiple source tasks to extract a common latent structure. Ishfaq et al. (2024) proposed Multitask Offline Representation Learning (MORL), which adopt the principle of pessimism and learn a shared representation from offline datasets across multiple tasks modeled by low-rank MDPs. However, these methods rely on the assumption that all tasks share a low-rank latent structure and that a common representation can be learned and reused, which may not hold in practical zero-shot transfer scenarios with large domain shifts.

# 4. Pessimism Principle for Transfer Learning

## 4.1. Major Barriers in Prior TL Methods

Before developing our pessimism principle for transfer learning, we first identify two fundamental limitations in existing TL methods: (1) **the lack of a clear connection between the optimization proxy and the target domain performance**, and (2) **the inability to identify and exclude source information that may cause negative transfer**.

As discussed, most existing TL approaches optimize a proximal objective function $f(\pi)$ and deploy its optimal policy in the target domain. However, there are limited guarantees regarding the relationship between the proxy $f(\pi)$ and the target domain value function $V_{P_0}^\pi$, except under some strong assumptions (e.g., (Chen et al., 2021)). Consequently, it is unclear whether the optimal policy for $f(\pi)$ will also perform well under $P_0$, leaving uncertainty in its performance. Furthermore, when multiple source domains are available, existing methods struggle to distinguish domains that are more dissimilar to the target, often treating all domains equally. This inability to prioritize relevant sources can lead to *negative transfer*, where the knowledge transferred from additional source domains hurt the performance of the target domain instead of improving it.

As an example, in DR, the objective proxy is typically the average performance $\mathbb{E}_{\omega \sim \mathbf{Unif}(\Omega)}[V_{P_\omega}^\pi]$ over an index set $\Omega$. As mentioned, this proxy often fails to accurately reflect the target domain's performance in general settings. It may overestimate the target performance, resulting in an overly optimistic policy. Deploying such a policy in the target domain can lead to suboptimal or even harmful outcomes. Additionally, the uniform distribution used in domain randomization cannot effectively exclude harmful source domains, increasing the risk of negative transfer.

## 4.2. Pessimism Principle for Transfer Learning

Identifying these two limitations of prior works, we aim to develop a more informative proxy that satisfies the following objectives: **(1).** It establishes concrete connections to the target domain performance, ensuring that optimizing the proxy guarantees the target performance. **(2).** It ensures the resulting policy remains conservative, thereby avoiding undesired decisions and their associated consequences.

To achieve this, we propose the pessimism principle framework, where we construct a conservative proxy $f(\pi) \leq V_{P_0}^\pi$, $\forall \pi$, then we transfer its optimal policy $\pi_f = \arg\max_\pi f(\pi)$. By optimizing this conservative proxy, we guarantee an optimized lower bound on the target domain performance, ensuring strong performance while mitigating potential negative outcomes. We then characterize its effectiveness as follows (proof deferred to Appendix 4.1).

**Lemma 4.1.** *(Effectiveness of Pessimism for Transfer Learning) Denote the level of pessimism of proxy $f(\pi)$ by*

$$\zeta^\pi \triangleq V_{P_0}^\pi - f(\pi) \geq 0.$$

*Then, the transferred policy $\pi_f \triangleq \arg\max_\pi f(\pi)$ has the following sub-optimality gap under the target environment:*

$$V_{P_0}^{\pi^*} - V_{P_0}^{\pi_f} \leq \|\zeta\| \triangleq \max_\pi \zeta^\pi. \tag{4}$$

The result demonstrates that optimizing a pessimistic proxy guarantees the target domain performance in the worst-case scenario, thereby avoiding undesired decisions—a feat that is unattainable with prior transfer learning methods. More importantly, the result establishes a **monotonic dependence** between the sub-optimality gap and the level of pessimism: as the level of pessimism $\|\zeta^\pi\|$ decreases, the transferred policy achieves smaller performance gap. This observation highlights a key advantage of the pessimism principle: an automatic improvement from an enhanced proxy. As long as the proxy remains conservative, a higher proxy value directly translates into better performance, eliminating the risk of overestimation as with previous methods.

**Remark 4.2.** *As detailed in the methods presented later, $\|\zeta^\pi\|$ can be controlled via the construction of local uncertainty sets. Consequently, the suboptimality gap can be bounded in terms of domain similarities, which is reasonable and subject to the problem nature. Moreover, if additional information becomes available (e.g., a small set of target domain data or allowance of explorations), the radii of the uncertainty sets constructed can be tightened, leading to improved transfer performance.*

In the following, we develop a robust RL based framework to provide concrete answers for the critical questions:

> *How can we construct effective conservative proxies for pessimistic transfer, and how can we further improve the effectiveness by enhancing proxy values?*

## 4.3. A Robust RL Based Pessimism Principle

Robust RL is inherently pessimistic when tackling environment uncertainty. Specifically, when the environment of interest lies within the defined uncertainty set, the robust value function naturally provides a conservative lower bound on performance. This property inspires our studies of robust RL in transfer learning, as constructing an uncertainty set that includes the target domain is not challenging, and thus we can obtain a conservative proxy. Generally, the learner has some prior knowledge about the domain similarities to motivate transfer learning, and such similarities are usually captured in terms of the source and target domain kernels. For instance, a common characterization is the distance/divergence between the source and target

kernels, e.g., (Qu et al., 2024). Based on this information, an uncertainty set can be constructed to include the target domain kernel, and result in its robust value function as a conservative proxy.

In the case of multi-domain transfer, we similarly assume that the distance between any source domain $P_k, k > 0$ and the target domain $P_0$ is upper bounded by a known $\Gamma$: $D(P_0, P_k) \leq \Gamma$ [3].

**Remark 4.3.** *Generally, such knowledge can be obtained through domain experts, or estimated from a small amount of target data. Moreover, our method remains applicable even without prior knowledge, by setting $\Gamma$ to a known upper bound on distributional distance (e.g., total variation between any two distributions is at most 1). While this yields an over-conservative proxy, it still helps prevent significant drops when no similarity information is available - a guarantee that DR methods do not offer.*

For each source domain $P_k$, we can construct a local uncertainty set as $\mathcal{P}_k = \bigotimes_{s,a} (\mathcal{P}_k)_s^a$, with

$$(\mathcal{P}_k)_s^a = \{q \in \Delta(\mathcal{S}) : D(q, (P_k)_s^a) \leq \Gamma_s^a\},$$

where $\Gamma_s^a \geq \Gamma$. In this way, $P_0 \in \mathcal{P}_k$ and each robust value function is a conservative proxy: $V_{\mathcal{P}_k}^\pi \leq V_{P_0}^\pi, \forall k \in \mathcal{K}, \forall \pi$. It is then of great importance to utilize these local uncertainty sets and robust value functions to construct conservative proxies with high value, to enable effective transfer learning. Several potential straightforward constructions are as follows: **(1).** Robust DR: $\bar{V}^\pi \triangleq \frac{\sum_{k=1}^K V_{\mathcal{P}_k}^\pi}{K}$; **(2).** Proximal robust DR: $V_{\bar{\mathcal{P}}}^\pi$, where $(\bar{\mathcal{P}})_s^a = \left\{ q \in \Delta(\mathcal{S}) : D\left(q, \bar{P}_s^a = \frac{\sum_k (P_k)_s^a}{K}\right) \leq \Gamma_s^a \right\}$; **(3).** Maximal robust value function: $\max_k V_{\mathcal{P}_k}^\pi$; **(4).** Robust value function of the intersected uncertainty set $\cap_k (\mathcal{P}_k)_s^a$.

It can be shown that all of these proxies are conservative estimations of $V_{P_0}^\pi$, however, none of these proxies can be efficiently solved in a distributed setting. Specifically, (robust) DR is generally intractable and is approximated by a proximal DR objective (Jin et al., 2022). Meanwhile, solving other proxies may require sharing local source domain models or data, which is often impractical or undesirable in a distributed setting. Therefore, we propose two conservative proxies based on robust RL, and show that they can be efficiently optimized under the distributed setting.

**Remark 4.4.** *Note that in certain cases, such as when the prior knowledge used to constructed the uncertainty set is inaccurate, our method may lead to conservative solutions that could be outperformed by DR. Nonetheless, our main contribution is to provide robust performance across all scenarios. Considering the inherent necessaries of conservativeness of robust transfer learning, such a guarantee is*

---

[3]We assume a universal bound $\Gamma$ for simplicity, but our methods can be directly extended to the case with different similarities $\Gamma_k$.

*particularly valuable in safety-critical or high-stakes applications.*

# 5. Averaged Operator Based Proxy

As discussed, while robust RL-based conservative proxies are straightforward to construct, it is crucial to develop ones that can be efficiently optimized in a distributed setting, where each source domain avoids sharing its underlying MDPs or raw data. Constructing proxies that rely on the underlying environments, as discussed earlier, may be impractical due to data privacy concerns in distributed frameworks.

Inspired by distributed and federated learning (Wang et al., 2024a), we propose an operator-based proxy that enables the design of distributed algorithms where each local domain shares only updated Q-tables instead of raw data. Specifically, we introduce an averaged operator-based proxy, which extends the design of vanilla federated learning to our pessimism transfer learning framework.

## 5.1. Averaged Operator Based proxy

By definition (2), we denote the robust Bellman operators for the $k$-th source domain by $\mathbf{T}_k^\pi$. Based on these, we construct an averaged operator and our conservative proxy.

**Lemma 5.1.** *Define the averaged robust Bellman operator as $\mathbf{T}_{\mathrm{AO}}^\pi Q(s,a) = \frac{1}{K} \sum_{k=1}^K \mathbf{T}_k^\pi Q(s,a)$. Then, $\mathbf{T}_{\mathrm{AO}}^\pi$ has a unique fixed point $Q_{\mathrm{AO}}^\pi$.*

We use its fixed point to construct a conservative proxy, named the averaged operator based proxy for any given policy $\pi$: $V_{\mathrm{AO}}^\pi(s) \triangleq \sum_{a \in \mathcal{A}} \pi(a|s) Q_{\mathrm{AO}}^\pi(s,a)$. We first show this proxy is indeed conservative.

**Theorem 5.2.** *(Pessimism of the proxy) The averaged operator based proxy is conservative: $V_{\mathrm{AO}}^\pi \leq V_{P_0}^\pi, \forall \pi$.*

The result indicates that the averaged operator based proxy effectively adopts the pessimism principle and provide a lower bound for the target domain performance.

**Remark 5.3.** *The construction of the proxy is not an direct extension of DR or multi-task RL. As mentioned, the DR method optimizes $V_{\bar{P}}^\pi$ under the averaged environment $\bar{P} = \frac{\sum_{k=1}^K P_k}{K}$. In the robust setting, its direct extension becomes $V_{\bar{\mathcal{P}}}^\pi$, the proximal robust DR proxy in Section 4.3. Due to the non-linearity of robust value functions w.r.t. the nominal kernel, $V_{\bar{P}}^\pi$ does not coincide with the fixed point of the averaged robust Bellman operator, as in the non-robust case. However, we can show that our averaged operator based proxy is more effective than the robust DR.*

We further provide the following result, indicating that for the two most commonly used distance measures, $V_{\mathrm{AO}}^\pi$ is more effective than proximal robust DR.

**Proposition 5.4.** *If uncertainty sets are defined through total variation or Wasserstein distance[4], then $V_{\mathcal{P}}^{\pi} \leq V_{\text{AO}}^{\pi}$.*

**Remark 5.5.** *Notice that according to the proof of Proposition 5.4, the result holds provided that $\sigma_{\mathcal{P}_s^a} - \frac{1}{K}\sum_{k=1}^{K} \sigma_{(\mathcal{P})_s^a}(V) \leq 0$. In Addition, for $l_p$-norm, the support function has a duality: $\sigma(V) = \max_{\alpha}\{PV_{\alpha} + f(\Gamma, V_{\alpha})\}$ for some function $f$ (Clavier et al., 2024). Thus $\sigma_{\mathcal{P}_s^a} - \frac{1}{K}\sum_{k=1}^{K}\sigma_{(\mathcal{P})_s^a}(V) = \max_{\alpha}\{\bar{P}V_{\alpha} + f(\Gamma, V_{\alpha})\} - \frac{1}{K}\sum_{k=1}^{K}\max_{\alpha}\{P_k V_{\alpha} + f(\Gamma, V_{\alpha})\} \leq 0$, and the similar result still holds.*

The result shows that for the most widely used distance measures, the averaged operator based proxy is less conservative than the proximal robust DR, hence is more effective and is more likely to result in a better policy, due to the monotonic improvement property. It is then of interest to design a concrete algorithm to optimize it, i.e., to obtain the policy $\pi_{\text{AO}} \triangleq \arg\max_{\pi} V_{\text{AO}}^{\pi}$.

### 5.2. Efficient and Distributed Algorithm Design

Although $\mathbf{T}_{\text{AO}}^{\pi}$ is a contraction and $V_{\text{AO}}^{\pi}$ can be estimated exponentially fast, it can be impractical to estimate and compare $V_{\text{AO}}^{\pi}$ over all policies to find the AO-optimal policy $\pi_{\text{AO}} = \arg\max_{\pi} V_{\text{AO}}^{\pi}$. To efficiently optimize it, we construct an averaged optimal Bellman operator as follows, and develop fundamental characterizations for it and $\pi_{\text{AO}}$.

**Theorem 5.6.** *Recall $\mathbf{T}_k$ being the optimal robust Bellman operator for the $k$-th source domain as Equation (3). Define the averaged optimal Bellman operator as*

$$\mathbf{T}_{\text{AO}}Q(s,a) \triangleq \frac{1}{K}\sum_{k=1}^{K}\mathbf{T}_k Q(s,a). \qquad (5)$$

*Then: (1). $\mathbf{T}_{\text{AO}}$ is a contraction and has a unique fixed point $Q_{\text{AO}}$; (2). Let $\pi_*(s) \triangleq \arg\max_{a \in \mathcal{A}} Q_{\text{AO}}(s,a)$, then $\pi_* = \arg\max_{\pi} V_{\text{AO}}^{\pi}$.*

The results imply that the optimal policy $\pi_{\text{AO}}$ can be obtained from the unique fixed point of the averaged optimal Bellman operator, hence it suffices to learn $Q_{\text{AO}}$ by recursively applying $\mathbf{T}_{\text{AO}}$. More importantly, due to the average structure over every source domain, the global learner does not require knowledge of each source domain, but only the update from each source domain, and hence can be trained in a distributed and efficient scheme. Specifically, each local source domain maintains a local $Q$-table to capture its Bellman operator, and sends the updated table to the global learner. After an average aggregation over all domains, a global $Q$-table is then sent back to each local domain.

In practice, the exact source domains may be unknown, but extensive samples from them are available, as the source

domains are more free to explore. We hence assume that each local source domain can obtain an unbiased estimate of its robust Bellman operator: for any vector $Q$, an unbiased estimate $\hat{\mathbf{T}}_k Q$ is available to each local agent. This is a mild assumption, as there are different ways to construct an unbiased estimator of the robust Bellman operator, e.g., (Wang et al., 2023c; Kumar et al., 2023b; Yang et al., 2023; Liu et al., 2022). We also note that due to the linear structure of $\mathbf{T}_{\text{AO}}$, its unbiased estimation can be directly constructed as the average of all local estimations as $\hat{\mathbf{T}}_{\text{AO}} = \frac{\sum_{k=1}^{K}\hat{\mathbf{T}}_k}{K}$.

On the other hand, to further avoid extensive communication, the frequency of aggregation can be reduced. Specifically, we update the local estimation of each source domain individually for $E$ steps, and then aggregate all the domains once. Such a learning scheme is also used in federated or distributed learning, e.g., (Wang et al., 2024a; Jin et al., 2022; Wang et al., 2024c; Woo et al., 2023).

We then present our Averaged Operator based multi-domain transfer (MDTL-Avg) algorithm as follows. The algorithm consists of two parts: local update to the local optimum, and global aggregation. Each local agent updates its own Q table with its data, and after $E$ steps, a global aggregation unifies all local Q tables. The convergence proof can be similarly decomposed in to the local (controlled by the local Bellman operator) and the global parts (controlled by aggregation). As mentioned, our algorithm does not require the knowledge

---

**Algorithm 1** MDTL-Avg and MDTL-Max Algorithms

---

1: **Initialization:** $0 \leq Q_k \leq \frac{1}{1-\gamma}, \forall k = 1, ..., K$
2: **for** $t = 0, ..., T-1$ **do**
3:     **for** $k = 1, ..., K$ **do**
4:         $Q_k(s,a) \leftarrow (1-\lambda)Q_k(s,a) + \lambda\hat{\mathbf{T}}_k(Q_k)(s,a),$
        $\forall(s,a) \in \mathcal{S} \times \mathcal{A}$       /Local Update/
5:     **end for**
6:     **if** $t \equiv 0 (\text{mod } E)$ **then**
7:         **for** $s \in \mathcal{S}, a \in \mathcal{A}$ **do**
8:             For MDTL-Avg:
9:             $Q(s,a) \leftarrow \frac{1}{K}\sum_{k=1}^{K}Q_k(s,a)$
10:          For MDTL-Max:
11:          $Q(s,a) \leftarrow$ **Max-Aggregation of** $\{Q_k(s,a)\}$
12:          $Q_k(s,a) \leftarrow Q(s,a), \forall k$   /Synchronize/
13:         **end for**
14:     **end if**
15: **end for**

---

of each source domain, but only an unbiased estimator of the updated $Q$-table, which is more applicable under large-scale environments and better preserve privacy. Moreover, we reduce the aggregation frequency by $E$ times, greatly enhancing the communication efficiency.

We then develop the convergence analysis of Algorithm 1.

**Theorem 5.7.** *Let $E - 1 \leq \min \frac{1}{\lambda}\{\frac{\gamma}{1-\gamma}, \frac{1}{K}\}$, and $\lambda =$*

---

[4]See Appendix D.3 for definitions.

$\frac{4\log^2(TK)}{T(1-\gamma)}$. *Run Algorithm 1 for T steps. If $\mathbb{E}[\hat{\mathbf{T}}_k] = \mathbf{T}_k$, then it holds that*

$$\left\| \mathbb{E}\left[ Q_{\text{AO}} - \frac{\sum_{k=1}^{K} Q_k}{K} \right] \right\| \leq \tilde{\mathcal{O}}\left( \frac{1}{TK} + \frac{(E-1)\Gamma}{T} \right). \quad (6)$$

As the result shows, our algorithm enjoys a partial linear speedup, implying the efficiency of our algorithm for multi-domain transfer. The second term measures the trade-off between the convergence rate and communication cost, which is also observed in federated learning (FL) with heterogeneous environments (Wang et al., 2024a;c). We note that the convergence rate of our stochastic algorithm still enjoys a partial linear speedup, which matches the minimax optimal robust up to polylog factors (Vershynin, 2018). It hence illustrates the efficiency of our algorithm, in terms of convergence rate and communication cost.

Our algorithm design and convergence result thus provide the first effective, efficient multi-domain transfer method with a conservative lower-bound guarantee.

# 6. Avoiding Negative Transfer: the Minimal Pessimism Principle

While our averaged operator-based method provides a conservative yet effective proxy for target domain performance, it remains vulnerable to a key challenge in multi-domain transfer: negative transfer. Similar to existing TL methods, the averaged operator-based proxy treats all source domains equally and cannot identify or down-weight the less relevant ones. Inspired by recent advances in personalized FL (Arivazhagan et al., 2019; Fallah et al., 2020; Deng et al., 2020; Tan et al., 2022), the global learner should assign higher weights to source domains that are more similar to the target domain. However, identifying these similar domains without additional information remains challenging.

Our pessimism principle offers a solution to this problem. As shown in Lemma 4.1, a less conservative proxy leads to a better-performing policy. Intuitively, more distinct source domains yield smaller robust value functions, allowing us to assign higher weights to domains with higher robust values to prioritize the more relevant ones. A straightforward proxy would be $\max_{k \in \mathcal{K}} V_{\mathcal{P}_k}^{\pi}$, which uses the maximum robust value function as the proxy. However, this approach may not be computationally efficient in a distributed setting. In this section, we propose a refined proxy that addresses these limitations, achieving both higher effectiveness and computational efficiency.

We define the minimal pessimism operator and the optimal minimal pessimism operator as $\mathbf{T}_{\text{MP}}^{\pi} Q(s,a) \triangleq \max_{k \in \mathcal{K}} \mathbf{T}_k^{\pi} Q(s,a)$, and $\mathbf{T}_{\text{MP}} Q(s,a) \triangleq \max_{k \in \mathcal{K}} \mathbf{T}_k Q(s,a)$, based on which

we define and characterize the minimal pessimism proxy.

**Theorem 6.1.** *The following results hold:*

*(1) Both $\mathbf{T}_{\text{MP}}^{\pi}$ and $\mathbf{T}_{\text{MP}}$ are $\gamma$-contractions, and have unique fixed points, $Q_{\text{MP}}^{\pi}$ and $Q_{\text{MP}}$, respectively.*

*(2) For any policy $\pi$, let $V_{\text{MP}}^{\pi}(s) = \sum_{a \in \mathcal{A}} \pi(a|s) Q_{\text{MP}}^{\pi}(s,a)$, then $V_{\mathcal{P}_k}^{\pi} \leq V_{\text{MP}}^{\pi} \leq V_{P_0}^{\pi}$. Moreover, $V_{\text{AO}}^{\pi} \leq V_{\text{MP}}^{\pi}$.*

*(3) Let $\pi_*(s) = \arg\max_{a \in \mathcal{A}} Q_{\text{MP}}(s,a)$, then $Q_{\text{MP}} = Q_{\text{MP}}^{\pi_*}$ and $\pi_* = \arg\max_{\pi} Q_{\text{MP}}^{\pi}$.*

Similar to the previous section, our construction is based on a minimal pessimism principle operator, which is shown to be a contraction and admits a unique fixed point. Moreover, our results imply that the proxy constructed is conservative, and more importantly, it is better than the robust value function of any source domain and the averaged operator based proxy. This hence avoids negative transfer by ruling out the more conservative source domains. As discussed, this less conservative proxy can result in a better estimate and is more likely to yield a better transferred policy.

The third result implies a practical approach to optimize $Q_{\text{MP}}^{\pi}$, through obtaining the fixed point $Q_{\text{MP}}$ of the optimal operator $\mathbf{T}_{\text{MP}}$. As it is a contraction, recursively applying it will converge to $Q_{\text{MP}}^{\pi}$. Similarly, considering the partial information from each source domain, we design a distributed algorithm to achieve the fixed point as Algorithm 1.

Notably, in MDTL-Max as Algorithm 1, the aggregation is obtained through some function **Max-Aggregation**. When each local Bellman operator is exact, i.e., $\hat{\mathbf{T}}_k = \mathbf{T}_k$, we simply set **Max-Aggregation of** $\{Q_k\}_{k=1}^K$ to be $\max_k\{Q_k\}$, and the resulting algorithm will converge to $Q_{\text{MP}}$ (see Theorem E.2). However, when each local operator is stochastic, such an aggregation may result in biased estimation of $\mathbf{T}_{\text{MP}}$. Specifically, even if $\hat{\mathbf{T}}_k$ are unbiased estimations, the straightforward application of the maximum function results in a biased estimation: $\mathbb{E}[\max_{k \in \mathcal{K}} Q_k] \neq \max_{k \in \mathcal{K}} \mathbb{E}[Q_k]$.

To address this issue, it is essential to design an alternative maximal aggregation scheme that ensures convergence despite the presence of estimation errors. We propose an aggregation strategy to overcome this challenge. Specifically, we adopt the multi-level Monte-Carlo method (Blanchet & Glynn, 2015; Blanchet et al., 2019; Wang & Wang, 2022) to construct an operator $\hat{\mathbf{M}}_{\text{MLMC}}$ of the maximal aggregation based on estimated local operators. We defer the detailed construction to Appendix E.1.

**Lemma 6.2.** $\mathbb{E}[\hat{\mathbf{M}}_{\text{MLMC}}(\{Q_k\}_{k=1}^K)] = \max_{k \in \mathcal{K}} \mathbb{E}[Q_k]$.

We hence constructed an unbiased estimation of the global minimal pessimism operator, which can then be adopted to implement MDTL-Max as Algorithm 1.

**Remark 6.3.** *The MLMC module for MDTL-Max will in-*

*crease the sample/computation complexity, which can be viewed as the price of improving effectiveness. Nevertheless, it can be reduced along two potential directions: One is to control the level number through techniques like threshold-MLMC (Wang et al., 2024g). Although it results in a biased estimation, the bias can be controlled and hence still implies convergence (see Appendix B.4). Another one is to relax the uncertainty set constraint. For example, for total variation, the relaxation results in the solution $P_0 V - \Gamma Span(V) \leq \sigma(V)$ (Kumar et al., 2023a), which is much easier to compute and remains conservative. Reducing the aggregation frequency is an alternative way to reduce the computational cost, but also introduces a trade-off in the convergence rate, as we will show in the following results.*

We can then incorporate such a multi-level Monte-Carlo construction into Algorithm 1 as an unbiased update step. We further characterize its convergence.

**Theorem 6.4.** *Let $E - 1 \leq \min \frac{1}{\lambda} \{ \frac{\gamma}{1-\gamma}, \frac{1}{K} \}$, $\lambda = \frac{4 \log^2(TK)}{T(1-\gamma)}$, and adopt MLMC-Max-Aggregation. Then after $T$ steps,*

$$\left\| \mathbb{E} \left[ Q_{\text{MP}} - \max_{k \in \mathcal{K}} Q_k \right] \right\| \leq \tilde{\mathcal{O}} \left( \frac{1}{TK} + \frac{(E-1)\Gamma}{T} \right). \quad (7)$$

The result implies the convergence of our algorithm in a more practical setting. More importantly, our algorithm converges to a more effective proxy than the averaged one, which potentially results in a better transferred policy due to our pessimism principle. Also, our algorithm enjoys a nearly optimal convergence rate, similar to Algorithm 1.

# 7. Additional Discussion

In the previous sections, we demonstrated that our pessimism principle-based framework offers two key advantages: (1) it guarantees a well-performing lower bound on the target domain performance, avoiding severe undesired outcomes, and (2) it establishes a monotonic relationship between the pessimism level and transfer effectiveness, allowing us to improve transfer performance and mitigate negative transfer. In this section, we highlight two additional advantages: improved robustness and scalability.

**Robustness.** By optimizing a conservative estimation of the target domain's performance, our framework inherently enhances the robustness of the transferred policy: it ensures that the policy performs well even under model uncertainties within the target domain, such as non-stationary parameters. We show the following result to justify its robustness:

**Proposition 7.1.** *There exists some connected uncertainty set $\tilde{\mathcal{P}}$ such that $P_0 \in \tilde{\mathcal{P}}$, and $V_{P_0}^\pi \geq V_{\tilde{\mathcal{P}}}^\pi \geq V_{\text{MP}}^\pi \geq V_{\text{AO}}^\pi$.*

This result implies that the proxies we construct are also

conservative bounds for the robust value function with respect to some uncertainty set around $P_0$. As a result, the transferred policy is guaranteed to perform well despite potential uncertainties in the target domain, thereby enhancing its generalizability against continuous perturbations of the environment. Specifically, the policy will also perform effectively in environments similar to $P_0$. We further validate this robustness numerically in our experiments.

**Scalability.** Our proposed algorithms are scalable, enabling effective transfer in large-scale problems. Firstly, our algorithms can be implemented in a model-free manner to enhance computational and memory efficiency. In particular, any model-free algorithm for robust RL can be integrated into the local update step in Algorithm 1. To demonstrate this, we design a model-free variant of Algorithm 1 in Appendix G, accompanied by a convergence guarantee:

**Theorem 7.2.** *(Informal) Denote the output of the algorithms with suitable step sizes $\lambda$ and total steps $T$ by $Q^T$. Let $\Delta_T := Q_{\text{AO}} - Q^T$ and $\Delta_T := Q_{\text{MP}} - Q^T$ for MDTL-Avg and MDTL-Max, respectively. With probability at least $1 - \delta$, we have*

$$\|\Delta_T\| \leq \tilde{\mathcal{O}} \left( \frac{(E-1)\Gamma}{T(1-\gamma)^3} + \frac{\log \frac{SATK}{\delta}}{(1-\gamma)^3 \sqrt{TK}} \right). \quad (8)$$

Notably, while previous methods, such as DR, lack concrete algorithms, our approach is both effective and scalable, offering a significant advantage.

Secondly, our method is not restricted to tabular setting, and can be integrated with function approximation or policy gradient. For example, when combine function approximation for robust RL (Zhou et al., 2023) or robust policy gradient (Wang & Zou, 2022), the global step becomes parameter aggregation as in (Jin et al., 2022).

# 8. Experimental Results

In this section, we aim to verify the effectiveness of our pessimism principle, and its ability to avoid negative transfer. More details and additional experiments results are provided in Appendix B.

## 8.1. Effectiveness of Pessimism Principle for Transfer Learning

We develop our simulation on the recycling robot problem (Sutton & Barto, 2018; Wang et al., 2023c). In this problem, a mobile robot powered by a rechargeable battery is tasked with collecting empty soda cans. The robot operates with two battery levels: low and high. It has two possible actions: (1) search for empty cans; (2) remain stationary and wait for someone to bring it a can. When the robot's battery is low (high), it has a probability of $\alpha(\beta)$ of finding an empty

can and maintaining its current battery level, receiving some high reward. If the robot searches but does not find any, it will deplete its battery completely and receive a large penalty. If the robot chooses to wait, it will remain at the same battery level and receive a relatively small reward.

In our experiments, we first compare our methods with the non-robust corresponding methods, including **proximal non-robust DR** and **maximal aggregation algorithm** (see Algorithm 2). We run 5 trials with different random seeds for each method and report means and standard deviations of values over training steps in Figure 1. We use a constant step size $= 0.1$, and total training steps $T = 5000$. We set the target environment as $\alpha = \beta = 0.1$, pre-specified radius of the uncertainty set $R = 0.8$. Then, we randomly generate $K = 7$ distinct source domains with $\alpha_k, \beta_k \in [0.85, 0.9]$ to construct the uncertainty set. Other parameters $E = 1$, and $\gamma = 0.95$. As the results show in Figure 1-left, both of the non-robust baselines achieve a low reward in the target domain, as they are overly optimistic and decide to search under both battery levels, whereas our pessimism principle methods tend to wait conservatively, avoiding undesired decisions and severe consequences.

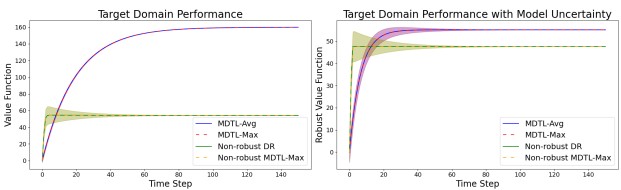

*Figure 1.* Recycling Robot Problem

To illustrate the robustness and generalizability of our methods against target model uncertainty, we evaluate the robust value functions of the learned policies against an uncertainty set centered at the target domain, which represents the worst-case performance under the model uncertainty. Figure 1-right shows that both of our pessimistic methods outperform the non-robust baseline, indicating that our approach can maintain a high performance, even with uncertainty or perturbations in the target domain. We also conduct an ablation study to compare the robust value functions under different target domain uncertainty levels in Appendix B.1, to further illustrate the enhanced robustness of our approaches.

In summary, our method outperforms non-robust baselines, achieving a **195.03%** performance improvement in the target domain and maintaining a **15.95%** improvement even under uncertainty. The experiment results hence verify that pessimism principle is more effective, especially when the source domains are relatively distinct from the target domain, or when model uncertainty exists in the target domain.

### 8.2. Experiments on Negative Transfer

We further tested our MDTL-Avg and MDTL-Max algorithms on FrozenLake Gym environments, aiming to vali-

date whether MDTL-Max can effectively mitigate negative transfer. The FrozenLake Gym environment provides an explicitly known transition kernel. This allows us to precisely control the distance between source domains and the target domain, thereby offering more rigorous and interpretable results. In this example, we manually perturb the agent using total variation distances from the default model: $D = [0.01, 0.02, 0.3]$. The source domain with $D = 0.3$ is intentionally designed to represent a domain that differs significantly to the target, potentially causing negative transfer. For each algorithm, we run experiments across 10 random seeds. For each seed, the policy is evaluated over 10 independent episodes to compute the average return. As shown in Figure 2, MDTL-Max effectively leverages the most informative source domain, and hence avoids negative transfer.

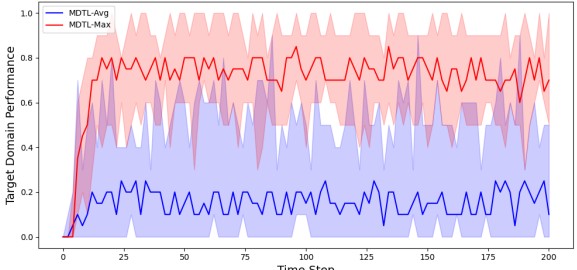

*Figure 2.* Negative Transfer under FrozenLake Gym environment

## 9. Conclusion

In this paper, we studied zero-shot transfer reinforcement learning and identified two critical limitations of existing methods: the lack of guarantees for the safety and performance of transferred policies and the inability to mitigate negative transfer when multiple source domains are involved. To overcome these challenges, we incorporate a pessimism principle into TL to conservatively estimate the target domain's performance, and mitigate risk of undesired decisions. Our framework establishes a monotonic dependence between the level of pessimism and target performance, effectively addressing the two identified issues by constructing less conservative estimations. We proposed and analyzed two types of conservative estimations, rigorously characterizing their effectiveness, and developed distributed, convergent algorithms to optimize them. These methods are well-suited for distributed transfer learning settings, offering performance guarantees and privacy protection. Our framework represents a foundational step toward zero-shot transfer RL with theoretical performance guarantees, providing a robust and practical solution for real-world applications.

## Acknowledgment

This work was supported by DARPA under Agreement No. HR0011-24-9-0427 and NSF under Award CCF-2106339.

The authors thank the anonymous reviewers whose constructive comments led to substantial improvement to the paper.

## Impact Statement

This paper presents work whose goal is to advance the field of Machine Learning. There are many potential societal consequences of our work, none which we feel must be specifically highlighted here.

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

## A. Additional Related Works

**Imitation Learning and Policy Distillation:** Imitation learning (IL) and policy distillation are extensively explored TL techniques that enable an agent to learn from demonstrations or from simpler policies. In IL, an agent mimics expert behavior through supervised learning, often using behavioral cloning or inverse RL (Hua et al., 2021; Desai et al., 2020; Kim et al., 2023; Zare et al., 2024). Policy distillation transfers knowledge from a teacher policy to a student policy by matching the teacher's action distribution (Rusu et al., 2015; Yim et al., 2017; Yin & Pan, 2017; Traoré et al., 2019b;a). The success of these approaches, however, rely on the assumption that the source domain's expert or teacher policy is highly aligned with the target domain, which may not always be the case.

**Contextual Reinforcement Learning:** In contextual reinforcement learning (cRL), the transition dynamics are assumed to be governed by an underlying context (Hallak et al., 2015; Modi et al., 2018). This context can represent physical properties—such as wind conditions (Koppejan & Whiteson, 2009) or the length of a pole in a balancing task (Seo et al., 2020; Kaddour et al., 2020; Benjamins et al., 2022)—or more abstract features that characterize environment dynamics (Biedenkapp et al., 2020). Kirk et al. (2023b) identify cRL as particularly relevant for studying zero-shot generalization in RL agents. The cRL framework allows for a systematic and principled analysis of how agents adapt to environmental changes by explicitly defining inter- and extrapolation distributions. Following the evaluation protocol introduced by Kirk et al. (2023b), Benjamins et al. (2022) examined the generalization capabilities of various model-free RL agents on a benchmark that incorporates different physical properties as context information. Their approach assumed a naive utilization of context by directly concatenating it with observations. In contrast, Beukman et al. (2024) proposed a hypernetwork-based method to enable adaptable RL agents. However, these approaches still fail to address the fundamental challenges discussed earlier.

**Federated Learning:** FL is a decentralized RL approach that aims to optimize the overall performance of K agents (McMahan et al., 2017). In this setting, each agent performs local updates based on its own environment and periodically communicates with a central server to aggregate the local models, without sharing raw trajectories with each other. While FL has been extensively studied (Jin et al., 2022; Wang et al., 2024c; Woo et al., 2023; Khodadadian et al., 2022; Wang et al., 2024a), existing work focuses on optimizing the average performance across local environments, rather than addressing the more challenging problem of multi-domain transfer. A fundamental challenge in applying FL to transfer learning lies in the nature of update rules: standard FL methods rely on linear updates (non-robust operators), whereas our proposed methods involve non-linear robust updates account for uncertainties.

**Robust RL:** Robust RL (Iyengar, 2005; Nilim & El Ghaoui, 2004) or DRO formulation (Wiesemann et al., 2014; He et al., 2020) offer a pessimism principle and a robust framework for addressing environmental uncertainty, which optimizes the worst-case performance within an uncertainty set of environments and can offer a theoretically sound lower bound on target domain performance when the target domain lies within this set. However, its application in transfer learning has been under explored, as robust RL often exhibits excessive pessimism, resulting in overly conservative policies for the target domain. Recent advances have extensively studied robust RL as a standalone problem (Wang & Zou, 2021; 2022; Wang et al., 2023b; Badrinath & Kalathil, 2021; Dong et al., 2022; Lu et al., 2024; Liu & Xu, 2024; Yang et al., 2021; Xu et al., 2023; Panaganti & Kalathil, 2022; Shi et al., 2023; Wang et al., 2024b; Panaganti et al., 2022; Yang et al., 2022; Liu et al., 2023; Wang et al., 2024d;e; Zhang et al., 2025; Sun et al., 2024), but its integration into transfer learning remains largely untapped. Leveraging robust RL principles in transfer learning offers an opportunity to develop methods that are not only theoretically grounded but also resilient to uncertainties in target domains. This proposal seeks to explore research directions that harness the strengths of robust RL to create more effective, efficient, and reliable transfer learning frameworks.

## B. Details on Experiments and Additional Experiments

All experiments are conducted on a MacBook Pro configured with an Apple M3 Pro chip, featuring a 11-core CPU and 18GB of unified memory, running macOS Sequoia 15.2. The experiments are performed using Python 3.12 in a Conda environment. Major libraries used include NumPy 2.2.1.

The baseline algorithms we compared are presented as follows. Here, $\hat{\mathbf{T}}_k$ is some unbiased estimation of the non-robust Bellman operator $\mathbf{T}(Q) = r_k + P_k Q$.

---

**Algorithm 2** Proximal Non-Robust DR and Non-Robust MDTL-Max Algorithms

---

1: **Initialization:** $0 \leq Q_k \leq \frac{1}{1-\gamma}, \forall k = 1, ..., K$
2: **for** $t = 0, ..., T - 1$ **do**
3:     **for** $k = 1, ..., K$ **do**
4:        $V_k(s) \leftarrow \max_{a \in \mathcal{A}} Q_k(s, a), \forall s \in \mathcal{S}$
5:        $Q_k(s, a) \leftarrow (1 - \lambda)Q_k(s, a) + \lambda \hat{\mathbf{T}}_k(Q_k)(s, a), \forall(s, a) \in \mathcal{S} \times \mathcal{A}$                     `/Local Update/`
6:     **end for**
7:     **if** $t \equiv 0 (\mod E)$ **then**
8:        **for** $s \in \mathcal{S}, a \in \mathcal{A}$ **do**
9:           `For Proximal DR:`
10:         $Q(s, a) \leftarrow \frac{1}{K} \sum_{k=1}^{K} Q_k(s, a)$
11:           `For Maximal Aggregation:`
12:         $Q(s, a) \leftarrow$ **Max-Aggregation of** $\{Q_k(s, a)\}$
13:         $Q_k(s, a) \leftarrow Q(s, a), \forall k$                               `/Synchronize/`
14:        **end for**
15:     **end if**
16: **end for**

---

## B.1. Ablation Study: experiments on different target domain uncertainty levels

To further evaluate the robustness of our methods, we conduct an ablation study analyzing the impact of different levels of uncertainty. In this experiment, we systematically vary the test uncertainty set radius $R_{\text{test}}$ and measure the optimal performance in the worst case over these uncertainty sets for the learned policies. Four different methods (MDTL-Avg, MDTL-Max, Non-robust DR, and Non-robust MDTL-Max) are evaluated across five levels of uncertainty, defined by the uncertainty set radius $\{0.01, 0.03, 0.05, 0.07, 0.1\}$. All testing experiments are conducted for 3 times. We also provide an optimal policy as a benchmark, which is trained on the target domain using non-robust vanilla value iteration (Sutton & Barto, 2018). The means and standard deviations of robust values of each method are reported over training time steps.

Let's conclude the ablation study before further analyzing the figures and tables: our ablation study provides empirical evidence that our robust learning methods effectively mitigate the impact of target domain uncertainty. Specifically, our approach achieves at least **15.95%** and at most **151.36%** performance dominance over non-robust baselines in the target domain under model uncertainty. Compared to the optimal policy, our method achieves at least **81.32%** and at most **90.66%** of the optimal performance under model uncertainty. These findings validate our pessimism principle, demonstrating its effectiveness in handling domain shifts and model uncertainty.

| Method | $R_{test} = 0.01$ | $R_{test} = 0.03$ | $R_{test} = 0.05$ | $R_{test} = 0.07$ | $R_{test} = 0.1$ |
|---|---|---|---|---|---|
| MDTL-Avg | **134.45** | **101.91** | **82.05** | **68.67** | **55.17** |
| MDTL-Max | **134.45** | **101.91** | **82.05** | **68.67** | **55.17** |
| Non-robust DR | 53.49 | 52.06 | 50.70 | 49.41 | 47.58 |
| Non-robust MDTL-Max | 53.49 | 52.06 | 50.70 | 49.41 | 47.58 |
| Non-robust Single-learn Nominal | 148.30 | 115.48 | 95.35 | 81.71 | 67.84 |

*Table 1.* Robot: Values of 5 Methods under Different Target Domain Uncertainty Levels

Table 1 provides a numerical comparison of the 4 methods across different uncertainty levels. Our methods consistently outperform non-robust baselines across all uncertainty levels.

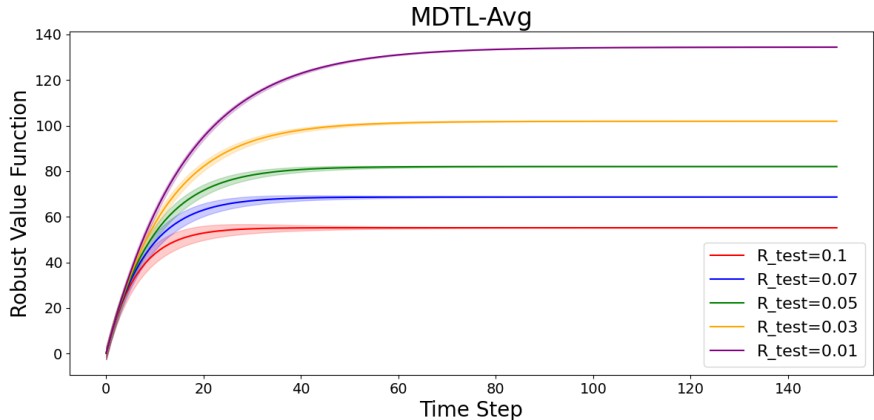

*Figure 3.* Robot: Values of MDTL-Avg under Different Uncertainty Levels

Figure 3 shows the evolution of the robust value function for the MDTL-Avg method as the uncertainty level increases. The results indicate that when the target domain uncertainty is low ($R_{test} = 0.01$), the learned policy achieves high rewards. However, as the uncertainty radius increases, the value function decreases, illustrating the inherent difficulty in handling a more uncertain environment. Despite this, our method still maintains a significantly higher value compared to non-robust baselines, confirming its adaptability to uncertainty.

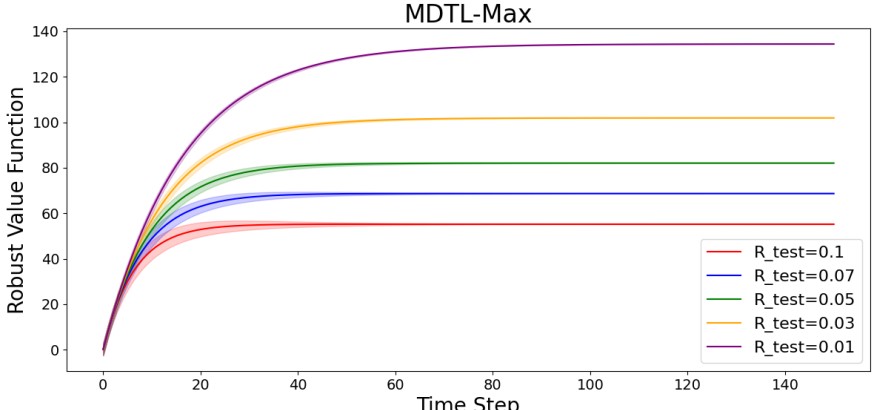

*Figure 4.* Robot: Values of MDTL-Max under Different Uncertainty Levels

Figure 4 presents the results for the MDTL-Max method, demonstrating a similar trend. While performance declines as uncertainty increases, the robust approach still consistently outperforms the non-robust baselines. Notably, both robust methods maintain high robustness under uncertainty, reinforcing the effectiveness of our pessimism principle.

### B.2. Experiments on the HPC Cluster Management Problem

We also validate our methods in an HPC cluster management problem. Each time a new task is submitted, the HPC cluster manager must decide whether to allocate it immediately or enqueue it for later processing. The HPC cluster operates in one of three states: normal, overloaded, or fully occupied, depending on the number of active tasks. Under normal conditions, allocating a task has a probability $p$ of transitioning to the overloaded state while yielding a large reward. When the cluster is overloaded, allocating a task has a probability $q$ of pushing the system into the fully occupied state, where only a small reward is received. In the fully occupied state, all new tasks are automatically enqueued, and no further rewards are obtained.

In our experiments, we set the target environment parameters to $p = q = 0.9$. To model distinct domains, we randomly

generate $K = 7$ source environments with $p_k, q_k \in [0.1, 0.15]$, while keeping all other parameters the same as in the above recycling robot problem. Specifically, we conduct 5 trials for each method using different random seeds, with a fixed step size of 0.1, a total of 5000 training steps, a pre-defined uncertainty set radius of $R = 0.8$, $E = 1$, and $\gamma = 0.95$. The results, presented in Figure 5, where our pessimistic methods consistently outperform the non-robust baselines in target domains, both with and without model uncertainty. In summary, our method outperforms non-robust baselines, achieving a **183.28%** performance improvement in the target domain and maintaining a **11.05%** improvement even under uncertainty. This further confirms the effectiveness of our approach.

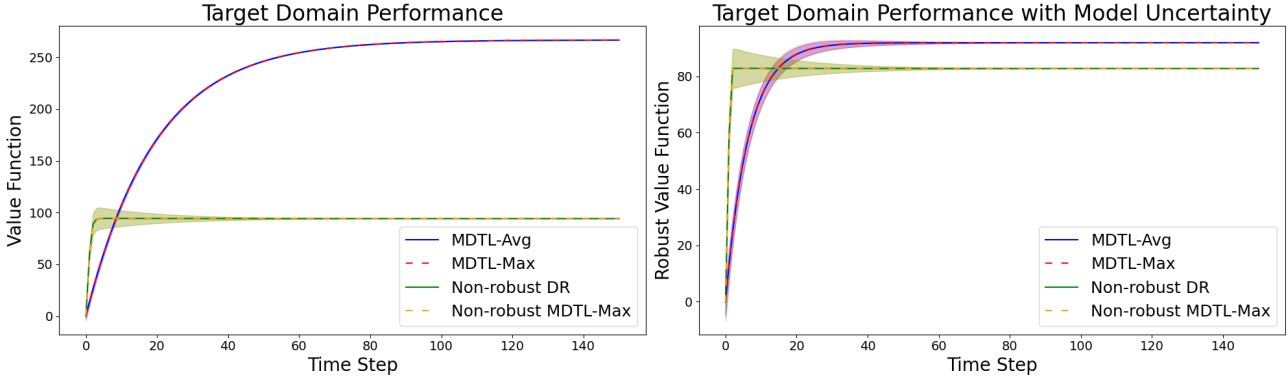

*Figure 5.* HPC Cluster Management Problem

Our ablation study highlights the substantial advantage of our approaches in handling different levels of uncertainty in target domain. Across different levels of model uncertainty, our method consistently outperforms the non-robust baselines. Specifically, our approach demonstrates a minimum improvement of **11.05%** and a maximum of **141.29%** over the best-performing non-robust baseline in the target domain under model uncertainty. Moreover, when compared to the optimal policy, our method retains at least **77.52%** and at most **88.49%** of the optimal performance under varying uncertainty levels. These results further reinforce the effectiveness of our approach.

| Method | $R_{test} = 0.01$ | $R_{test} = 0.03$ | $R_{test} = 0.05$ | $R_{test} = 0.07$ | $R_{test} = 0.1$ |
|---|---|---|---|---|---|
| MDTL-Avg | **224.09** | **169.85** | **136.75** | **114.45** | **91.95** |
| MDTL-Max | **224.09** | **169.85** | **136.75** | **114.45** | **91.95** |
| Non-robust DR | 92.87 | 90.44 | 88.12 | 85.92 | 82.80 |
| Non-robust MDTL-Max | 92.87 | 90.44 | 88.12 | 85.92 | 82.80 |
| Optimal policy | 253.25 | 198.42 | 164.75 | 141.89 | 118.62 |

*Table 2.* HPC: Values of 5 Methods under Different Uncertainty Levels

Table 2 presents a quantitative comparison of the four methods under varying levels of uncertainty. Our approach consistently surpasses the non-robust baselines across all uncertainty settings.

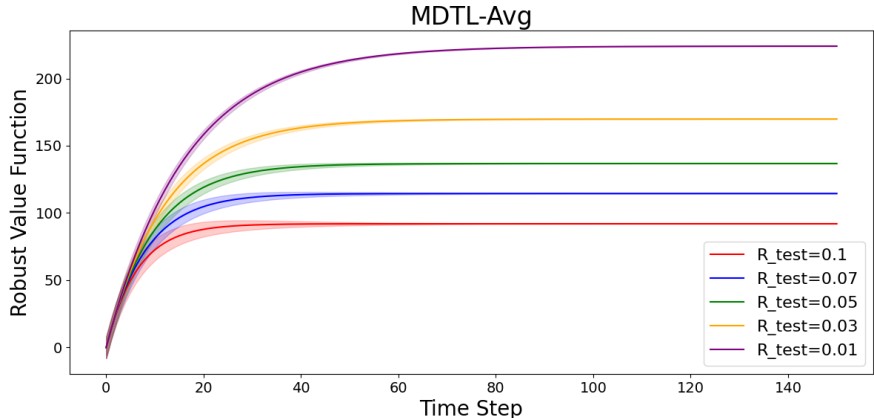

*Figure 6.* HPC: Values of MDTL-Avg under Different Uncertainty Levels

Figure 6 illustrates how the robust value function of the MDTL-Avg method evolves with increasing uncertainty levels. The results show that when target domain uncertainty is minimal ($R_{\text{test}} = 0.01$), the learned policy attains high rewards. As the uncertainty radius expands, the value function declines, reflecting the growing challenge of navigating a more uncertain environment. Nevertheless, our approach consistently outperforms non-robust baselines, demonstrating its resilience in adapting to uncertainty.

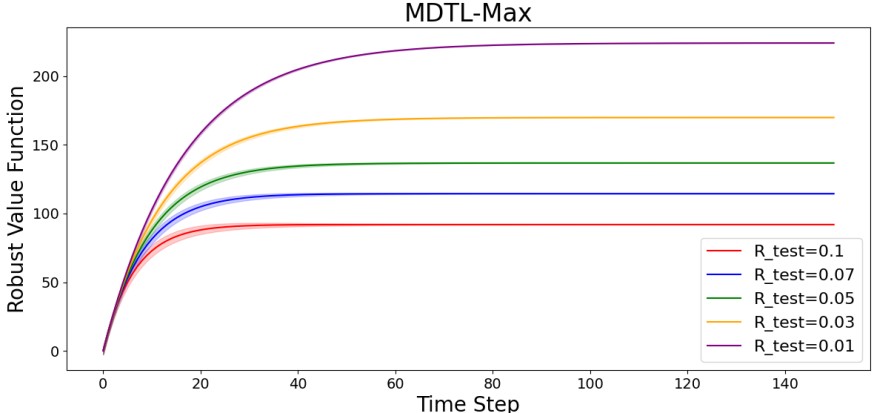

*Figure 7.* HPC: Values of MDTL-Max under Different Uncertainty Levels

Figure 7 depicts the performance of the MDTL-Max method, revealing a similar pattern. Although the value function decreases as uncertainty grows, the robust approach consistently surpasses non-robust baselines. Notably, both robust methods exhibit strong resilience to uncertainty, further validating the effectiveness of our pessimism-driven strategy.

### B.3. Experiments on Dynamic Vehicle Routing Problem

We further implemented our proposed method in a real-world problem: DVRP (Jia et al., 2025), to further validate the proposed method. DVRP extends the classic vehicle routing problem by incorporating dynamic elements such as real-time customer requests, and environmental uncertainty, making it more representative of practical logistics and mobility applications. In our experiments, we consider multiple objectives that reflect both operational efficiency and service quality. Specifically, we aim to minimize overall routing cost, enhance route smoothness by reducing unnecessary detours and zigzag patterns, and improve stability by minimizing abrupt changes in planned routes caused by re-routing in highly dynamic environments. These objectives are crucial in real-world deployments where balancing efficiency and robustness is essential. More details about the DVRP problem setting can be found in (Jia et al., 2025). The implementations of our proposed

methods, MDTL-Avg and MDTL-Max, as well as the baseline algorithm Non-robust DR, are based on (Jia et al., 2025). The baseline Non-robust Single-learn is also adapted from (Jia et al., 2025).

| Method | Route Distance ↓ | Route Smoothness ↑ | Route Stability ↑ |
|---|---|---|---|
| MDTL-Avg (ours) | **4.01** ± 0.3 | 51.76 ± 1.96 | 0.63 ± 0.08 |
| MDTL-Max (ours) | **4.01** ± 0.27 | **52.46** ± 1.45 | **0.68** ± 0.07 |
| Non-robust DR | 4.53 ± 0.52 | 50.32 ± 1.35 | 0.55 ± 0.08 |
| Non-robust Single-learn | 5.20 ± 0.58 | 50.70 ± 0.54 | 0.40 ± 0.03 |

*Table 3.* DVRP: Results of 4 Methods. ↓: smaller is better. ↑: larger is better

Table 2 presents the performance of four methods evaluated on DVRP. Our proposed approaches, MDTL-Avg and MDTL-Max, consistently outperform the baselines across all considered objectives. In terms of Route Distance, both MDTL-Avg and MDTL-Max achieve the lowest average values (4.01), indicating high efficiency in minimizing total travel cost. Regarding Route Smoothness, MDTL-Max achieves the best performance (52.46), suggesting it can effectively reduce unnecessary detours and zigzag behaviors in dynamic environments. Finally, Route Stability, which measures the resilience of planned routes under dynamic re-routing, is significantly higher in both of our methods, with MDTL-Max achieving the highest score of 0.68, demonstrating its robustness to environmental changes.

### B.4. Experiments on Biased Aggregation

The unbiased assumption made in MDTL-Max is to facilitate the convergence analysis. Nevertheless, our convergence results can still be extended to the case that the existing bias can be controlled, e.g., the bias introduced by the max aggregation can be controlled through techniques like threshold-MLMC (Wang et al., 2024g). On the other hand, even if there is bias, as long as the expected proxy is still pessimistic, our transfer learning framework and pessimism guarantees still hold. This indicates that our methods are robust to the bias.

To illustrate this, we develop an experiment on Cartpole to show the effect of bias. As shown in Figure 8, even if there is bias, our proxy is still conservative and our pessimism framework outperforms the baseline, and are hence robust.

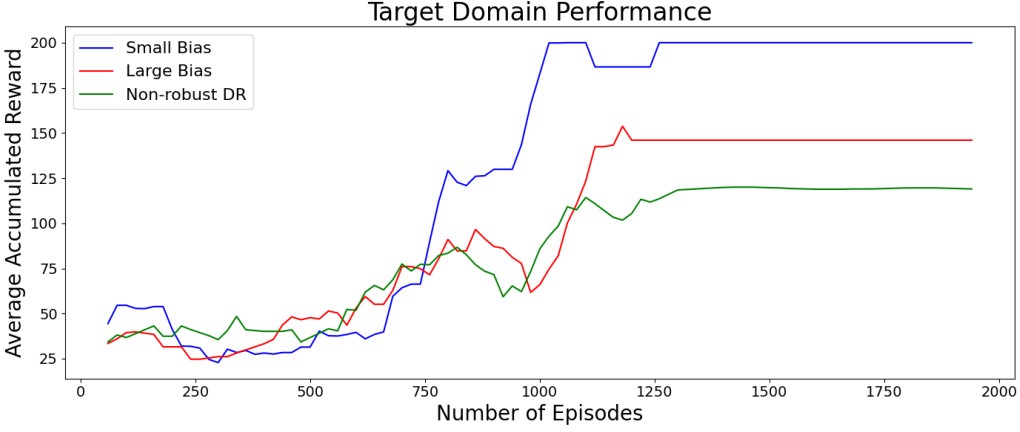

*Figure 8.* Effect of Aggregation Bias for MDTL-Max

### B.5. Additional Experiments on Negative Transfer

We further tested our MDTL-Avg and MDTL-Max algorithms on two additional environments, aiming to validate whether MDTL-Max can effectively mitigate negative transfer. For the CartPole Gym environment, we treat the default environment as the target domain and introduce Gaussian perturbations to the pole length in three source domains, with variances of 0.01, 0.02, and 0.03, respectively. For the recycling robot problem, we set most of the source domains with $\alpha \in (0, 0.1)$ and $\beta \in (0, 0.1)$, in which the optimal policy should be waiting. We add another source domain with $\alpha = \beta = 0.9$, in which the optimal policy should be searching. As shown in Figure 9 and 10, MDTL-Max effectively leverages the most informative

source domain, and hence avoids negative transfer.

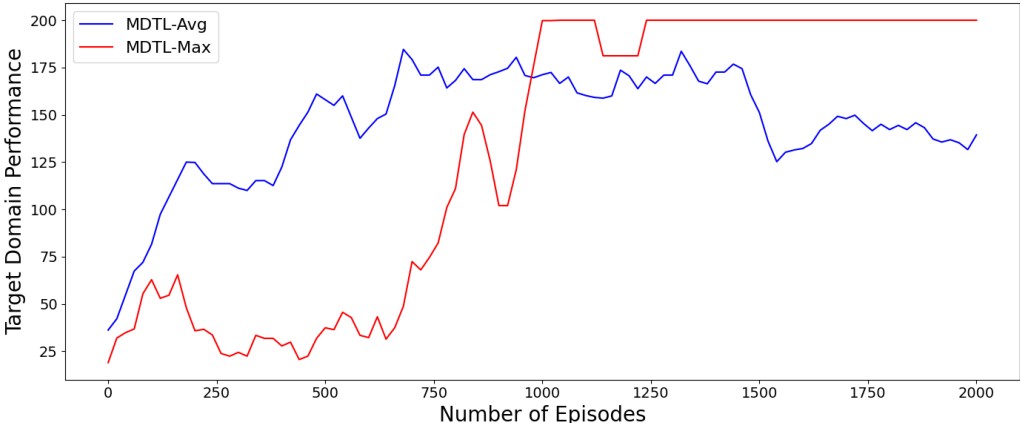

*Figure 9.* Effect of Negative Transfer under CartPole Gym environment

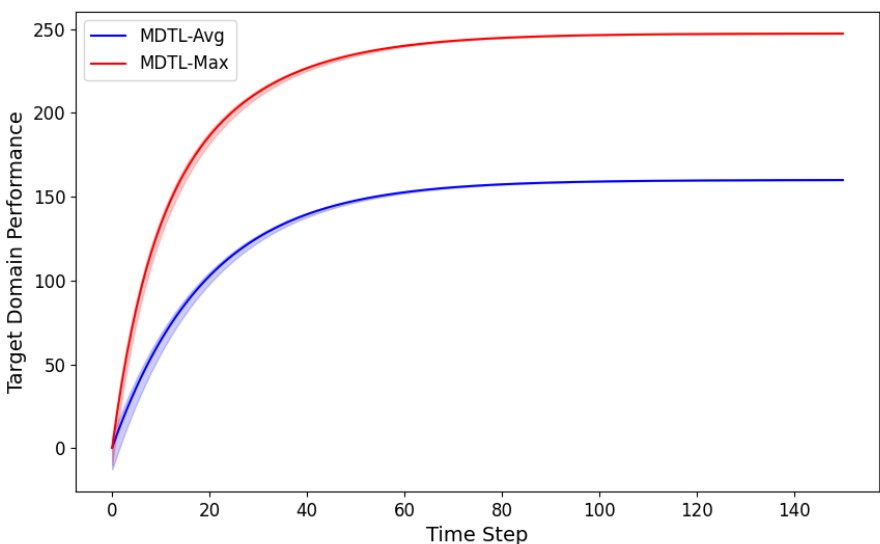

*Figure 10.* Effect of Negative Transfer under Recycling Robot

### B.6. Comparison to Related Robust RL Approaches

Note that our methods are based on distributionally robust RL, where our uncertainty set is constructed to account for the potential distributional shift. We thus compare our proposed methods with other related robust RL approaches. We numerically verify that our methods enjoy both convergence and performance guarantees, and present the results under the recycling robot environment, where we use adversarial action-robust RL (Tessler et al., 2019) and distributionally robust RL (Panaganti & Kalathil, 2022) as baselines.

As shown in Table 4, although adversarial robust and distributionally robust RL outperforms non-robust methods, their performance remain significantly inferior to ours.

| Method | $R_{test} = 0.01$ | $R_{test} = 0.03$ | $R_{test} = 0.05$ | $R_{test} = 0.07$ | $R_{test} = 0.1$ |
|---|---|---|---|---|---|
| MDTL-Avg (ours) | **134.45** | **101.91** | **82.05** | **68.67** | **55.17** |
| MDTL-Max (ours) | **134.45** | **101.91** | **82.05** | **68.67** | **55.17** |
| Adversarial Robust RL | 61.06 | 58.10 | 55.39 | 52.92 | 49.58 |
| Distributionally Robust RL | 53.86 | 53.12 | 52.41 | 51.71 | 50.70 |

*Table 4.* Robot: Robustness comparison of 4 Methods under Different Target Domain Uncertainty Levels

## C. Theoretical Proof for Lemma 4.1

*Proof.* Denote the optimal policy of $f$ by $\pi_f$. Note that

$$V_{P_0}^{\pi^*} - V_{P_0}^{\pi_f}$$

$$= V_{P_0}^{\pi^*} - f(\pi^*) + \underbrace{f(\pi^*) - f(\pi_f)}_{\leq 0, \text{ as } \pi_f = \arg\max_\pi f(\pi)} + \underbrace{f(\pi_f) - V_{P_0}^{\pi_f}}_{\leq 0, \text{ as } f(\pi) \leq V_{P_0}^\pi}$$

$$\leq V_{P_0}^{\pi^*} - f(\pi^*). \tag{9}$$

$\square$

## D. Theoretical Proofs for Averaged Operator Based Transfer

In this section, we provide the proofs of the theoretical results for the averaged operator based transfer method.

### D.1. Proof of Lemma 5.1

Recall that the robust and optimal robust Bellman operators for agent $k$ are defined as

$$\mathbf{T}_k^\pi Q(s,a) = r(s,a) + \gamma \sigma_{(\mathcal{P}_k)_s^a}(V), \tag{10}$$

where $V(s) = \sum_{a \in \mathcal{A}} \pi(a|s)Q(s,a)$. Since $\mathbf{T}_k^\pi$ is a $\gamma$-contraction for any $k \in \mathcal{K}$, then for some $Q_1$ and $Q_2$ we have

$$\|\mathbf{T}_{AO}^\pi Q_1 - \mathbf{T}_{AO}^\pi Q_2\| \leq \frac{1}{K}\sum_{k=1}^K \|\mathbf{T}_k^\pi Q_1 - \mathbf{T}_k^\pi Q_2\| \leq \gamma\|Q_1 - Q_2\|. \tag{11}$$

Hence $\mathbf{T}_{AO}^\pi$ is also a $\gamma$-contraction and has a unique fixed point $Q_{AO}^\pi$, which completes the proof.

### D.2. Proof of Theorem 5.2

In order to verify that $V_{AO}^\pi$ is a lower bound on $V_{P_0}^\pi$ we proceed as follows. Let $V_{AO}^\pi(s) = \sum_{a \in \mathcal{A}} \pi(a|s)Q_{AO}^\pi(s,a)$, since $P_0 \in \mathcal{P}_k$, we have

$$Q_{AO}^\pi(s,a) - Q_{P_0}^\pi(s,a) = \frac{\gamma}{K}\sum_{k=1}^K \sigma_{(\mathcal{P}_k)_s^a}(V_{AO}^\pi) - \gamma(P_0)_s^a V_{P_0}^\pi$$

$$= \gamma\left(\frac{1}{K}\sum_k \sigma_{(\mathcal{P}_k)_s^a}(V_{AO}^\pi) - (P_0)_s^a V_{AO}^\pi\right) + \gamma(P_0)_s^a(V_{AO}^\pi - V_{P_0}^\pi)$$

$$\leq 0 + \gamma(P_0)_s^a(V_{AO}^\pi - V_{P_0}^\pi). \tag{12}$$

Consequently,

$$V_{AO}^\pi(s) - V_{P_0}^\pi(s) = \sum_{a \in \mathcal{A}} \pi(a|s)Q_{AO}^\pi(s,a) - \sum_{a \in \mathcal{A}} \pi(a|s)Q_{P_0}^\pi(s,a)$$

$$= \sum_{a \in \mathcal{A}} \pi(a|s)(Q_{AO}^\pi(s,a) - Q_{P_0}^\pi(s,a))$$

$$\leq \gamma(P_0^\pi)_s(V_{AO}^\pi - V_{P_0}^\pi), \tag{13}$$

where $(P_0^\pi)_s$ is the $s$-entry transition kernel induced by $\pi$ under $P_0$. Let $\mathbf{P}_0 = ((P_0^\pi)_{s_1}, ..., (P_0^\pi)_{s_{|\mathcal{S}|}})^\top$ be a transition matrix, and consider entry-wise relation, we have

$$V_{\text{AO}}^\pi - V_{P_0}^\pi \le \gamma \mathbf{P}_0 (V_{\text{AO}}^\pi - V_{P_0}^\pi), \tag{14}$$

and hence

$$(I - \gamma \mathbf{P}_0)(V_{\text{AO}}^\pi - V_{P_0}^\pi) \le 0. \tag{15}$$

Note that $(I - \gamma \mathbf{P}_0)^{-1} = I + \gamma \mathbf{P}_0 + \gamma^2 \mathbf{P}_0^2 + \dots$ exists and has all positive entries, we thus obtain

$$V_{\text{AO}}^\pi - V_{P_0}^\pi \le ((I - \gamma \mathbf{P}_0))^{-1} 0 = 0, \tag{16}$$

which completes the proof.

### D.3. Proof of Proposition 5.4

For two distribution $p, q \in \Delta(\mathcal{S})$, total variation is defined as $D(p, q) = \frac{1}{2}\|p - q\|_1$, and Wasserstein distance is defined as $D(p, q) = \inf_{\mu \in \Gamma(p,q)} (\mathbb{E}_{(X,Y) \sim \mu}[(X - Y)^t])^{\frac{1}{t}}$, where $\Gamma(p, q)$ is the set of all couplings of $p, q$.

We have

$$Q_{\bar{\mathcal{P}}}^\pi(s, a) - Q_{\text{AO}}^\pi(s, a) \tag{17}$$

$$= \gamma \sigma_{\bar{\mathcal{P}}_s^a}(V_{\bar{\mathcal{P}}}^\pi) - \gamma \frac{1}{K} \sum_{k=1}^K \sigma_{(\mathcal{P}_k)_s^a}(V_{\text{AO}}^\pi) \tag{18}$$

$$= \frac{\gamma}{K} \sum_{k=1}^K (\sigma_{\bar{\mathcal{P}}_s^a}(V_{\bar{\mathcal{P}}}^\pi) - \sigma_{(\mathcal{P}^k)_s^a}(V_{\text{AO}}^\pi)) \tag{19}$$

$$= \frac{\gamma}{K} \sum_{k=1}^K (\sigma_{\bar{\mathcal{P}}_s^a}(V_{\bar{\mathcal{P}}}^\pi) - \sigma_{(\mathcal{P}^k)_s^a}(V_{\bar{\mathcal{P}}}^\pi) + \sigma_{(\mathcal{P}^k)_s^a}(V_{\bar{\mathcal{P}}}^\pi) - \sigma_{(\mathcal{P}^k)_s^a}(V_{\text{AO}}^\pi)). \tag{20}$$

For any vector $v$ and total variation distance, it holds that

$$\sigma_{\bar{\mathcal{P}}_s^a}(v) - \frac{1}{K} \sum_{k=1}^K \sigma_{(\mathcal{P}_k)_s^a}(v) = \max_\alpha\{\bar{P} v_\alpha - \Gamma \mathbf{Sp}(v_\alpha)\} - \frac{1}{K} \sum_{k=1}^K \max_\alpha\{P_k V_\alpha - \Gamma \mathbf{Sp}(v_\alpha)\}$$

$$\le \max_\alpha\{\bar{P} v_\alpha - \Gamma \mathbf{Sp}(v_\alpha)\} - \frac{1}{K} \max_\alpha \left\{ \sum_{k=1}^K P^k v_\alpha - \Gamma \mathbf{Sp}(v_\alpha) \right\}$$

$$= 0; \tag{21}$$

Similarly, for Wasserstein distance, it holds that

$$\sigma_{\bar{\mathcal{P}}_s^a}(v) - \frac{1}{K} \sum_{k=1}^K \sigma_{(\mathcal{P}_k)_s^a}(v) = \sup_{\lambda \ge 0}\{-\lambda \Gamma^l + \sum_{s \in \mathcal{S}} \bar{P}(s) \inf_{y \in \mathcal{S}}(v(y) + \lambda d(s, y)^l)\}$$

$$- \frac{1}{K} \sum_k \sup_{\lambda \ge 0}\{-\lambda \Gamma^l + \sum_{s \in \mathcal{S}} P_k(s) \inf_{y \in \mathcal{S}}(v(y) + \lambda d(s, y)^l)\}$$

$$\le \sup_{\lambda \ge 0}\{-\lambda \Gamma^l + \sum_{s \in \mathcal{S}} \bar{P}(s) \inf_{y \in \mathcal{S}}(v(y) + \lambda d(s, y)^l)\}$$

$$- \sup_{\lambda \ge 0}\{-\lambda \Gamma^l + \sum_{s \in \mathcal{S}} \bar{P}(s) \inf_{y \in \mathcal{S}}(v(y) + \lambda d(s, y)^l)\}$$

$$= 0. \tag{22}$$

The remaining proof follows similarly as Theorem 5.2.

## D.4. Proof of Theorem 5.6

The contraction property of $\mathbf{T}_{\mathrm{AO}}$ follows similarly as Lemma 5.1.

For policy $\pi_*(s) \triangleq \arg\max_{a \in \mathcal{A}} Q_{\mathrm{AO}}(s, a)$, we have

$$\mathbf{T}_{\mathrm{AO}}^{\pi_*} Q_{\mathrm{AO}}(s, a) = r(s, a) + \frac{\gamma}{K} \sum_{k=1}^{K} \sigma_{(\mathcal{P}_k)_s^a} \max_{a \in \mathcal{A}} Q_{\mathrm{AO}}(s, a) = \mathbf{T}_{\mathrm{AO}} Q_{\mathrm{AO}}(s, a) = Q_{\mathrm{AO}}(s, a). \tag{23}$$

Hence $Q_{\mathrm{AO}}(s, a)$ is also the unique fixed point of $\mathbf{T}_{\mathrm{AO}}^{\pi_*}$, moreover, $V_{\mathrm{AO}}^{\pi_*} = V_{\mathrm{AO}}$.

For any $k \in \mathcal{K}$, denote the worst-case kernel of $V_{\mathrm{AO}}$ in $\mathcal{P}_k$ by $P'_k$, i.e.,

$$\sigma_{(\mathcal{P}_k)_s^a}(V_{\mathrm{AO}}) = (P'_k)_s^a V_{\mathrm{AO}}, \ \forall (s, a) \in \mathcal{S} \times \mathcal{A}. \tag{24}$$

Since $V_{\mathrm{AO}} = \max_{a \in \mathcal{A}} Q_{\mathrm{AO}}(s, a)$, it holds that

$$V_{\mathrm{AO}}^{\pi}(s) - V_{\mathrm{AO}}(s) \leq \sum_{a \in \mathcal{A}} \pi(a|s) \left( Q_{\mathrm{AO}}^{\pi}(s, a) - Q_{\mathrm{AO}}(s, a) \right)$$

$$= \sum_{a \in \mathcal{A}} \pi(a|s) \frac{\gamma}{K} \sum_{k=1}^{K} (\sigma_{(\mathcal{P}_k)_s^a}(V_{\mathrm{AO}}^{\pi}) - \sigma_{(\mathcal{P}_k)_s^a}(V_{\mathrm{AO}}))$$

$$\leq \sum_{a \in \mathcal{A}} \pi(a|s) \frac{\gamma}{K} \sum_{k=1}^{K} ((P'_k)_s^a V_{\mathrm{AO}}^{\pi} - (P'_k)_s^a V_{\mathrm{AO}}). \tag{25}$$

Let $\tilde{k} = \arg\max_{k \in \mathcal{K}} \sum_{a \in \mathcal{A}} \pi(a|s)\gamma(P'_k)_s^a(V_{\mathrm{MP}}^{\pi} - V_{\mathrm{MP}})$ and $\tilde{P}_s = \sum_{a \in \mathcal{A}} \pi(a|s)\gamma(P'_{\tilde{k}})_s^a$, we obtain

$$V_{\mathrm{AO}}^{\pi}(s) - V_{\mathrm{AO}}(s) \leq \sum_{a \in \mathcal{A}} \pi(a|s) \frac{\gamma}{K} \sum_{k=1}^{K} ((P'_k)_s^a V_{\mathrm{AO}}^{\pi} - (P'_k)_s^a V_{\mathrm{AO}})$$

$$\triangleq \gamma \tilde{P}_s(V_{\mathrm{AO}}^{\pi} - V_{\mathrm{AO}}), \tag{26}$$

and hence by noting that $(I - \gamma\tilde{P})^{-1}$ has all positive entries we have

$$(I - \gamma\tilde{P})(V_{\mathrm{AO}}^{\pi} - V_{\mathrm{AO}}) \leq 0, \tag{27}$$

which implies that $\max_{\pi} V_{\mathrm{AO}}^{\pi} \leq V_{\mathrm{AO}}$. On the other hand, since $V_{\mathrm{AO}}^{\pi_*} = V_{\mathrm{AO}}$ and $\max_{\pi} V_{\mathrm{AO}}^{\pi} \geq V_{\mathrm{AO}}^{\pi_*}$, together with the upper bound it follows that $\max_{\pi} V_{\mathrm{AO}}^{\pi} = V_{\mathrm{AO}}$. Note that $V_{\mathrm{AO}}(s) = \max_{a \in \mathcal{A}} Q_{\mathrm{AO}}(s, a)$, hence $\arg\max_{\pi} V_{\mathrm{AO}}^{\pi} = \pi_*$, which completes the proof.

## D.5. Proof of Theorem D.1

We first show the convergence of Algorithm 1 with the accurate operator to illustrate the efficiency of our algorithm design.

**Theorem D.1.** *Let* $E - 1 \leq \min \frac{1}{\lambda}\{\frac{\gamma}{1-\gamma}, \frac{1}{K}\}$, *and* $\lambda = \frac{4\log^2(TK)}{T(1-\gamma)}$. *If* $\hat{\mathbf{T}}_k = \mathbf{T}_k$, *it holds that*

$$\left\| Q_{\mathrm{AO}} - \frac{\sum_{k=1}^{K} Q_k}{K} \right\| \leq \tilde{\mathcal{O}}\left( \frac{1}{TK} + \frac{(E-1)\Gamma}{T} \right). \tag{28}$$

In this proof, we first assume that $E - 1 \leq \frac{1-\gamma}{4\gamma\lambda}$ and $\lambda \leq \frac{1}{E}$ to establish the general convergence rate, which is independent of $K$. Then we prove that carefully selected $E$ and $\lambda$ can balance each term in the convergence rate to achieve partial linear speedup.

Denote $\bar{Q}^{t+1}$ and $Q_k^{t+1}$ the values of $\bar{Q}$ and $Q_k$ at iteration $t + 1$. Since $\hat{\mathbf{T}}_k = \mathbf{T}_k$, we have

$$\bar{Q}^{t+1} = \frac{1}{K} \sum_{k=1}^{K} Q_k^{t+1}$$

$$= \frac{1}{K} \sum_{k=1}^{K} ((1 - \lambda)Q_k^t + \lambda(r + \gamma\sigma_{\mathcal{P}_k}(V_k^t))), \tag{29}$$

where $V_k^t(s) = \max_{a \in \mathcal{A}} Q_k^t(s, a)$. We also denote the value function associate with $Q_{\text{AO}}$ by $V_{\text{AO}} = \max_{a \in \mathcal{A}} Q_{\text{AO}}(s, a)$. We define the iteration error as $\Delta^{t+1} := Q_{\text{AO}} - \bar{Q}^{t+1}$, additionally, $\Delta^0 := Q_{\text{AO}} - Q^0$. The iteration error then takes the following form:

$$
\begin{aligned}
\Delta^{t+1} &= Q_{\text{AO}} - \bar{Q}^{t+1} \\
&= \frac{1}{K} \sum_{k=1}^{K} (Q_{\text{AO}} - ((1-\lambda)Q_k^t + \lambda(r + \gamma \sigma_{\mathcal{P}_k}(V_k^t)))) \\
&= \frac{1}{K} \sum_{k=1}^{K} ((1-\lambda)(Q_{\text{AO}} - Q_k^t) + \lambda(Q_{\text{AO}} - r - \gamma \sigma_{\mathcal{P}_k}(V_k^t)) \\
&= (1-\lambda)\Delta^t + \frac{\gamma\lambda}{K} \sum_{k=1}^{K} (\sigma_{\mathcal{P}_k}(V_{\text{AO}}) - \sigma_{\mathcal{P}_k}(V_k^t)) \\
&= (1-\lambda)^{t+1}\Delta^0 + \gamma\lambda \sum_{i=0}^{t} (1-\lambda)^{t-i} \frac{1}{K} \sum_{k=1}^{K} (\sigma_{\mathcal{P}_k}(V_{\text{AO}}) - \sigma_{\mathcal{P}_k}(V_k^i)).
\end{aligned}
\tag{30}
$$

By taking the $l_\infty$-norm on both sides, we obtain

$$
\left\| \Delta^{t+1} \right\| \leq (1-\lambda)^{t+1} \left\| \Delta^0 \right\| + \left\| \gamma\lambda \sum_{i=0}^{t} (1-\lambda)^{t-i} \frac{1}{K} \sum_{k=1}^{K} (\sigma_{\mathcal{P}_k}(V_{\text{AO}}) - \sigma_{\mathcal{P}_k}(V_k^i)) \right\|.
\tag{31}
$$

Since $0 \leq Q^0 \leq \frac{1}{1-\gamma}$, the first term in (30) can be bounded as

$$
(1-\lambda)^{t+1} \left\| \Delta^0 \right\| \leq (1-\lambda)^{t+1} \frac{1}{1-\gamma}.
\tag{32}
$$

To bound the second term in (30), we divide the summation into two parts. Define the most recent aggregation step of $t$ as $\chi(t) := t - (t \mod E)$. Then for any $\beta E \leq t \leq T$, we have

$$
\begin{aligned}
&\sum_{i=0}^{t} (1-\lambda)^{t-i} \lambda\gamma \left\| \frac{1}{K} \sum_{k=1}^{K} (\sigma_{\mathcal{P}_k}(V_{\text{AO}}) - \sigma_{\mathcal{P}_k}(V_k^i)) \right\| \\
&= \sum_{i=0}^{\chi(t)-\beta E} (1-\lambda)^{t-i} \lambda\gamma \left\| \frac{1}{K} \sum_{k=1}^{K} (\sigma_{\mathcal{P}_k}(V_{\text{AO}}) - \sigma_{\mathcal{P}_k}(V_k^i)) \right\| + \sum_{i=\chi(t)-\beta E+1}^{t} (1-\lambda)^{t-i} \lambda\gamma \left\| \frac{1}{K} \sum_{k=1}^{K} (\sigma_{\mathcal{P}_k}(V_{\text{AO}}) - \sigma_{\mathcal{P}_k}(V_k^i)) \right\| \\
&\leq \frac{\gamma}{1-\gamma} (1-\lambda)^{t-\chi(t)+\beta E} + \sum_{i=\chi(t)-\beta E+1}^{t} (1-\lambda)^{t-i} \lambda\gamma \left\| \frac{1}{K} \sum_{k=1}^{K} (\sigma_{\mathcal{P}_k}(V_{\text{AO}}) - \sigma_{\mathcal{P}_k}(V_k^i)) \right\|.
\end{aligned}
\tag{33}
$$

Before proceeding with the proof, we introduce the following lemma to bound the second term in (33). The proof of this lemma can be found in Appendix D.6. We define $\Delta_k^t := Q_{\text{AO}} - Q_k^t$ the local iteration error of agent $k$. We also define $\bar{\sigma}_{(\mathcal{P}_k)_s^a}(v) := \frac{1}{K} \sum_k^K \sigma_{(\mathcal{P}_k)_s^a}(v)$ for any vector $v$.

**Lemma D.2.** *If $t \mod E = 0$, then $\left\| \frac{1}{K} \sum_{k=1}^{K} (\sigma_{\mathcal{P}_k}(V_{\text{AO}}) - \sigma_{\mathcal{P}_k}(V_k^t)) \right\| \leq \|\Delta^t\|$. Otherwise,*

$$
\begin{aligned}
\left\| \frac{1}{K} \sum_{k=1}^{K} (\sigma_{\mathcal{P}_k}(V_{\text{AO}}) - \sigma_{\mathcal{P}_k}(V_k^t)) \right\| &\leq \left\| \Delta^{\chi(t)} \right\| + 2\lambda \frac{1}{K} \sum_{k=1}^{K} \sum_{t'=\chi(t)}^{t-1} \left\| \Delta_k^{t'} \right\| \\
&+ \gamma\lambda \frac{t-1-\chi(t)}{K} \sum_{k=1}^{K} \max_{(s,a) \in \mathcal{S} \times \mathcal{A}} \left| \sigma_{(\mathcal{P}_k)_s^a}(V_{\text{AO}}) - \bar{\sigma}_{(\mathcal{P}_k)_s^a}(V_{\text{AO}}) \right|,
\end{aligned}
\tag{34}
$$

*where we use the convention that $\sum_{t'=\chi(t)}^{\chi(t)-1} \left\| \Delta_{t'}^k \right\| = 0$.*

Then by Lemma D.2, (33) yields

$$
\sum_{i=\chi(t)-\beta E+1}^{t} (1-\lambda)^{t-i}\lambda\gamma \left\| \frac{1}{K}\sum_{k=1}^{K}(\sigma_{\mathcal{P}_k}(V_{\mathrm{AO}}) - \sigma_{\mathcal{P}_k}(V_k^i)) \right\|
$$
$$
\leq \sum_{i=\chi(i)-\beta E+1}^{t} (1-\lambda)^{t-i}\lambda\gamma \left( \left\| \Delta^{\chi(i)} \right\| + 2\lambda\frac{1}{K}\sum_{k=1}^{K}\sum_{j=\chi(t)}^{i-1} \left\| \Delta_k^j \right\| \right.
$$
$$
\left. + \gamma\lambda\frac{t-1-\chi(i)}{K}\sum_{k=1}^{K}\max_{(s,a)\in\mathcal{S}\times\mathcal{A}} \left| \sigma_{(\mathcal{P}_k)_s^a}(V_{\mathrm{AO}}) - \bar{\sigma}_{(\mathcal{P}_k)_s^a}(V_{\mathrm{AO}}) \right| \right)
$$
$$
\leq \sum_{i=\chi(i)-\beta E+1}^{t} (1-\lambda)^{t-i}\lambda\gamma \left( \left\| \Delta^{\chi(i)} \right\| + 2\lambda\frac{1}{K}\sum_{k=1}^{K}\sum_{j=\chi(t)}^{i-1} \left\| \Delta_k^j \right\| + \gamma\lambda\Gamma\frac{E-1}{1-\gamma} \right). \tag{35}
$$

Furthermore, the local iteration error $\left\| \Delta_k^j \right\|$ can be bounded using the following lemma. The proof of this lemma can be found in Appendix D.7.

**Lemma D.3.** *If* $\lambda \leq \frac{1}{E}$, *then for any* $t \in [0, T-1]$ *and* $k \in \mathcal{K}$, $\|\Delta_k^t\| \leq \left\| \Delta^{\chi(t)} \right\| + \frac{3\gamma}{1-\gamma}\lambda(E-1)\Gamma$.

Then by Lemma D.3, we have

$$
\sum_{i=\chi(i)-\beta E+1}^{t} (1-\lambda)^{t-i}\lambda\gamma 2\lambda\frac{1}{K}\sum_{k=1}^{K}\sum_{j=\chi(t)}^{i-1} \left\| \Delta_k^j \right\|
$$
$$
\leq 2\lambda^2\gamma \sum_{i=\chi(t)-\beta E+1}^{t} (1-\lambda)^{t-i}\sum_{j=\chi(i)}^{i-1} \left( \left\| \Delta^{\chi(i)} \right\| + 3\frac{\gamma}{1-\gamma}\lambda(E-1)\Gamma \right)
$$
$$
\leq 2\lambda\gamma(E-1)\max_{\chi(t)-\beta E\leq i\leq t} \left\| \Delta^{\chi(i)} \right\| + \frac{6\gamma^2\lambda^2}{1-\gamma}(E-1)^2\Gamma. \tag{36}
$$

By combining both bounds (35) and (36), we obtain

$$
\sum_{i=\chi(t)-\beta E+1}^{t} (1-\lambda)^{t-i}\lambda\gamma \left\| \frac{1}{K}\sum_{k=1}^{K}(\sigma_{\mathcal{P}_k}(V_{\mathrm{AO}}) - \sigma_{\mathcal{P}_k}(V_k^i)) \right\|
$$
$$
\leq \gamma\max_{\chi(t)-\beta E\leq i\leq t} \left\| \Delta^{\chi(i)} \right\| + 2\lambda\gamma(E-1)\max_{\chi(t)-\beta E\leq i\leq t} \left\| \Delta^{\chi(i)} \right\| + \frac{6\gamma^2\lambda^2}{1-\gamma}(E-1)^2\Gamma
$$
$$
+ \sum_{i=\chi(t)-\beta E+1}^{t} (1-\lambda)^{t-i}\lambda\gamma(\frac{\gamma\lambda}{1-\gamma}(E-1)\Gamma)
$$
$$
= \gamma(1+2\lambda(E-1))\max_{\chi(t)-\beta E\leq i\leq t} \left\| \Delta^{\chi(i)} \right\| + \frac{\gamma^2}{1-\gamma}(6\lambda^2(E-1)^2 + \lambda(E-1))\Gamma. \tag{37}
$$

Plugging the bound (37) into (33) then yields

$$
\sum_{i=0}^{t} (1-\lambda)^{t-i}\lambda\gamma \left\| \frac{1}{K}\sum_{k=1}^{K}(\sigma_{\mathcal{P}_k}(V_{\mathrm{AO}}) - \sigma_{\mathcal{P}_k}(V_k^i)) \right\|
$$
$$
\leq \frac{\gamma}{1-\gamma}(1-\lambda)^{t-\chi(t)+\beta E} + \gamma(1+2\lambda(E-1))\max_{\chi(t)-\beta E\leq i\leq t} \left\| \Delta^{\chi(i)} \right\|
$$
$$
+ \frac{\gamma^2}{1-\gamma}(6\lambda^2(E-1)^2 + \lambda(E-1))\Gamma. \tag{38}
$$

Consequently, the iteration error $\left\|\Delta^{t+1}\right\|$ can be bounded using (32) and (38) for all rounds $t \in [0, T-1]$:

$$
\begin{aligned}
\left\|\Delta^{t+1}\right\| &\le (1-\lambda)^{t+1}\frac{1}{1-\gamma} + \frac{\gamma}{1-\gamma}(1-\lambda)^{t-\chi(t)+\beta E} \\
&\quad + \gamma(1+2\lambda(E-1))\max_{\chi(t)-\beta E \le i \le t}\left\|\Delta^{\chi(i)}\right\| + \frac{\gamma^2}{1-\gamma}(6\lambda^2(E-1)^2 + \lambda(E-1))\Gamma \\
&\le \gamma(1+2\lambda(E-1))\max_{\chi(t)-\beta E \le i \le t}\left\|\Delta^{\chi(i)}\right\| + \frac{2}{1-\gamma}(1-\lambda)^{\beta E} \\
&\quad + \frac{\gamma^2}{1-\gamma}(6\lambda^2(E-1)^2 + \lambda(E-1))\Gamma.
\end{aligned}
\tag{39}
$$

Let $\rho := \frac{2}{1-\gamma}(1-\lambda)^{\beta E} + \frac{\gamma^2}{1-\gamma}(6\lambda^2(E-1)^2 + \lambda(E-1))\Gamma$, then with the assumption $\lambda \le \frac{1-\gamma}{4\gamma(E-1)}$, (39) can be written as

$$
\left\|\Delta^{t+1}\right\| \le \frac{1+\gamma}{2}\max_{\chi(t)-\beta E \le i \le t}\left\|\Delta^{\chi(i)}\right\| + \rho.
\tag{40}
$$

Unrolling the above recursion $L$ times with $L\beta E \le t \le T-1$, we obtain

$$
\begin{aligned}
\left\|\Delta^{t+1}\right\| &\le \left(\frac{1+\gamma}{2}\right)^L \max_{\chi(t)-L\beta E \le i \le t}\left\|\Delta^{\chi(i)}\right\| + \sum_{i=0}^{L-1}\left(\frac{1+\gamma}{2}\right)^i \rho \\
&\le \left(\frac{1+\gamma}{2}\right)^L \frac{1}{1-\gamma} + \frac{2}{1-\gamma}\rho.
\end{aligned}
\tag{41}
$$

By choosing $\beta = \lfloor \frac{1}{E}\sqrt{\frac{(1-\gamma)T}{2\lambda}} \rfloor$, $L = \lceil \sqrt{\frac{\lambda T}{1-\gamma}} \rceil$ and $t+1 = T$, (41) yields

$$
\begin{aligned}
\left\|\Delta^T\right\| &\le \frac{1}{1-\gamma}\left(\frac{1+\gamma}{2}\right)^{\sqrt{\frac{\lambda T}{1-\gamma}}} + \frac{2}{1-\gamma}\left(\frac{2}{1-\gamma}(1-\lambda)^{\beta E} + \frac{\gamma^2}{1-\gamma}(6\lambda^2(E-1)^2 + \lambda(E-1))\Gamma\right) \\
&\le \frac{1}{1-\gamma}\exp\left\{-\frac{1}{2}\sqrt{(1-\gamma)\lambda T}\right\} + \frac{4}{(1-\gamma)^2}\exp\left\{-\sqrt{(1-\gamma)\lambda T}\right\} \\
&\quad + \frac{2\gamma^2}{(1-\gamma)^2}(6\lambda^2(E-1)^2 + \lambda(E-1))\Gamma \\
&\le \frac{4}{(1-\gamma)^2}\exp\left\{-\frac{1}{2}\sqrt{(1-\gamma)\lambda T}\right\} + \frac{2\gamma^2}{(1-\gamma)^2}(6\lambda^2(E-1)^2 + \lambda(E-1))\Gamma.
\end{aligned}
\tag{42}
$$

(42) reveals the effect of $E$ and $\lambda$ on the convergence rate. Although $E > 1$ slows down the convergence, we can set $(E-1) \le \min \frac{1}{\lambda}\{\frac{\gamma}{1-\gamma}, \frac{1}{K}\}$, and $\lambda = \frac{4\log^2(TK)}{T(1-\gamma)}$ to bring partial linear speedup with respect to the number of agents $K$. Then (39) becomes

$$
\left\|\Delta^T\right\| \le \tilde{\mathcal{O}}\left(\frac{1}{TK} + \frac{(E-1)\Gamma}{T}\right)
\tag{43}
$$

for $T \ge E$, which completes the proof.

### D.6. Proof of Lemma D.2

When $t \mod E = 0$, i.e., $t$ is an aggregation step, $Q_k^t = Q_{k'}^t$ and the associate value functions $V_k^t = V_{k'}^t$ for any $k$ and $k' \in \mathcal{K}$. We denote $\bar{V}^t = \frac{1}{K}\sum_{k=1}^K V_k^t$, then we have

$$
\begin{aligned}
\frac{1}{K}\sum_{k=1}^K(\sigma_{(\mathcal{P}_k)_s^a}(V_{\text{AO}}) - \sigma_{(\mathcal{P}_k)_s^a}(V_k^t)) &\le \frac{1}{K}\sum_{k=1}^K \bar{\sigma}_{(\mathcal{P}_k)_s^a}(V_{\text{AO}} - \bar{V}^t) \\
&\le \frac{1}{K}\sum_{k=1}^K\left\|V_{\text{AO}} - \bar{V}^t\right\| \\
&\le \left\|Q_{\text{AO}} - \bar{Q}^t\right\| \\
&= \left\|\Delta^t\right\|.
\end{aligned}
\tag{44}
$$

For general $t$, we have

$$\left\| \frac{1}{K} \sum_{k=1}^{K} (\sigma_{\mathcal{P}_k}(V_{\text{AO}}) - \sigma_{\mathcal{P}_k}(V_k^t)) \right\| = \left\| \frac{1}{K} \sum_{k=1}^{K} (\sigma_{\mathcal{P}_k}(V_{\text{AO}}) - \sigma_{\mathcal{P}_k}(V_k^{\chi(t)}) + \sigma_{\mathcal{P}_k}(V_k^{\chi(t)}) - \sigma_{\mathcal{P}_k}(V_k^t)) \right\|$$

$$\leq \left\| \frac{1}{K} \sum_{k=1}^{K} (\sigma_{\mathcal{P}_k}(V_{\text{AO}}) - \sigma_{\mathcal{P}_k}(V_k^{\chi(t)})) \right\| + \left\| \frac{1}{K} \sum_{k=1}^{K} (\sigma_{\mathcal{P}_k}(V_k^{\chi(t)}) - \sigma_{\mathcal{P}_k}(V_k^t)) \right\|$$

$$\leq \left\| \Delta^{\chi(t)} \right\| + \left\| \frac{1}{K} \sum_{k=1}^{K} (\sigma_{\mathcal{P}_k}(V_k^{\chi(t)}) - \sigma_{\mathcal{P}_k}(V_k^t)) \right\|$$

$$\leq \left\| \Delta^{\chi(t)} \right\| + \frac{1}{K} \sum_{k=1}^{K} \left\| V_k^{\chi(t)} - V_k^t \right\|. \tag{45}$$

For any state $s$, we have

$$V_k^t(s) - V_k^{\chi(t)}(s) = Q_k^t(s, a_k^t(s)) - Q_k^{\chi(t)}(s, a_k^{\chi(t)}(s))$$

$$\overset{(a)}{\leq} Q_k^t(s, a_k^t(s)) - Q_k^{\chi(t)}(s, a_k^t(s))$$

$$= Q_k^t(s, a_k^t(s)) - Q_k^{t-1}(s, a_k^t(s)) + Q_k^{t-1}(s, a_k^t(s)) - Q_k^{t-2}(s, a_k^t(s))$$

$$+ \cdots + Q_k^{\chi(t)+1}(s, a_k^t(s)) - Q_k^{\chi(t)}(s, a_k^t(s)). \tag{46}$$

where $a_k^t(s) = \arg\max_{a \in \mathcal{A}} Q_k^t(s, a)$ and inequality (a) holds because $Q_k^{\chi(t)}(s, a_k^t(s)) \leq Q_k^{\chi(t)}(s, a_k^{\chi(t)}(s))$.

For each $t'$ such that $\chi(t) \leq t' \leq t$, it holds that,

$$Q_k^{t'+1}(s, a_k^t(s)) - Q_k^{t'}(s, a_k^t(s)) = (1-\lambda)Q_k^{t'}(s, a_k^t(s)) + \lambda(r(s, a_k^t(s)) + \gamma\sigma_{(\mathcal{P}_k)_s^{a_k^t(s)}}(V_k^{t'})) - Q_k^{t'}(s, a_k^t(s))$$

$$\overset{(a)}{=} -\lambda Q_k^{t'}(s, a_k^t(s)) + \lambda(Q_{\text{AO}}(s, a_k^t(s)) - r(s, a_k^t(s)) - \gamma\bar{\sigma}_{(\mathcal{P}_k)_s^{a_k^t(s)}}(V_{\text{AO}})$$

$$+ r(s, a_k^t(s)) + \gamma\sigma_{(\mathcal{P}_k)_s^{a_k^t(s)}}(V_k^{t'}))$$

$$\leq 2\lambda \left\| \Delta_k^{t'} \right\| + \gamma\lambda(\sigma_{(\mathcal{P}_k)_s^{a_k^t(s)}}(V_{\text{AO}}) - \bar{\sigma}_{(\mathcal{P}_k)_s^{a_k^t(s)}}(V_{\text{AO}})), \tag{47}$$

where equality (a) follows from the average optimal Bellman equation. Thus,

$$V_k^t(s) - V_k^{\chi(t)}(s) \leq \sum_{t'=\chi(t)}^{t-1} (Q_k^{t'+1}(s, a_k^t(s)) - Q_k^{t'}(s, a_k^t(s)))$$

$$= 2\lambda \sum_{t'=\chi(t)}^{t-1} \left\| \Delta_k^{t'} \right\| + \gamma\lambda(t-1-\chi(t))(\sigma_{(\mathcal{P}_k)_s^{a_k^t(s)}}(V_{\text{AO}}) - \bar{\sigma}_{(\mathcal{P}_k)_s^{a_k^t(s)}}(V_{\text{AO}})). \tag{48}$$

Similarly, we have

$$V_k^t(s) - V_k^{\chi(t)}(s) \geq \sum_{t'=\chi(t)}^{t-1} (Q_k^{t'+1}(s, a_k^{\chi(t)}(s)) - Q_k^{t'}(s, a_k^{\chi(t)}(s)))$$

$$\geq -2\lambda \sum_{t'=\chi(t)}^{t-1} \left\| \Delta_k^{t'} \right\| + \gamma\lambda(t-1-\chi(t))(\sigma_{(\mathcal{P}_k)_s^{a_k^t(s)}}(V_{\text{AO}}) - \bar{\sigma}_{(\mathcal{P}_k)_s^{a_k^t(s)}}(V_{\text{AO}})). \tag{49}$$

By plugging the bounds in (48) and in (49) back into (45), we obtain

$$
\begin{aligned}
\left\| \frac{1}{K} \sum_{k=1}^{K} (\sigma_{\mathcal{P}_k}(V_{\mathrm{AO}}) - \sigma_{\mathcal{P}_k}(V_k^t)) \right\| &\leq \left\| \Delta_{\chi(t)} \right\| + \frac{1}{K} \sum_{k=1}^{K} \left\| V_k^{\chi(t)} - V_k^t \right\| \\
&\leq \left\| \Delta_{\chi(t)} \right\| + 2\lambda \frac{1}{K} \sum_{k=1}^{K} \sum_{t'=\chi(t)}^{t-1} \left\| \Delta_k^{t'} \right\| \\
&\quad + \gamma\lambda(t - 1 - \chi(t)) \frac{1}{K} \sum_{k=1}^{K} \max_{s,a \in \mathcal{S} \times \mathcal{A}} |\sigma_{(\mathcal{P}_k)_s^a}(V_{\mathrm{AO}}) - \bar{\sigma}_{(\mathcal{P}_k)_s^a}(V_{\mathrm{AO}})|.
\end{aligned}
\tag{50}
$$

This hence completes the proof.

### D.7. Proof of Lemma D.3

When $t \mod E = 0$, then $\Delta_k^t = \Delta^{\chi(t)}$. When $t \mod E \neq 0$, we have

$$
\begin{aligned}
Q_k^t &= (1-\lambda)Q_k^{t-1} + \lambda(r + \gamma\sigma_{\mathcal{P}_k}(V_k^{t-1})) \\
&= (1-\lambda)Q_k^{t-1} + \lambda(Q_{\mathrm{AO}} - r - \gamma\bar{\sigma}_{\mathcal{P}_k}(V_{\mathrm{AO}}) + r + \gamma\sigma_{\mathcal{P}_k}(V_k^{t-1})).
\end{aligned}
\tag{51}
$$

Thus

$$
\begin{aligned}
\Delta_k^t &= (1-\lambda)\Delta_k^{t-1} + \lambda\gamma(\bar{\sigma}_{\mathcal{P}_k}(V_{\mathrm{AO}}) - \sigma_{\mathcal{P}_k}(V_k^{t-1})) \\
&= (1-\lambda)\Delta_k^{t-1} + \lambda\gamma(\bar{\sigma}_{\mathcal{P}_k}(V_{\mathrm{AO}}) - \sigma_{\mathcal{P}_k}(V_{\mathrm{AO}}) + \sigma_{\mathcal{P}_k}(V_{\mathrm{AO}}) - \sigma_{\mathcal{P}_k}(V_k^{t-1})) \\
&= (1-\lambda)^{t-\chi(t)}\Delta^{\chi(i)} + \gamma\lambda \sum_{t'=\chi(t)}^{t-1} (1-\lambda)^{t-t'-1}(\bar{\sigma}_{\mathcal{P}_k}(V_{\mathrm{AO}}) - \sigma_{\mathcal{P}_k}(V_{\mathrm{AO}})) \\
&\quad + \gamma\lambda \sum_{t'=\chi(t)}^{t-1} (1-\lambda)^{t-t'-1}(\sigma_{\mathcal{P}_k}(V_{\mathrm{AO}}) - \sigma_{\mathcal{P}_k}(V_j^k)).
\end{aligned}
\tag{52}
$$

For any state-action pair $(s, a) \in \mathcal{S} \times \mathcal{A}$,

$$
\left| (1-\lambda)^{t-\chi(t)}\Delta^{\chi(t)}(s, a) \right| \leq (1-\lambda)^{t-\chi(t)} \left\| \Delta^{\chi(t)} \right\|.
\tag{53}
$$

Note that

$$
\begin{aligned}
\gamma\lambda \sum_{t'=\chi(t)}^{t-1} (1-\lambda)^{t-t'-1}(\bar{\sigma}_{\mathcal{P}_k}(V_{\mathrm{AO}}) - \sigma_{\mathcal{P}_k}(V_{\mathrm{AO}})) &\leq \frac{\gamma}{1-\gamma}\lambda \sum_{t'=\chi(t)}^{t-1} (1-\lambda)^{t-t'-1}\Gamma \\
&\leq \frac{\gamma}{1-\gamma}\lambda(E-1)\Gamma,
\end{aligned}
\tag{54}
$$

for all $(s, a) \in \mathcal{S} \times \mathcal{A}$, $t \in [0, T-1]$ and $k \in \mathcal{K}$. In addition, we have

$$
\left\| \gamma\lambda \sum_{t'=\chi(t)}^{t-1} (1-\lambda)^{t-t'-1}(\sigma_{\mathcal{P}_k}(V_{\mathrm{AO}}) - \sigma_{\mathcal{P}_k}(V_k^{t'})) \right\| \leq \gamma\lambda \sum_{t'=\chi(t)}^{t-1} (1-\lambda)^{t-t'-1} \left\| \Delta_k^{t'} \right\|.
\tag{55}
$$

By combining the bounds in (53), (54), and (55), we obtain

$$
\begin{aligned}
\left\| \Delta_k^t \right\| &\leq (1-\lambda)^{t-\chi(t)} \left\| \Delta^{\chi(t)} \right\| + \frac{\gamma}{1-\gamma}\lambda(E-1)\Gamma + \gamma\lambda \sum_{t'=\chi(t)}^{t-1} (1-\lambda)^{t-t'-1} \left\| \Delta_k^{t'} \right\| \\
&\leq (1 - (1-\gamma)\lambda)^{t-\chi(t)} \left\| \Delta^{\chi(t)} \right\| + (1 + \gamma\lambda)^{t-\chi(t)}\left(\frac{\gamma}{1-\gamma}\lambda(E-1)\Gamma\right),
\end{aligned}
\tag{56}
$$

where the last inequality can be shown via inducting on $t - \chi(t) \in \{0, \cdots, E - 1\}$. When $\lambda \leq \frac{1}{E}$,

$$(1 + \gamma\lambda)^{t-\chi(t)} \leq (1 + \lambda)^E \leq (1 + 1/E)^E \leq e \leq 3. \tag{57}$$

Hence

$$\left\| \Delta_k^t \right\| \leq \left\| \Delta^{\chi(t)} \right\| + 3\frac{\gamma}{1-\gamma}\lambda(E-1)\Gamma, \tag{58}$$

which completes the proof.

### D.8. Proof of Theorem 5.7

Since $\mathbb{E}[\hat{\mathbf{T}}_k] = \mathbf{T}_k$ and $\hat{\mathbf{T}}_{\mathrm{AO}} = \frac{\sum_{k=1}^K \hat{\mathbf{T}}_k}{K}$, it follows that $\mathbb{E}[\hat{\mathbf{T}}_{\mathrm{AO}}] = \frac{\sum_{k=1}^K \mathbb{E}[\hat{\mathbf{T}}_k]}{K} = \mathbf{T}_{\mathrm{AO}}$. Then by replacing $\mathbf{T}_k$ and $\mathbf{T}_{\mathrm{AO}}$ with their expectation versions, the proof follows naturally as that in Theorem D.1, and is thus omitted.

## E. Theoretical Proofs for Minimal Pessimism Principle

In this section, we provide the construction and proofs of the theoretical results for the minimal pessimism principle.

### E.1. Multi-level Monte Carlo Operator Construction

We first generate $N$ according to a geometric distribution with parameter $\Psi \in (0, 1)$. Then, we generate $2^{N+1}$ unbiased estimator $\hat{\mathbf{T}}_k^i, i = 1, ..., 2^{N+1}$. We divide these estimators into two groups: estimators with odd indices, and estimators with even indices. We then individually calculate the maximal aggregation using the even-index estimators, odd-index ones, all of the estimators, and the first one:

$$\hat{\mathbf{T}}^E Q = \max_{k,i}\{\hat{\mathbf{T}}_k^i Q, i = 2, 4, ..., 2^{N+1}\}, \tag{59}$$

$$\hat{\mathbf{T}}^O Q = \max_{k,i}\{\hat{\mathbf{T}}_k^i Q, i = 1, 3, ..., 2^{N+1} - 1\}, \tag{60}$$

$$\hat{\mathbf{T}}^A Q = \max_{k,i}\{\hat{\mathbf{T}}_k^i Q, i = 1, 2, ..., 2^{N+1}\}, \tag{61}$$

$$\hat{\mathbf{T}}^1 Q = \max_k\{\hat{\mathbf{T}}_k^1\}. \tag{62}$$

The multi-level estimator is then constructed as

$$\hat{\mathbf{T}}_{\mathrm{MLMC}} Q \triangleq \hat{\mathbf{T}}^1 Q + \frac{\Delta(Q)}{p_N}, \tag{63}$$

where $p_N = \Psi(1 - \Psi)^N$ and

$$\Delta(Q) \triangleq \hat{\mathbf{T}}^A Q - \frac{\hat{\mathbf{T}}^O Q + \hat{\mathbf{T}}^E Q}{2}. \tag{64}$$

### E.2. Proof of Theorem 6.1

We first introduce some useful inequalities to extend the results in averaged operator based transfer to the max-aggregation version.

**Lemma E.1.** *For any vector sequences $\{X_k\}$ and $\{Y_k\}$ with $k \in \mathcal{K}$, it holds that*

1) $\max_{k\in\mathcal{K}} X_k - \max_{k\in\mathcal{K}} Y_k \leq \max_{k\in\mathcal{K}}(X_k - Y_k)$,

2) $\| \max_{k\in\mathcal{K}} X_k - \max_{k\in\mathcal{K}} Y_k \| \leq \max_{k\in\mathcal{K}} \|X_k - Y_k\|$,

3) $\max_{k\in\mathcal{K}} \|X_k - \max_{k\in\mathcal{K}} X_k\| \leq 2\max_{k\in\mathcal{K}} \|X_k\|$.

The proof of this lemma can be found in Appendix E.3.

Since $\mathbf{T}_k^\pi$ is a $\gamma$-contraction for any $k \in \mathcal{K}$, then for some $Q_1$ and $Q_2$, by Lemma E.1 we have

$$
\begin{aligned}
\|\mathbf{T}_{\mathrm{MP}}^\pi Q_1 - \mathbf{T}_{\mathrm{MP}}^\pi Q_2\| &= \|\max_{k \in \mathcal{K}} \mathbf{T}_k^\pi Q_1 - \max_{k \in \mathcal{K}} \mathbf{T}_k^\pi Q_2\| \\
&\leq \max_{k \in \mathcal{K}} \|\mathbf{T}_k^\pi Q_1 - \mathbf{T}_k^\pi Q_2\| \\
&\leq \gamma \|Q_1 - Q_2\|.
\end{aligned}
\tag{65}
$$

Hence $\mathbf{T}_{\mathrm{MP}}^\pi$ is also a $\gamma$-contraction and has a unique fixed point $Q_{\mathrm{MP}}^\pi$. The claim for $\mathbf{T}_{\mathrm{MP}}$ can be similarly derived.

In order to verify that $V_{\mathrm{MP}}^\pi$ is a lower bound on $V_{P_0}^\pi$ we proceed as follows. Let $V_{\mathrm{MP}}^\pi = \sum_{a \in \mathcal{A}} \pi(a|s) Q_{\mathrm{MP}}^\pi(s,a)$, since $P_0 \in \mathcal{P}_k$, we have

$$
\begin{aligned}
Q_{\mathrm{MP}}^\pi(s,a) - Q_{P_0}^\pi(s,a) &= \gamma \left( \max_{k \in \mathcal{K}} \sigma_{(\mathcal{P}_k)_s^a}(V_{\mathrm{MP}}^\pi) - (P_0)_s^a V_{\mathrm{MP}}^\pi \right) + \gamma (P_0)_s^a (V_{\mathrm{MP}}^\pi - V_{P_0}^\pi) \\
&\leq 0 + \gamma (P_0)_s^a (V_{\mathrm{MP}}^\pi - V_{P_0}^\pi).
\end{aligned}
\tag{66}
$$

Consequently,

$$
\begin{aligned}
V_{\mathrm{MP}}^\pi(s) - V_{P_0}^\pi(s) &= \sum_{a \in \mathcal{A}} \pi(a|s) Q_{\mathrm{MP}}^\pi(s,a) - \sum_{a \in \mathcal{A}} \pi(a|s) Q_{P_0}^\pi(s,a) \\
&= \sum_{a \in \mathcal{A}} \pi(a|s) (Q_{\mathrm{MP}}^\pi(s,a) - Q_{P_0}^\pi(s,a)) \\
&\leq \gamma (P_0^\pi)_s (V_{\mathrm{MP}}^\pi - V_{P_0}^\pi),
\end{aligned}
\tag{67}
$$

where $(P_0^\pi)_s$ is the $s$-entry transition kernel induced by $\pi$ under $P_0$. Let $\mathbf{P}_0 = ((P_0^\pi)_{s_1}, ..., (P_0^\pi)_{s_{|\mathcal{S}|}})^\top$ be a transition matrix, and consider entry-wise relation, we have

$$
V_{\mathrm{MP}}^\pi - V_{P_0}^\pi \leq \gamma \mathbf{P}_0 (V_{\mathrm{MP}}^\pi - V_{P_0}^\pi),
\tag{68}
$$

and hence

$$
(I - \gamma \mathbf{P}_0)(V_{\mathrm{MP}}^\pi - V_{P_0}^\pi) \leq 0.
\tag{69}
$$

Note that $(I - \gamma \mathbf{P}_0)^{-1} = I + \gamma \mathbf{P}_0 + \gamma^2 \mathbf{P}_0^2 + \ldots$ exists and has all positive entries, we thus obtain

$$
V_{\mathrm{MP}}^\pi - V_{P_0}^\pi \leq ((I - \gamma \mathbf{P}_0))^{-1} 0 = 0.
\tag{70}
$$

The claim $V_{\mathcal{P}_k}^\pi \leq V_{\mathrm{MP}}^\pi$ can be derived similarly.

For policy $\pi_*(s) \triangleq \arg\max_{a \in \mathcal{A}} Q_{\mathrm{MP}}(s,a)$, we have

$$
\mathbf{T}_{\mathrm{MP}}^{\pi_*} Q_{\mathrm{MP}}(s,a) = r(s,a) + \gamma \max_{k \in \mathcal{K}}(\sigma_{(\mathcal{P}_k)_s^a} \max_{a \in \mathcal{A}} Q_{\mathrm{MP}}(s,a)) = \mathbf{T}_{\mathrm{MP}} Q_{\mathrm{MP}}(s,a) = Q_{\mathrm{MP}}(s,a).
\tag{71}
$$

Hence $Q_{\mathrm{MP}}(s,a)$ is also the unique fixed point of $\mathbf{T}_{\mathrm{MP}}^{\pi_*}$, moreover, $V_{\mathrm{MP}}^{\pi_*} = V_{\mathrm{MP}}$.

For any $k \in \mathcal{K}$, denote the worst-case kernel of $V_{\mathrm{MP}}$ in $\mathcal{P}_k$ by $P_k'$, i.e.,

$$
\sigma_{(\mathcal{P}_k)_s^a}(V_{\mathrm{MP}}) = (P_k')_s^a V_{\mathrm{MP}}, \ \forall (s,a) \in \mathcal{S} \times \mathcal{A}.
\tag{72}
$$

Since $V_{\mathrm{MP}} = \max_{a \in \mathcal{A}} Q_{\mathrm{MP}}(s, a)$, by Lemma E.1 it holds that

$$
\begin{aligned}
V_{\mathrm{MP}}^{\pi}(s) - V_{\mathrm{MP}}(s) &\leq \sum_{a \in \mathcal{A}} \pi(a|s) \left( Q_{\mathrm{MP}}^{\pi}(s, a) - Q_{\mathrm{MP}}(s, a) \right) \\
&= \sum_{a \in \mathcal{A}} \pi(a|s) \gamma \left( \max_{k \in \mathcal{K}} \sigma_{(\mathcal{P}_k)_s^a}(V_{\mathrm{MP}}^{\pi}) - \max_{k \in \mathcal{K}} \sigma_{(\mathcal{P}_k)_s^a}(V_{\mathrm{MP}}) \right) \\
&\leq \sum_{a \in \mathcal{A}} \pi(a|s) \gamma \left( \max_{k \in \mathcal{K}} ((P_k')_s^a V_{\mathrm{MP}}^{\pi}) - \max_{k \in \mathcal{K}} ((P_k'))_s^a V_{\mathrm{MP}}) \right) \\
&\leq \sum_{a \in \mathcal{A}} \pi(a|s) \gamma \max_{k \in \mathcal{K}} ((P_k')_s^a V_{\mathrm{MP}}^{\pi} - (P_k')_s^a V_{\mathrm{MP}}) \\
&= \sum_{a \in \mathcal{A}} \pi(a|s) \gamma \max_{k \in \mathcal{K}} ((P_k')_s^a (V_{\mathrm{MP}}^{\pi} - V_{\mathrm{MP}})).
\end{aligned}
\tag{73}
$$

Let $\tilde{P}_s = \arg\max_{k \in \mathcal{K}} \sum_{a \in \mathcal{A}} \pi(a|s) \gamma (P_k')_s^a (V_{\mathrm{MP}}^{\pi} - V_{\mathrm{MP}})$, we obtain

$$
V_{\mathrm{MP}}^{\pi}(s) - V_{\mathrm{MP}}(s) \leq \gamma \tilde{P}_s (V_{\mathrm{MP}}^{\pi} - V_{\mathrm{MP}}),
\tag{74}
$$

and hence by noting that $(I - \gamma \tilde{P})^{-1}$ has all positive entries we have

$$
(I - \gamma \tilde{P})(V_{\mathrm{MP}}^{\pi} - V_{\mathrm{MP}}) \leq 0,
\tag{75}
$$

which implies that $\max_{\pi} V_{\mathrm{MP}}^{\pi} \leq V_{\mathrm{MP}}$. On the other hand, since $V_{\mathrm{MP}}^{\pi_*} = V_{\mathrm{MP}}$ and $\max_{\pi} V_{\mathrm{MP}}^{\pi} \geq V_{\mathrm{MP}}^{\pi_*}$, together with the upper bound it follows that $\max_{\pi} V_{\mathrm{MP}}^{\pi} = V_{\mathrm{MP}}$. Note that $V_{\mathrm{MP}}(s) = \max_{a \in \mathcal{A}} Q_{\mathrm{MP}}(s, a)$, hence $\arg\max_{\pi} V_{\mathrm{MP}}^{\pi} = \pi_*$, which completes the proof.

### E.3. Proof of Lemma E.1

The proof of the first two statements can be directly derived from reverse triangle inequality and Lipschitz continuity, and is thus omitted. We now consider the third inequality.

Denote $X_k^i$ the $i$-th element of $X_k$, we have

$$
X_k^i - \max_{k \in \mathcal{K}} X_k^i \leq 0, \forall i,
\tag{76}
$$

which indicates that $X_k - \max_{k \in \mathcal{K}} X_k$ is a vector with non-positive elements. Therefore

$$
\begin{aligned}
\max_{k \in \mathcal{K}} \| X_k - \max_{k \in \mathcal{K}} X_k \| &= \max_{k \in \mathcal{K}} \max_{i} | X_k^i - \max_{k \in \mathcal{K}} X_k^i | \\
&= \max_{i} \max_{k \in \mathcal{K}} (\max_{k \in \mathcal{K}} X_k^i - X_k^i) \\
&= \max_{i} (\max_{k \in \mathcal{K}} X_k^i - \min_{k \in \mathcal{K}} X_k^i).
\end{aligned}
\tag{77}
$$

We also have

$$
\max_{k \in \mathcal{K}} \| X_k \| = \max_{k \in \mathcal{K}} \max_{i} | X_k^i | \geq \max_{i} | \min_{k \in \mathcal{K}} X_k^i |.
\tag{78}
$$

Hence for each $i$,

$$
\max_{k \in \mathcal{K}} X_k^i - \min_{k \in \mathcal{K}} X_k^i \leq | \max_{k \in \mathcal{K}} X_k^i | + | \min_{k \in \mathcal{K}} X_k^i | \leq 2 \max_{k \in \mathcal{K}} \| X_k \|.
\tag{79}
$$

By taking the maximum over $i$, we thus obtain

$$
\max_{k \in \mathcal{K}} \| X_k - \max_{k \in \mathcal{K}} X_k \| = \max_{i} (\max_{k \in \mathcal{K}} X_k^i - \min_{k \in \mathcal{K}} X_k^i) \leq 2 \max_{k \in \mathcal{K}} \| X_k \|,
\tag{80}
$$

which completes the proof.

### E.4. Proof of Theorem E.2

**Theorem E.2.** *Let* $E - 1 \leq \min \frac{1}{\lambda} \{ \frac{\gamma}{1-\gamma}, \frac{1}{K} \}$, *and* $\lambda = \frac{4 \log^2(TK)}{T(1-\gamma)}$, *we set* ***Max-Aggregation*** *of* $Q_k(s, a)$ *to be* $\max_k Q_k(s, a)$. *If* $\hat{\mathbf{T}}_k = \mathbf{T}_k$, *it holds that*

$$\left\| Q_{\mathrm{MP}} - \max_{k \in \mathcal{K}} Q_k \right\| \leq \tilde{\mathcal{O}} \left( \frac{1}{TK} + \frac{(E-1)\Gamma}{T} \right). \tag{81}$$

In this proof, we follow the same road map with Theorem D.1 based on Lemma E.1. we first assume that $E - 1 \leq \frac{1-\gamma}{4\gamma\lambda}$ and $\lambda \leq \frac{1}{E}$ to establish the general convergence rate, which is independent of $K$. Then we prove that carefully selected $E$ and $\lambda$ can balance each term in the convergence rate to achieve partial linear speedup.

Denote $\hat{Q}^{t+1}$ and $Q_k^{t+1}$ the values of $\hat{Q}$ and $Q_k$ at iteration $t + 1$. Since $\hat{\mathbf{T}}_k = \mathbf{T}_k$, we have

$$\begin{aligned}
\hat{Q}^{t+1} &= \max_{k \in \mathcal{K}} Q_k^{t+1} \\
&= \max_{k \in \mathcal{K}} ((1 - \lambda)Q_k^t + \lambda(r + \gamma \sigma_{\mathcal{P}_k}(V_k^t))),
\end{aligned} \tag{82}$$

where $V_k^t(s) = \max_{a \in \mathcal{A}} Q_k^t(s, a)$. We also denote the value function associate with $Q_{\mathrm{MP}}$ by $V_{\mathrm{MP}} = \max_{a \in \mathcal{A}} Q_{\mathrm{MP}}(s, a)$. We define the iteration error as $\Delta^{t+1} := Q_{\mathrm{MP}} - \hat{Q}^{t+1}$, additionally, $\Delta^0 := Q_{\mathrm{MP}} - Q^0$. The iteration error then takes the following form:

$$\begin{aligned}
\Delta^{t+1} &= Q_{\mathrm{MP}} - \hat{Q}^{t+1} \\
&= \max_{k \in \mathcal{K}} (Q_{\mathrm{MP}} - ((1 - \lambda)Q_k^t + \lambda(r + \gamma \sigma_{\mathcal{P}_k}(V_k^t)))) \\
&= \max_{k \in \mathcal{K}} ((1 - \lambda)(Q_{\mathrm{MP}} - Q_k^t) + \lambda(Q_{\mathrm{MP}} - r - \gamma \sigma_{\mathcal{P}_k}(V_k^t)) \\
&= (1 - \lambda)\Delta^t + \gamma\lambda \max_{k \in \mathcal{K}} (\sigma_{\mathcal{P}_k}(V_{\mathrm{MP}}) - \sigma_{\mathcal{P}_k}(V_k^t)) \\
&= (1 - \lambda)^{t+1}\Delta^0 + \gamma\lambda \sum_{i=0}^{t} (1 - \lambda)^{t-i} \max_{k \in \mathcal{K}} (\sigma_{\mathcal{P}_k}(V_{\mathrm{MP}}) - \sigma_{\mathcal{P}_k}(V_k^i)).
\end{aligned} \tag{83}$$

By taking the $l_\infty$-norm on both sides, we obtain

$$\left\| \Delta^{t+1} \right\| \leq (1 - \lambda)^{t+1} \left\| \Delta^0 \right\| + \left\| \gamma\lambda \sum_{i=0}^{t} (1 - \lambda)^{t-i} \max_{k \in \mathcal{K}} (\sigma_{\mathcal{P}_k}(V_{\mathrm{MP}}) - \sigma_{\mathcal{P}_k}(V_k^i)) \right\|. \tag{84}$$

Since $0 \leq Q^0 \leq \frac{1}{1-\gamma}$, the first term in (30) can be bounded as

$$(1 - \lambda)^{t+1} \left\| \Delta^0 \right\| \leq (1 - \lambda)^{t+1} \frac{1}{1 - \gamma}. \tag{85}$$

To bound the second term in (83), we divide the summation into two parts. Define the most recent aggregation step of $t$ as $\chi(t) := t - (t \mod E)$. Then for any $\beta E \leq t \leq T$, we have

$$\begin{aligned}
&\sum_{i=0}^{t} (1-\lambda)^{t-i} \lambda\gamma \left\| \max_{k \in \mathcal{K}} (\sigma_{\mathcal{P}_k}(V_{\mathrm{MP}}) - \sigma_{\mathcal{P}_k}(V_k^i)) \right\| \\
&= \sum_{i=0}^{\chi(t)-\beta E} (1-\lambda)^{t-i} \lambda\gamma \left\| \max_{k \in \mathcal{K}} (\sigma_{\mathcal{P}_k}(V_{\mathrm{MP}}) - \sigma_{\mathcal{P}_k}(V_k^i)) \right\| + \sum_{i=\chi(t)-\beta E+1}^{t} (1-\lambda)^{t-i} \lambda\gamma \left\| \max_{k \in \mathcal{K}} (\sigma_{\mathcal{P}_k}(V_{\mathrm{MP}}) - \sigma_{\mathcal{P}_k}(V_k^i)) \right\| \\
&\leq \frac{\gamma}{1-\gamma} (1 - \lambda)^{t-\chi(t)+\beta E} + \sum_{i=\chi(t)-\beta E+1}^{t} (1-\lambda)^{t-i} \lambda\gamma \left\| \max_{k \in \mathcal{K}} (\sigma_{\mathcal{P}_k}(V_{\mathrm{MP}}) - \sigma_{\mathcal{P}_k}(V_k^i)) \right\|.
\end{aligned} \tag{86}$$

Before proceeding with the proof, we introduce the following lemma to bound the second term in (86). The lemma can be seen as a variant of Lemma D.2, and the proof can be found in Appendix E.5. We define $\Delta_k^t := Q_{\mathrm{MP}} - Q_k^t$ the local iteration error of agent $k$. We also define $\hat{\sigma}_{(\mathcal{P}_k)_s^a}(v) := \max_{k \in \mathcal{K}} \sigma_{(\mathcal{P}_k)_s^a}(v)$ for any vector $v$.

**Lemma E.3.** *If* $t \mod E = 0$, *then* $\|\max_{k\in\mathcal{K}}(\sigma_{\mathcal{P}_k}(V_{\mathrm{MP}}) - \sigma_{\mathcal{P}_k}(V_k^t))\| \leq \|\Delta^t\|$. *Otherwise,*

$$\left\|\max_{k\in\mathcal{K}}(\sigma_{\mathcal{P}_k}(V_{\mathrm{MP}}) - \sigma_{\mathcal{P}_k}(V_k^t))\right\| \leq \left\|\Delta^{\chi(t)}\right\| + 2\lambda \max_{k\in\mathcal{K}} \sum_{t'=\chi(t)}^{t-1} \left\|\Delta_k^{t'}\right\|$$
$$+ \gamma\lambda(t-1-\chi(t)) \max_{(k,s,a)\in\mathcal{K}\times\mathcal{S}\times\mathcal{A}} \left|\sigma_{(\mathcal{P}_k)_s^a}(V_{\mathrm{MP}}) - \hat{\sigma}_{(\mathcal{P}_k)_s^a}(V_{\mathrm{MP}})\right|, \qquad (87)$$

*where we use the convention that* $\sum_{t'=\chi(t)}^{\chi(t)-1}\left\|\Delta_{t'}^k\right\| = 0$.

Then by Lemma E.3, (86) yields

$$\sum_{i=\chi(t)-\beta E+1}^{t} (1-\lambda)^{t-i}\lambda\gamma \left\|\max_{k\in\mathcal{K}}(\sigma_{\mathcal{P}_k}(V_{\mathrm{MP}}) - \sigma_{\mathcal{P}_k}(V_k^i))\right\|$$

$$\leq \sum_{i=\chi(i)-\beta E+1}^{t} (1-\lambda)^{t-i}\lambda\gamma \left(\left\|\Delta^{\chi(i)}\right\| + 2\lambda \max_{k\in\mathcal{K}} \sum_{j=\chi(t)}^{i-1} \left\|\Delta_k^j\right\|\right.$$

$$\left.+\gamma\lambda(t-1-\chi(i)) \max_{(k,s,a)\in\mathcal{K}\times\mathcal{S}\times\mathcal{A}} \left|\sigma_{(\mathcal{P}_k)_s^a}(V_{\mathrm{MP}}) - \hat{\sigma}_{(\mathcal{P}_k)_s^a}(V_{\mathrm{MP}})\right|\right)$$

$$\leq \sum_{i=\chi(i)-\beta E+1}^{t} (1-\lambda)^{t-i}\lambda\gamma \left(\left\|\Delta^{\chi(i)}\right\| + 2\lambda \max_{k\in\mathcal{K}} \sum_{j=\chi(t)}^{i-1} \left\|\Delta_k^j\right\| + \gamma\lambda\Gamma\frac{E-1}{1-\gamma}\right). \qquad (88)$$

Furthermore, the local iteration error $\left\|\Delta_k^j\right\|$ can be bounded using the following lemma. This lemma can also be seen as a variant of Lemma D.3 and the proof of can be found in Appendix E.6.

**Lemma E.4.** *If* $\lambda \leq \frac{1}{E}$, *then for any* $t \in [0, T-1]$ *and* $k \in \mathcal{K}$, $\|\Delta_k^t\| \leq \left\|\Delta^{\chi(t)}\right\| + \frac{3\gamma}{1-\gamma}\lambda(E-1)\Gamma$.

Then by Lemma E.4, we have

$$\sum_{i=\chi(i)-\beta E+1}^{t} (1-\lambda)^{t-i}\lambda\gamma 2\lambda \max_{k\in\mathcal{K}} \sum_{j=\chi(t)}^{i-1} \left\|\Delta_k^j\right\|$$

$$\leq 2\lambda^2\gamma \sum_{i=\chi(t)-\beta E+1}^{t} (1-\lambda)^{t-i} \sum_{j=\chi(i)}^{i-1} \left(\left\|\Delta^{\chi(i)}\right\| + 3\frac{\gamma}{1-\gamma}\lambda(E-1)\Gamma\right)$$

$$\leq 2\lambda\gamma(E-1) \max_{\chi(t)-\beta E\leq i\leq t} \left\|\Delta^{\chi(i)}\right\| + \frac{6\gamma^2\lambda^2}{1-\gamma}(E-1)^2\Gamma. \qquad (89)$$

By combining both bounds (88) and (89), we obtain

$$\sum_{i=\chi(t)-\beta E+1}^{t} (1-\lambda)^{t-i}\lambda\gamma \left\|\max_{k\in\mathcal{K}}(\sigma_{\mathcal{P}_k}(V_{\mathrm{MP}}) - \sigma_{\mathcal{P}_k}(V_k^i))\right\|$$

$$\leq \gamma \max_{\chi(t)-\beta E\leq i\leq t} \left\|\Delta^{\chi(i)}\right\| + 2\lambda\gamma(E-1) \max_{\chi(t)-\beta E\leq i\leq t} \left\|\Delta^{\chi(i)}\right\| + \frac{6\gamma^2\lambda^2}{1-\gamma}(E-1)^2\Gamma$$

$$+ \sum_{i=\chi(t)-\beta E+1}^{t} (1-\lambda)^{t-i}\lambda\gamma(\frac{\gamma\lambda}{1-\gamma}(E-1)\Gamma)$$

$$= \gamma(1 + 2\lambda(E-1)) \max_{\chi(t)-\beta E\leq i\leq t} \left\|\Delta^{\chi(i)}\right\| + \frac{\gamma^2}{1-\gamma}(6\lambda^2(E-1)^2 + \lambda(E-1))\Gamma. \qquad (90)$$

Plugging the bound (90) into (86) then yields

$$
\sum_{i=0}^{t}(1-\lambda)^{t-i}\lambda\gamma\left\|\max_{k\in\mathcal{K}}(\sigma_{\mathcal{P}_k}(V_{\mathrm{MP}})-\sigma_{\mathcal{P}_k}(V_k^i))\right\|
$$

$$
\leq \frac{\gamma}{1-\gamma}(1-\lambda)^{t-\chi(t)+\beta E}+\gamma(1+2\lambda(E-1))\max_{\chi(t)-\beta E\leq i\leq t}\left\|\Delta^{\chi(i)}\right\|
$$

$$
+\frac{\gamma^2}{1-\gamma}(6\lambda^2(E-1)^2+\lambda(E-1))\Gamma. \tag{91}
$$

Consequently, the iteration error $\left\|\Delta^{t+1}\right\|$ can be bounded using (85) and (91) for all rounds $t\in[0,T-1]$:

$$
\left\|\Delta^{t+1}\right\|\leq(1-\lambda)^{t+1}\frac{1}{1-\gamma}+\frac{\gamma}{1-\gamma}(1-\lambda)^{t-\chi(t)+\beta E}
$$

$$
+\gamma(1+2\lambda(E-1))\max_{\chi(t)-\beta E\leq i\leq t}\left\|\Delta^{\chi(i)}\right\|+\frac{\gamma^2}{1-\gamma}(6\lambda^2(E-1)^2+\lambda(E-1))\Gamma
$$

$$
\leq\gamma(1+2\lambda(E-1))\max_{\chi(t)-\beta E\leq i\leq t}\left\|\Delta^{\chi(i)}\right\|+\frac{2}{1-\gamma}(1-\lambda)^{\beta E}
$$

$$
+\frac{\gamma^2}{1-\gamma}(6\lambda^2(E-1)^2+\lambda(E-1))\Gamma. \tag{92}
$$

Let $\rho:=\frac{2}{1-\gamma}(1-\lambda)^{\beta E}+\frac{\gamma^2}{1-\gamma}(6\lambda^2(E-1)^2+\lambda(E-1))\Gamma$, then with the assumption $\lambda\leq\frac{1-\gamma}{4\gamma(E-1)}$, (92) can be written as

$$
\left\|\Delta^{t+1}\right\|\leq\frac{1+\gamma}{2}\max_{\chi(t)-\beta E\leq i\leq t}\left\|\Delta^{\chi(i)}\right\|+\rho. \tag{93}
$$

Unrolling the above recursion $L$ times with $L\beta E\leq t\leq T-1$, we obtain

$$
\left\|\Delta^{t+1}\right\|\leq(\frac{1+\gamma}{2})^L\max_{\chi(t)-L\beta E\leq i\leq t}\left\|\Delta^{\chi(i)}\right\|+\sum_{i=0}^{L-1}(\frac{1+\gamma}{2})^i\rho
$$

$$
\leq(\frac{1+\gamma}{2})^L\frac{1}{1-\gamma}+\frac{2}{1-\gamma}\rho. \tag{94}
$$

By choosing $\beta=\lfloor\frac{1}{E}\sqrt{\frac{(1-\gamma)T}{2\lambda}}\rfloor$, $L=\lceil\sqrt{\frac{\lambda T}{1-\gamma}}\rceil$ and $t+1=T$, (94) yields

$$
\left\|\Delta^T\right\|\leq\frac{1}{1-\gamma}(\frac{1+\gamma}{2})^{\sqrt{\frac{\lambda T}{1-\gamma}}}+\frac{2}{1-\gamma}\left(\frac{2}{1-\gamma}(1-\lambda)^{\beta E}+\frac{\gamma^2}{1-\gamma}(6\lambda^2(E-1)^2+\lambda(E-1))\Gamma\right)
$$

$$
\leq\frac{1}{1-\gamma}\exp\{-\frac{1}{2}\sqrt{(1-\gamma)\lambda T}\}+\frac{4}{(1-\gamma)^2}\exp\{-\sqrt{(1-\gamma)\lambda T}\}
$$

$$
+\frac{2\gamma^2}{(1-\gamma)^2}(6\lambda^2(E-1)^2+\lambda(E-1))\Gamma
$$

$$
\leq\frac{4}{(1-\gamma)^2}\exp\{-\frac{1}{2}\sqrt{(1-\gamma)\lambda T}\}+\frac{2\gamma^2}{(1-\gamma)^2}(6\lambda^2(E-1)^2+\lambda(E-1))\Gamma. \tag{95}
$$

(95) reveals the effect of $E$ and $\lambda$ on the convergence rate. Although $E>1$ slows down the convergence, we can set $(E-1)\leq\min\frac{1}{\lambda}\{\frac{\gamma}{1-\gamma},\frac{1}{K}\}$, and $\lambda=\frac{4\log^2(TK)}{T(1-\gamma)}$ to bring partial linear speedup with respect to the number of agents $K$. Then (92) becomes

$$
\left\|\Delta^T\right\|\leq\tilde{\mathcal{O}}\left(\frac{1}{TK}+\frac{(E-1)\Gamma}{T}\right) \tag{96}
$$

for $T\geq E$, which completes the proof.

### E.5. Proof of Lemma E.3

When $t \mod E = 0$, i.e., $t$ is an aggregation step, $Q_k^t = Q_{k'}^t$ and the associate value functions $V_k^t = V_{k'}^t$ for any $k$ and $k' \in \mathcal{K}$. We denote $\hat{V}^t = \max_{k \in \mathcal{K}} V_k^t$, then we have

$$
\begin{aligned}
\max_{k \in \mathcal{K}} (\sigma_{(\mathcal{P}_k)_s^a}(V_{\mathrm{MP}}) - \sigma_{(\mathcal{P}_k)_s^a}(V_k^t)) &\leq \max_{k \in \mathcal{K}} (\sigma_{(\mathcal{P}_k)_s^a}(V_{\mathrm{MP}}) - \sigma_{(\mathcal{P}_k)_s^a}(\hat{V}^t)) \\
&\leq \max_{k \in \mathcal{K}} \left\| V_{\mathrm{MP}} - \hat{V}^t \right\| \\
&\leq \left\| Q_{\mathrm{MP}} - \hat{Q}^t \right\| \\
&= \left\| \Delta^t \right\|.
\end{aligned}
\tag{97}
$$

For general $t$, we have

$$
\begin{aligned}
\left\| \max_{k \in \mathcal{K}} (\sigma_{\mathcal{P}_k}(V_{\mathrm{MP}}) - \sigma_{\mathcal{P}_k}(V_k^t)) \right\| &= \left\| \max_{k \in \mathcal{K}} (\sigma_{\mathcal{P}_k}(V_{\mathrm{MP}}) - \sigma_{\mathcal{P}_k}(V_k^{\chi(t)}) + \sigma_{\mathcal{P}_k}(V_k^{\chi(t)}) - \sigma_{\mathcal{P}_k}(V_k^t)) \right\| \\
&\leq \left\| \max_{k \in \mathcal{K}} (\sigma_{\mathcal{P}_k}(V_{\mathrm{MP}}) - \sigma_{\mathcal{P}_k}(V_k^{\chi(t)})) \right\| + \left\| \max_{k \in \mathcal{K}} (\sigma_{\mathcal{P}_k}(V_k^{\chi(t)}) - \sigma_{\mathcal{P}_k}(V_k^t)) \right\| \\
&\leq \left\| \Delta^{\chi(t)} \right\| + \left\| \max_{k \in \mathcal{K}} (\sigma_{\mathcal{P}_k}(V_k^{\chi(t)}) - \sigma_{\mathcal{P}_k}(V_k^t)) \right\| \\
&\leq \left\| \Delta^{\chi(t)} \right\| + \max_{k \in \mathcal{K}} \left\| V_k^{\chi(t)} - V_k^t \right\|.
\end{aligned}
\tag{98}
$$

For any state $s$, we have

$$
\begin{aligned}
V_k^t(s) - V_k^{\chi(t)}(s) &= Q_k^t(s, a_k^t(s)) - Q_k^{\chi(t)}(s, a_k^{\chi(t)}(s)) \\
&\overset{(a)}{\leq} Q_k^t(s, a_k^t(s)) - Q_k^{\chi(t)}(s, a_k^t(s)) \\
&= Q_k^t(s, a_k^t(s)) - Q_k^{t-1}(s, a_k^t(s)) + Q_k^{t-1}(s, a_k^t(s)) - Q_k^{t-2}(s, a_k^t(s)) \\
&\quad + \cdots + Q_k^{\chi(t)+1}(s, a_k^t(s)) - Q_k^{\chi(t)}(s, a_k^t(s)).
\end{aligned}
\tag{99}
$$

where $a_k^t(s) = \arg\max_{a \in \mathcal{A}} Q_k^t(s, a)$ and inequality (a) holds because $Q_k^{\chi(t)}(s, a_k^t(s)) \leq Q_k^{\chi(t)}(s, a_k^{\chi(t)}(s))$.
For each $t'$ such that $\chi(t) \leq t' \leq t$, it holds that,

$$
\begin{aligned}
Q_k^{t'+1}(s, a_k^t(s)) - Q_k^{t'}(s, a_k^t(s)) &= (1-\lambda)Q_k^{t'}(s, a_k^t(s)) + \lambda(r(s, a_k^t(s)) + \gamma \sigma_{(\mathcal{P}_k)_s^{a_k^t(s)}}(V_k^{t'})) - Q_k^{t'}(s, a_k^t(s)) \\
&\overset{(a)}{=} -\lambda Q_k^{t'}(s, a_k^t(s)) + \lambda(Q_{\mathrm{MP}}(s, a_k^t(s)) - r(s, a_k^t(s)) - \gamma \hat{\sigma}_{(\mathcal{P}_k)_s^{a_k^t(s)}}(V_{\mathrm{MP}}) \\
&\quad + r(s, a_k^t(s)) + \gamma \sigma_{(\mathcal{P}_k)_s^{a_k^t(s)}}(V_k^{t'})) \\
&\leq 2\lambda \left\| \Delta_k^{t'} \right\| + \gamma\lambda(\sigma_{(\mathcal{P}_k)_s^{a_k^t(s)}}(V_{\mathrm{MP}}) - \hat{\sigma}_{(\mathcal{P}_k)_s^{a_k^t(s)}}(V_{\mathrm{MP}})),
\end{aligned}
\tag{100}
$$

where equality (a) follows from the average optimal Bellman equation. Thus,

$$
\begin{aligned}
V_k^t(s) - V_k^{\chi(t)}(s) &\leq \sum_{t'=\chi(t)}^{t-1} (Q_k^{t'+1}(s, a_k^t(s)) - Q_k^{t'}(s, a_k^t(s))) \\
&= 2\lambda \sum_{t'=\chi(t)}^{t-1} \left\| \Delta_k^{t'} \right\| + \gamma\lambda(t - 1 - \chi(t))(\sigma_{(\mathcal{P}_k)_s^{a_k^t(s)}}(V_{\mathrm{MP}}) - \hat{\sigma}_{(\mathcal{P}_k)_s^{a_k^t(s)}}(V_{\mathrm{MP}})).
\end{aligned}
\tag{101}
$$

Similarly, we have

$$V_k^t(s) - V_k^{\chi(t)}(s) \geq \sum_{t'=\chi(t)}^{t-1} (Q_k^{t'+1}(s, a_k^{\chi(t)}(s)) - Q_k^{t'}(s, a_k^{\chi(t)}(s)))$$

$$\geq -2\lambda \sum_{t'=\chi(t)}^{t-1} \left\| \Delta_k^{t'} \right\| + \gamma\lambda(t - 1 - \chi(t))(\sigma_{(\mathcal{P}_k)_s^{a_k^t(s)}}(V_{\mathrm{MP}}) - \hat{\sigma}_{(\mathcal{P}_k)_s^{a_k^t(s)}}(V_{\mathrm{MP}})). \tag{102}$$

By plugging the bounds in (101) and in (102) back into (98), we obtain

$$\left\| \max_{k\in\mathcal{K}}(\sigma_{\mathcal{P}_k}(V_{\mathrm{MP}}) - \sigma_{\mathcal{P}_k}(V_k^t)) \right\| \leq \left\| \Delta_{\chi(t)} \right\| + \max_{k\in\mathcal{K}} \left\| V_k^{\chi(t)} - V_k^t \right\|$$

$$\leq \left\| \Delta_{\chi(t)} \right\| + 2\lambda \max_{k\in\mathcal{K}} \sum_{t'=\chi(t)}^{t-1} \| \Delta_k^{t'} \|$$

$$+ \gamma\lambda(t - 1 - \chi(t)) \max_{k,s,a\in\mathcal{K}\times\mathcal{S}\times\mathcal{A}} |\sigma_{(\mathcal{P}_k)_s^a}(V_{\mathrm{MP}}) - \hat{\sigma}_{(\mathcal{P}_k)_s^a}(V_{\mathrm{MP}})|. \tag{103}$$

This hence completes the proof.

### E.6. Proof of Lemma E.4

When $t \mod E = 0$, then $\Delta_k^t = \Delta^{\chi(t)}$. When $t \mod E \neq 0$, we have

$$Q_k^t = (1-\lambda)Q_k^{t-1} + \lambda(r + \gamma\sigma_{\mathcal{P}_k}(V_k^{t-1}))$$
$$= (1-\lambda)Q_k^{t-1} + \lambda(Q_{\mathrm{MP}} - r - \gamma\hat{\sigma}_{\mathcal{P}_k}(V_{\mathrm{MP}}) + r + \gamma\sigma_{\mathcal{P}_k}(V_k^{t-1})). \tag{104}$$

Thus

$$\Delta_k^t = (1-\lambda)\Delta_k^{t-1} + \lambda\gamma(\hat{\sigma}_{\mathcal{P}_k}(V_{\mathrm{MP}}) - \sigma_{\mathcal{P}_k}(V_k^{t-1}))$$
$$= (1-\lambda)\Delta_k^{t-1} + \lambda\gamma(\hat{\sigma}_{\mathcal{P}_k}(V_{\mathrm{MP}}) - \sigma_{\mathcal{P}_k}(V_{\mathrm{MP}}) + \sigma_{\mathcal{P}_k}(V_{\mathrm{MP}}) - \sigma_{\mathcal{P}_k}(V_k^{t-1}))$$
$$= (1-\lambda)^{t-\chi(t)}\Delta^{\chi(i)} + \gamma\lambda \sum_{t'=\chi(t)}^{t-1} (1-\lambda)^{t-t'-1}(\hat{\sigma}_{\mathcal{P}_k}(V_{\mathrm{MP}}) - \sigma_{\mathcal{P}_k}(V_{\mathrm{MP}}))$$
$$+ \gamma\lambda \sum_{t'=\chi(t)}^{t-1} (1-\lambda)^{t-t'-1}(\sigma_{\mathcal{P}_k}(V_{\mathrm{MP}}) - \sigma_{\mathcal{P}_k}(V_j^k)). \tag{105}$$

For any state-action pair $(s, a) \in \mathcal{S} \times \mathcal{A}$,

$$\left| (1-\lambda)^{t-\chi(t)}\Delta^{\chi(t)}(s, a) \right| \leq (1-\lambda)^{t-\chi(t)} \left\| \Delta^{\chi(t)} \right\|. \tag{106}$$

Note that

$$\gamma\lambda \sum_{t'=\chi(t)}^{t-1} (1-\lambda)^{t-t'-1}(\hat{\sigma}_{\mathcal{P}_k}(V_{\mathrm{MP}}) - \sigma_{\mathcal{P}_k}(V_{\mathrm{MP}})) \leq \frac{\gamma}{1-\gamma}\lambda \sum_{t'=\chi(t)}^{t-1} (1-\lambda)^{t-t'-1}\Gamma$$

$$\leq \frac{\gamma}{1-\gamma}\lambda(E-1)\Gamma, \tag{107}$$

for all $(s, a) \in \mathcal{S} \times \mathcal{A}$, $t \in [0, T-1]$ and $k \in \mathcal{K}$. In addition, we have

$$\left\| \gamma\lambda \sum_{t'=\chi(t)}^{t-1} (1-\lambda)^{t-t'-1}(\sigma_{\mathcal{P}_k}(V_{\mathrm{MP}}) - \sigma_{\mathcal{P}_k}(V_k^{t'})) \right\| \leq \gamma\lambda \sum_{t'=\chi(t)}^{t-1} (1-\lambda)^{t-t'-1} \left\| \Delta_k^{t'} \right\|. \tag{108}$$

By combining the bounds in (106), (107), and (108), we obtain

$$
\begin{aligned}
\left\|\Delta_k^t\right\| &\leq (1-\lambda)^{t-\chi(t)}\left\|\Delta^{\chi(t)}\right\| + \frac{\gamma}{1-\gamma}\lambda(E-1)\Gamma + \gamma\lambda\sum_{t'=\chi(t)}^{t-1}(1-\lambda)^{t-t'-1}\left\|\Delta_k^{t'}\right\| \\
&\leq (1-(1-\gamma)\lambda)^{t-\chi(t)}\left\|\Delta^{\chi(t)}\right\| + (1+\gamma\lambda)^{t-\chi(t)}(\frac{\gamma}{1-\gamma}\lambda(E-1)\Gamma),
\end{aligned}
\tag{109}
$$

where the last inequality can be shown via inducting on $t-\chi(t)\in\{0,\cdots,E-1\}$. When $\lambda\leq\frac{1}{E}$,

$$
(1+\gamma\lambda)^{t-\chi(t)}\leq(1+\lambda)^E\leq(1+1/E)^E\leq e\leq 3.
\tag{110}
$$

Hence

$$
\left\|\Delta_k^t\right\|\leq\left\|\Delta^{\chi(t)}\right\| + 3\frac{\gamma}{1-\gamma}\lambda(E-1)\Gamma,
\tag{111}
$$

which completes the proof.

### E.7. Proof of Theorem 6.4

Since $\mathbb{E}[\hat{\mathbf{T}}_k]=\mathbf{T}_k$, by Lemma 6.2 it follows that $\mathbb{E}[\hat{\mathbf{T}}_{\mathrm{MLMC}}]=\mathbf{T}_{\mathrm{MP}}$. Then by replacing $\mathbf{T}_k$ and $\mathbf{T}_{\mathrm{MP}}$ with their expectation versions, the proof follows naturally as that in Theorem E.2, and is thus omitted.

# F. Proof of Proposition 7.1

*Proof.* We show that $\cap_k^K \mathcal{P}_k$ satisfies the condition. We denote the robust value function w.r.t. $\cap_k^K \mathcal{P}_k$ by $V_\cap^\pi$, then we have that

$$
\begin{aligned}
V_\cap^\pi - V_{\mathrm{MP}}^\pi &= \gamma \sigma_{\cap \mathcal{P}_k}^\pi (V_\cap^\pi) - \gamma \max_k \sigma_{\mathcal{P}_k}^\pi (V_{\mathrm{MP}}^\pi) \\
&= \gamma \sigma_{\cap \mathcal{P}_k}^\pi (V_\cap^\pi) - \gamma \sigma_{\cap \mathcal{P}_k}^\pi (V_{\mathrm{MP}}^\pi) + \gamma \sigma_{\cap \mathcal{P}_k}^\pi (V_{\mathrm{MP}}^\pi) - \gamma \max_k \sigma_{\mathcal{P}_k}^\pi (V_{\mathrm{MP}}^\pi) \\
&\geq \gamma \sigma_{\cap \mathcal{P}_k}^\pi (V_\cap^\pi) - \gamma \sigma_{\cap \mathcal{P}_k}^\pi (V_{\mathrm{MP}}^\pi),
\end{aligned}
\tag{112}
$$

which is due to $\cap \mathcal{P}_k \subseteq \mathcal{P}_k$. Thus

$$
V_\cap^\pi - V_{\mathrm{MP}}^\pi \geq \gamma \sigma_{\cap \mathcal{P}_k}^\pi (V_\cap^\pi) - \gamma \sigma_{\cap \mathcal{P}_k}^\pi (V_{\mathrm{MP}}^\pi) \geq \gamma \mathsf{P}'(V_\cap^\pi - V_{\mathrm{MP}}^\pi),
\tag{113}
$$

and thus

$$
V_\cap^\pi \geq V_{\mathrm{MP}}^\pi.
\tag{114}
$$

$\square$

# G. Model-Free Algorithm Design and Analysis

We then develop studies for model-free setting. We will focus on $l_p$-norm defined uncertainty set (Kumar et al., 2023b; Derman et al., 2021):

$$
\mathcal{P}_{s,a} = \{P_{s,a} + q : q \in \mathcal{Q}\}, \mathcal{Q} = \left\{ q \in \mathbb{R}^S : \sum_i q(i) = 0, \|q\|_\alpha \leq \Gamma_{s,a} \right\},
\tag{115}
$$

where $\Gamma_{s,a}$ is small enough so that $P_{s,a} + q \in \Delta(\mathcal{S}), \forall q \in \mathcal{Q}$.

For these uncertainty sets, their robust Bellman operator can be estimated through a model-free estimator. Specifically, for any $p$, set

$$
\kappa(V) \triangleq R \min_{w \in \mathbb{R}} \|we - V\|_q,
\tag{116}
$$

with $q = \frac{1}{1 - \frac{1}{p}}$. For popular choices of $p$, the optimization problem in (116) has a closed-form solution, specified in Table 5 (Kumar et al., 2023b). It then holds that

| $p$ | $\kappa(v)$ |
|---|---|
| $\infty$ | $\frac{\max_s v(s) - \min_s v(s)}{2}$ |
| $2$ | $\sqrt{\sum_s \left(v(s) - \frac{\sum_s v(s)}{S}\right)^2}$ |
| $1$ | $\sum_{i=1}^{\lfloor (S+1)/2 \rfloor} v(s_i) - \sum_{i=\lfloor (S+1)/2 \rfloor}^{S} v(s_i)$ |

*Table 5.* Penalty term for $l_p$-norm uncertainty set

**Lemma G.1.** *(Theorem 1 in (Kumar et al., 2023b)) Let $\mathcal{P}_{s,a}$ be the uncertainty set defined using the $l_p$-norm. For any vector $V$, the following relationship holds:*

$$
\sigma_{\mathcal{P}_{s,a}}(V) = P_{s,a}V - \kappa_{s,a}(V),
\tag{117}
$$

*where $\kappa$ is defined as in* (116).

With these results, we use the model-free estimator $V(s') - \Gamma \kappa(V)$ to estimate the robust Bellman operator and design model-free MTDL-Avg algorithm. The model-free variant of MDTL-Max can be designed similarly.

We then study its convergence.

---

**Algorithm 3** Model-Free MDTL-Avg

---
1: **Initialization:** $Q^k \leftarrow 0$
2: **for** $t = 0, ..., T - 1$ **do**
3:     **for** $k = 1, ..., K$ **do**
4:         **for** All $s$ **do**
5:             $V^k(s) \leftarrow \max_a Q^k(s, a)$
6:             **for** All $a$ **do**
7:                 Take action $a$ and observe the next state $s'(s, a)$
8:                 $Q^k(s, a) \leftarrow (1 - \lambda_t)Q^k(s, a) + \lambda_t(r(s, a) + \gamma V^k(s'(s, a)) - \gamma \Gamma \kappa(V^k))$
9:                 $Q^k(s, a) \leftarrow \max\{0, Q^k(s, a)\}$
10:             **end for**
11:         **end for**
12:     **end for**
13:     **if** $t \equiv 0 (\mathrm{mod} E)$ **then**
14:         $\bar{Q}(s, a) \leftarrow \frac{\sum_k Q^k(s,a)}{K}, \forall s, a$
15:         $Q^k(s, a) \leftarrow \bar{Q}(s, a), \forall s, a, k$
16:     **end if**
17: **end for**

---

Similarly, define

$$\Delta_{t+1} := Q^* - \bar{Q}_{t+1}, \text{ and } \Delta_0 := Q^* - Q_0, \tag{118}$$

where we denote $Q^*_{\mathrm{AO}}$ by $Q^*$.

**Lemma G.2** (Error iteration). *For any $t \geq 0$,*

$$\Delta_{t+1} \leq (1 - \lambda)^{t+1}\Delta_0 + \gamma\lambda \sum_{i=0}^{t} (1 - \lambda)^{t-i} \frac{1}{K} \sum_{k=1}^{K} (\sigma^k(V^*) - \sigma^k(V_i^k))$$

$$+ \gamma\lambda \sum_{i=0}^{t} (1 - \lambda)^{t-i} \frac{1}{K} \sum_{k=1}^{K} (P^k V_i^k - P_i^k V_i^k). \tag{119}$$

*Proof.* The update of $\Delta_{t+1}$ is as follows:

$$
\begin{aligned}
\Delta_{t+1} &= Q^* - \bar{Q}_{t+1} \\
&= \frac{1}{K} \sum_{k=1}^{K} (Q^* - Q_{t+1}^k) \\
&\leq \frac{1}{K} \sum_{k=1}^{K} (Q^* - (1-\lambda)Q_t^k - \lambda(r + \gamma P_t^k V_t^k - \gamma\kappa(V_t^k))) \\
&= \frac{1}{K} \sum_{k=1}^{K} ((1-\lambda)(Q^* - Q_t^k) + \lambda(Q^* - r - \gamma P_t^k V_t^k + \gamma\kappa(V_t^k))) \\
&= (1-\lambda)\Delta_t + \gamma\lambda\frac{1}{K} \sum_{k=1}^{K} (\sigma^k(V^*) - \sigma^k(V_t^k) + \sigma^k(V_t^k) - \gamma P_t^k V_t^k + \gamma\kappa(V_t^k)) \\
&= (1-\lambda)\Delta_t + \frac{\gamma\lambda}{K} \sum_{k=1}^{K} (\sigma^k(V^*) - \sigma^k(V_t^k)) + \frac{\gamma\lambda}{K} \sum_{k=1}^{K} (P^k V_t^k - P_t^k V_t^k) \\
&= (1-\lambda)^{t+1}\Delta_0 + \gamma\lambda \sum_{i=0}^{t} (1-\lambda)^{t-i} \frac{1}{K} \sum_{k=1}^{K} (\sigma^k(V^*) - \sigma^k(V_i^k)) \\
&\quad + \gamma\lambda \sum_{i=0}^{t} (1-\lambda)^{t-i} \frac{1}{K} \sum_{k=1}^{K} (P^k V_i^k - P_i^k V_i^k).
\end{aligned}
$$

$\square$

**Lemma G.3.** *It holds that*

$$0 \leq Q_t^k(s,a) \leq \frac{1}{1-\gamma}, \tag{120}$$

$$\|Q^* - Q_t^k\| \leq \frac{1}{1-\gamma}, \tag{121}$$

$$\|V^* - V_t^k\| \leq \frac{1}{1-\gamma}, \quad \forall\, t \geq 0, \text{and } k \in [K]. \tag{122}$$

*Proof.* Due to the update rule, $Q_t^k \geq 0$. On the other hand, if $Q_t^k(s,a) = (1-\lambda_t)Q_{t-1}^k(s,a) + \lambda_t(r(s,a) + \gamma V_{t-1}^k(s'(s,a)) - \gamma\Gamma\kappa(V_{t-1}^k))$, then

$$Q_t^k(s,a) \leq (1-\lambda_t)Q_{t-1}^k(s,a) + \lambda_t(r(s,a) + \gamma V_{t-1}^k(s'(s,a))) \leq \frac{1}{1-\gamma}, \tag{123}$$

which is from introduction assumption on $V_{t-1}^k \leq \frac{1}{1-\gamma}$.

(122) then directly follows. $\square$

**Lemma G.4.** *If $t \mod E = 0$, then $\|\frac{1}{K} \sum_{k=1}^{K} (\sigma^k(V^*) - \sigma^k(V_t^k))\| \leq \|\Delta_t\|$. Otherwise,*

$$
\begin{aligned}
&\|\frac{1}{K} \sum_{k=1}^{K} (\sigma^k(V^*) - \sigma^k(V_t^k))\| \\
&\leq \|\Delta_{\chi(t)}\| + 4\lambda \frac{1}{K} \sum_{k=1}^{K} \sum_{t'=\chi(t)}^{t-1} \|\Delta_{t'}^k\| + \gamma\lambda \frac{1}{K} \sum_{k=1}^{K} \max_{s,a} |\sum_{t'=\chi(t)}^{t-1} (\sigma_{s,a}^k)_{t'}(V^*) - \bar{\sigma}_{s,a}(V^*)|.
\end{aligned}
$$

*where we use the convention that $\sum_{t'=\chi(t)}^{\chi(t)-1} \|\Delta_{t'}^k\| = 0$.*

*Proof.* When $t \mod E = 0$, i.e., $i$ is a synchronization round, $Q_t^k = Q_t^{k'}$ for any pair of agents $k, k' \in [K]$. Hence,

$$|\frac{1}{K}\sum_{k=1}^{K}(\sigma_{s,a}^k(V^*) - \sigma_{s,a}^k(V_t^k))| \tag{124}$$

$$= |\frac{1}{K}\sum_{k=1}^{K}(\sigma_{s,a}^k(V^*) - \sigma_{s,a}^k(\bar{V}_t))| \tag{125}$$

$$\leq \frac{1}{K}\sum_{k=1}^{K}\|V^* - \bar{V}_t\|$$

$$\leq \|Q^* - \bar{Q}_t\|$$

$$= \|\Delta_t\|. \tag{126}$$

For general $t$, we have

$$\|\frac{1}{K}\sum_{k=1}^{K}(\sigma^k(V^*) - \sigma^k(V_t^k))\|$$

$$= \|\frac{1}{K}\sum_{k=1}^{K}(\sigma^k(V^*) - \sigma^k(V_{\chi(t)}^k) + \sigma^k(V_{\chi(t)}^k) - \sigma^k(V_t^k))\| \tag{127}$$

$$\leq \|\frac{1}{K}\sum_{k=1}^{K}\sigma^k(V^*) - \sigma^k(V_{\chi(t)}^k)\| + \|\frac{1}{K}\sum_{k=1}^{K}(\sigma^k(V_{\chi(t)}^k) - \sigma^k(V_t^k))\|$$

$$\leq \|\Delta_{\chi(t)}\| + \|\frac{1}{K}\sum_{k=1}^{K}(\sigma^k(V_{\chi(t)}^k) - \sigma^k(V_t^k))\|$$

$$\leq \|\Delta_{\chi(t)}\| + \frac{1}{K}\sum_{k=1}^{K}\|V_{\chi(t)}^k - V_t^k\|. \tag{128}$$

For any state $s$, we have

$$V_t^k(s) - V_{\chi(t)}^k(s)$$

$$= Q_t^k(s, a_t^k(s)) - Q_{\chi(t)}^k(s, a_{\chi(t)}^k(s)), \text{ where } a_t^k(s) = \text{argmax}_{a'} Q_t^k(s, a')$$

$$\overset{(a)}{\leq} Q_t^k(s, a_t^k(s)) - Q_{\chi(t)}^k(s, a_t^k(s))$$

$$= Q_t^k(s, a_t^k(s)) - Q_{t-1}^k(s, a_t^k(s)) + Q_{t-1}^k(s, a_t^k(s)) - Q_{t-2}^k(s, a_t^k(s))$$

$$+ \cdots + Q_{\chi(t)+1}^k(s, a_t^k(s)) - Q_{\chi(t)}^k(s, a_t^k(s)). \tag{129}$$

where inequality (a) holds because $Q_{\chi(t)}^k(s, a_t^k(s)) \leq Q_{\chi(t)}^k(s, a_{\chi(t)}^k(s))$.

Now consider each $t'$ such that $\chi(t) \leq t' \leq t$. If $Q_{t'+1}^k(s, a_t^k(s)) = 0$, then

$$Q_{t'+1}^k(s, a_t^k(s)) - Q_{t'}^k(s, a_t^k(s)) = -Q_{t'}^k(s, a_t^k(s)) \leq 2\lambda\|\Delta_{t'}^k\|.$$

On the other hand, if $Q_{t'+1}^k(s, a_t^k(s)) \neq 0$, it holds that

$$
\begin{aligned}
& Q_{t'+1}^k(s, a_t^k(s)) - Q_{t'}^k(s, a_t^k(s)) \\
& = (1 - \lambda)Q_{t'}^k(s, a_t^k(s)) + \lambda(r(s, a_t^k(s)) + \gamma(P^k)_{s, a_t^k(s)}(V_{t'}^k) - \gamma \Gamma \kappa(V_{t'}^k) - Q_{t'}^k(s, a_t^k(s)) \\
& = -\lambda Q_{t'}^k(s, a_t^k(s)) + \lambda(Q^*(s, a_t^k(s)) - r(s, a_t^k(s)) - \gamma \bar{\sigma}_{s, a_t^k(s)}(V^*) + r(s, a_t^k(s)) \\
& \quad + \gamma(P_t^k)_{s, a_t^k(s)}(V_{t'}^k) - \gamma \Gamma \kappa(V_{t'}^k)) \\
& = \lambda \|\Delta_{t'}^k\| + \lambda\big(-\gamma(\bar{\sigma}_{s, a_t^k(s)}(V^*) + \gamma(\sigma_{s, a_t^k(s)}^k)_{t'}(V^*) \\
& \quad - \gamma(\sigma_{s, a_t^k(s)}^k)_{t'}(V^*) + \gamma(P_t^k)_{s, a_t^k(s)}(V_{t'}^k) - \gamma \Gamma \kappa(V_{t'}^k)\big) \\
& \leq 2\lambda \|\Delta_{t'}^k\| + \gamma\lambda|(\sigma_{s, a_t^k(s)}^k)_{t'}(V^*) - \bar{\sigma}_{s, a_t^k(s)}(V^*)|,
\end{aligned}
$$

where the last inequality is from the fact that $\kappa$ is 1-Lipschitz (Kumar et al., 2023b).

Thus,

$$
\begin{aligned}
& V_t^k(s) - V_{\chi(t)}^k(s) \\
& \leq \sum_{t'=\chi(t)}^{t-1} Q_{t'+1}^k(s, a_t^k(s)) - Q_{t'}^k(s, a_t^k(s)) \\
& \leq 2\lambda \sum_{t'=\chi(t)}^{t-1} \|\Delta_{t'}^k\| + \gamma\lambda \sum_{t'=\chi(t)}^{t-1} |(\sigma_{s, a_t^k(s)}^k)_{t'}(V^*) - \bar{\sigma}_{s, a_t^k(s)}(V^*)|.
\end{aligned} \tag{130}
$$

Similarly, we have

$$
\begin{aligned}
& V_t^k(s) - V_{\chi(t)}^k(s) \\
& \geq -2\lambda \sum_{t'=\chi(t)}^{t-1} \|\Delta_{t'}^k\| + \gamma\lambda \sum_{t'=\chi(t)}^{t-1} ((\sigma_{s, a_{\chi(t)}^k(s)}^k)_{t'}(V^*) - \bar{\sigma}_{s, a_{\chi(t)}^k(s)}(V^*)).
\end{aligned} \tag{131}
$$

Plugging the bounds in (130) and in (131) back into (127), we get

$$
\begin{aligned}
& \|\frac{1}{K} \sum_{k=1}^K (\sigma^k(V^*) - \sigma^k(V_t^k))\| \\
& \leq \|\Delta_{\chi(t)}\| + \frac{1}{K} \sum_{k=1}^K \|V_{\chi(t)}^k - V_t^k\| \\
& \leq \|\Delta_{\chi(t)}\| + 2\lambda \frac{1}{K} \sum_{k=1}^K \sum_{t'=\chi(t)}^{t-1} \|\Delta_{t'}^k\| \\
& \quad + \gamma\lambda \frac{1}{K} \sum_{k=1}^K \max_{s,a} |\sum_{t'=\chi(t)}^{t-1} (\sigma_{s,a}^k)_{t'}(V^*) - \bar{\sigma}_{s,a}(V^*)|.
\end{aligned}
$$

This hence completes the proof.

$\square$

**Lemma G.5.** *Choose $\lambda \leq \frac{1}{E}$.*

$$
\|\Delta_i^k\| \leq \|\Delta_{\chi(i)}\| + \frac{3\gamma}{1-\gamma}\lambda(E-1)\Gamma + \frac{6\gamma}{1-\gamma}\sqrt{\lambda \log \frac{SATK}{\delta}}. \tag{132}
$$

*Proof.* When $i \mod E = 0$, then $\Delta_i^k = \Delta_{\chi(i)}$.

When $i \mod E \neq 0$,

$$\Delta_i^k$$
$$= Q^* - Q_i^k \tag{133}$$
$$\leq Q^* - \left((1-\lambda)Q_{i-1}^k + \lambda(r + \gamma\sigma_{i-1}^k(V_{i-1}^k))\right) \tag{134}$$
$$= (1-\lambda)\Delta_{i-1}^k + \lambda(\gamma\bar{\sigma}(V^*) - \gamma\sigma_{i-1}^k(V_{i-1}^k))) \tag{135}$$
$$= (1-\lambda)\Delta_{i-1}^k + \lambda\gamma(\bar{\sigma}(V^*) - \sigma^k(V_{i-1}^k)) \tag{136}$$
$$= (1-\lambda)\Delta_{i-1}^k + \lambda\gamma(\bar{\sigma}(V^*) - \sigma_{i-1}^k(V^*) + \sigma_{i-1}^k(V^*) - \sigma_{i-1}^k(V_{i-1}^k))$$
$$= (1-\lambda)^{i-\chi(i)}\Delta_{\chi(i)} + \gamma\lambda\sum_{j=\chi(i)}^{i-1}(1-\lambda)^{i-j-1}(\bar{\sigma}(V^*) - \sigma_{j-1}^k(V^*))$$
$$+ \gamma\lambda\sum_{j=\chi(i)}^{i-1}(1-\lambda)^{i-j-1}(\sigma_{j-1}^k(V^*) - \sigma_{j-1}^k(V_j^k)). \tag{137}$$

We then bound the term $\gamma\lambda\sum_{j=\chi(i)}^{i-1}(1-\lambda)^{i-j-1}(\bar{\sigma}(V^*) - \sigma_{j-1}^k(V^*))$. Note that

$$\bar{\sigma}(V^*) - \sigma_{j-1}^k(V^*)$$
$$= \bar{\sigma}(V^*) - \sigma^k(V^*) + \sigma^k(V^*) - \sigma_{j-1}^k(V^*)$$
$$\leq \frac{\Gamma}{1-\gamma} + P^kV^* - P_{j-1}^kV^*$$
$$\leq \frac{\Gamma}{1-\gamma} + \frac{1}{1-\gamma}\sqrt{\log\frac{SAKT}{\delta}}, \tag{138}$$

where the last inequality due to Hoeffding's inequality.

For any state-action pair $(s, a)$,

$$|(1-\lambda)^{i-\chi(i)}\Delta_{\chi(i)}(s, a)| \leq (1-\lambda)^{i-\chi(i)}\|\Delta_{\chi(i)}\|. \tag{139}$$

In addition, we have

$$\left\|\gamma\lambda\sum_{j=\chi(i)}^{i-1}(1-\lambda)^{i-j-1}(\sigma_{j-1}^k(V^*) - \sigma_{j-1}^k(V_j^k))\right\| \leq \gamma\lambda\sum_{j=\chi(i)}^{i-1}(1-\lambda)^{i-j-1}\|\Delta_j^k\|. \tag{140}$$

Combining the bounds in (139), (138), and (140), we get

$$\|\Delta_i^k\|$$
$$\leq (1-\lambda)^{i-\chi(i)}\|\Delta_{\chi(i)}\| + \frac{\gamma}{1-\gamma}\lambda(E-1)\Gamma + \gamma\lambda\sum_{j=\chi(i)}^{i-1}(1-\lambda)^{i-j-1}\|\Delta_j^k\|$$
$$+ \frac{\gamma}{1-\gamma}\sqrt{\lambda\log\frac{SAKT}{\delta}}. \tag{141}$$

The remaining proofs similarly follows the one for Lemma D.3. $\qquad\square$

**Theorem G.6** (Convergence). *Choose $E - 1 \leq \frac{1-\gamma}{4\gamma\lambda}$ and $\lambda \leq \frac{1}{E}$. It holds that*

$$\|\Delta_T\| \leq \frac{4}{(1-\gamma)^2}\exp\{-\frac{1}{2}\sqrt{(1-\gamma)\lambda T}\} + \frac{2\gamma^2}{(1-\gamma)^2}(6\lambda^2(E-1)^2 + \lambda(E-1))\Gamma.$$

*Proof.* By Lemma G.2,

$$\|\Delta_{t+1}\| \le (1-\lambda)^{t+1}\|\Delta_0\| + \gamma\lambda\|\sum_{i=0}^{t}(1-\lambda)^{t-i}\frac{1}{K}\sum_{k=1}^{K}(\sigma^k(V^*) - \sigma^k(V_i^k))\|$$

$$+ \gamma\lambda\|\sum_{i=0}^{t}(1-\lambda)^{t-i}\frac{1}{K}\sum_{k=1}^{K}(P^k V_i^k - P_i^k V_i^k)\|.$$

Since $0 \le Q_0(s,a) \le \frac{1}{1-\gamma}$, the first term can be bounded as

$$(1-\lambda)^{t+1}\|\Delta_0\| \le (1-\lambda)^{t+1}\frac{1}{1-\gamma}. \tag{142}$$

To bound the term $\|\gamma\lambda\sum_{i=0}^{t}(1-\lambda)^{t-i}\frac{1}{K}\sum_{k=1}^{K}(\sigma^k(V^*) - \sigma^k(V_i^k))\|$, we divide the summation into two parts as follows. For any $\beta E \le t \le T$, we have

$$\sum_{i=0}^{t}(1-\lambda)^{t-i}\lambda\gamma\|\frac{1}{K}\sum_{k=1}^{K}(\sigma^k(V^*) - \sigma^k(V_i^k))\|$$

$$= \sum_{i=0}^{\chi(t)-\beta E}(1-\lambda)^{t-i}\lambda\gamma\|\frac{1}{K}\sum_{k=1}^{K}\sigma^k(V^*) - \sigma^k(V_i^k))\|$$

$$+ \sum_{i=\chi(t)-\beta E+1}^{t}(1-\lambda)^{t-i}\lambda\gamma\|\frac{1}{K}\sum_{k=1}^{K}\sigma^k(V^*) - \sigma^k(V_i^k))\|$$

$$\le \frac{\gamma}{1-\gamma}(1-\lambda)^{t-\chi(t)+\beta E} + \sum_{i=\chi(t)-\beta E+1}^{t}(1-\lambda)^{t-i}\lambda\gamma\|\frac{1}{K}\sum_{k=1}^{K}\sigma^k(V^*) - \sigma^k(V_i^k))\|.$$

By Lemma G.4,

$$\|\frac{1}{K}\sum_{k=1}^{K}(\sigma^k(V^*) - \sigma^k(V_t^k))\|$$

$$\le \|\Delta_{\chi(t)}\| + 2\lambda\frac{1}{K}\sum_{k=1}^{K}\sum_{t'=\chi(t)}^{t-1}\|\Delta_{t'}^k\| + \gamma\lambda\frac{1}{K}\sum_{k=1}^{K}\max_{s,a}|\sum_{t'=\chi(t)}^{t-1}(\sigma_{s,a}^k)_{t'}(V^*) - \bar{\sigma}_{s,a}(V^*)|,$$

hence

$$\sum_{i=\chi(t)-\beta E+1}^{t}(1-\lambda)^{t-i}\lambda\gamma\|\frac{1}{K}\sum_{k=1}^{K}\sigma^k(V^*) - \sigma^k(V_i^k))\|$$

$$\le \sum_{i=\chi(t)-\beta E+1}^{t}(1-\lambda)^{t-i}\lambda\gamma\Big(\|\Delta_{\chi(i)}\| + 2\lambda\frac{1}{K}\sum_{k=1}^{K}\sum_{t'=\chi(t)}^{t-1}\|\Delta_{t'}^k\|$$

$$+ \gamma\lambda\frac{1}{K}\sum_{k=1}^{K}\max_{s,a}|\sum_{j=\chi(i)}^{i-1}(\sigma_{s,a}^k)_j(V^*) - \bar{\sigma}_{s,a}(V^*)|\Big)$$

$$\le \sum_{i=\chi(t)-\beta E+1}^{t}(1-\lambda)^{t-i}\lambda\gamma\Big(\|\Delta_{\chi(t)}\| + 2\lambda\frac{1}{K}\sum_{k=1}^{K}\sum_{t'=\chi(t)}^{t-1}\|\Delta_{t'}^k\| + \gamma\lambda\Gamma\frac{E-1}{1-\gamma} + \frac{1}{1-\gamma}\sqrt{(E-1)\log\frac{SAKT}{\delta}}\Big).$$

By Lemma G.5, we have that

$$\sum_{i=\chi(t)-\beta E+1}^{t} (1-\lambda)^{t-i}\lambda\gamma 2\lambda \frac{1}{K}\sum_{k=1}^{K}\sum_{j=\chi(i)}^{i-1}\|\Delta_j^k\|$$

$$\leq 2\lambda^2\gamma \sum_{i=\chi(t)-\beta E+1}^{t}(1-\lambda)^{t-i}\frac{1}{K}\sum_{k=1}^{K}\sum_{j=\chi(i)}^{i-1}\left(\|\Delta_{\chi(j)}\| + \frac{3\gamma}{1-\gamma}\lambda(E-1)\Gamma + \frac{6\gamma}{1-\gamma}\sqrt{\lambda\log\frac{SATK}{\delta}}\right)$$

$$\leq 2\lambda\gamma(E-1)\max_{\chi(t)-\beta E\leq i\leq t}\|\Delta_{\chi(i)}\| + \frac{6\gamma^2\lambda^2}{1-\gamma}(E-1)^2\Gamma + \frac{12\lambda}{1-\gamma}(E-1)\sqrt{\lambda\log\frac{SATK}{\delta}}.$$

Similarly, we get the following recursion holds for all rounds $T$:

$$\|\Delta_{t+1}\|$$

$$\leq \gamma(1+2\lambda(E-1))\max_{\chi(t)-\beta E\leq i\leq t}\|\Delta_{\chi(i)}\| + \frac{2}{1-\gamma}(1-\lambda)^{\beta E} + \frac{\gamma^2}{1-\gamma}(6\lambda^2(E-1)^2 + \lambda(E-1))\Gamma$$

$$+ C\left(\frac{\lambda\sqrt{E-1}}{1-\gamma} + \frac{\lambda^{1.5}(E-1)}{1-\gamma} + \frac{1}{1-\gamma}\sqrt{\frac{\lambda}{K}}\right)\sqrt{\log\frac{SATK}{\delta}}.$$

Similarly, choosing $\beta = \lfloor\frac{1}{E}\sqrt{\frac{(1-\gamma)T}{2\lambda}}\rfloor$, $L = \lceil\sqrt{\frac{\lambda T}{1-\gamma}}\rceil$, $t+1 = T$, we get

$$\|\Delta_T\|$$

$$\leq \frac{4}{(1-\gamma)^2}\exp\{-\frac{1}{2}\sqrt{(1-\gamma)\lambda T}\} + \frac{2\gamma^2}{(1-\gamma)^2}(6\lambda^2(E-1)^2 + \lambda(E-1))\Gamma$$

$$+ C'\left(\frac{(E-1)\lambda^{1.5}}{(1-\gamma)^2} + \frac{\sqrt{E-1}\lambda}{(1-\gamma)^2} + \frac{\sqrt{\lambda}}{(1-\gamma)^2\sqrt{K}}\right)\sqrt{\log\frac{SATK}{\delta}}.$$

$\square$

**Corollary G.7.** *Choose $(E-1) \leq \min\frac{1}{\lambda}\{\frac{\gamma}{1-\gamma}, \frac{1}{K}\}$, and $\lambda = \frac{4\log^2(TK)}{T(1-\gamma)}$. Let $T \geq E$. Then with probability at least $1-\delta$ it holds that*

$$\|\Delta_T\| \leq \frac{C_1}{(1-\gamma)^2 TK} + \frac{C_2\log^2(TK)}{(1-\gamma)^3}\frac{E-1}{T}\Gamma + \frac{C_3}{(1-\gamma)^3}\frac{\log(TK)}{\sqrt{TK}}\sqrt{\log\frac{SATK}{\delta}}.$$

