# OpenReview forum: "Pessimism Principle Can Be Effective: Towards a Framework for Zero-Shot Transfer Reinforcement Learning"
_ICML.cc/2025/Conference — ICML 2025 poster_

### Official Review · Reviewer_QZdX · 2025-03-11

**Overall Recommendation:** 3

**Summary:**

This paper studies transfer learning where one aims to learn a good policy for a target domain with data collected from multiple source domains. And they consider the distributed and decentralized setting, where one central server can only access partial data from source domains. The authors apply the principle of pessimism in this problem. Specifically, their algorithm estimates a pessimistic evaluator which lowers bounds the performance of the policy in the target domain by utilizing an average of robust Bellman operators. With this conservative estimator, they can provide performance guarantees of their learned policy. What's more, they provide a minimal pessimism operator with which they can mitigate the issue of negative learning which previous methods may suffer in transfer learning.

**Claims And Evidence:**

One of the claims seems not clearly supported.

The authors made the following claims:

They introduce pessimism into transfer learning and based on that principle, they can construct the performance estimator of the learned policy which is a lower bound of the performance in the target domain. With this conservative estimator, they solve two problems existing in transfer learning, first, there is no performance guarantee of the learned policy in the target domain, second, the existing of source domains which are very distinct from the target domain may lead to negative transfer learning.

The claim on avoiding negative transfer is not supported clearly. They do show their algorithm can treat source domains differently, but I am expecting a theorem that can quantify how much performance gained or an example, that can show by treating all source domains equally, we suffer negative transfer but with the proposed algorithm, it is avoided.

**Essential References Not Discussed:**

I don't see essential related works that are not discussed. The discussion on robust RL which severs the base stone in this paper is placed in Appendix.

**Experimental Designs Or Analyses:**

No, I cannot evaluate the soundness of the experimental designs. The authors implemented experiments on two tasks with different target domain uncertainty parameters and compared their algorithms with two baselines which are non-robust algorithms.

**Methods And Evaluation Criteria:**

Yes, it makes sense. The method proposed in this paper primarily builds on the idea of robust RL and q-learning. With robust RL, the authors build pessimistic performance estimator of the learned policy in the target domain which can lower bound the true performance. And with Q-learning, they propose practical algorithms to learn that pessimistic estimator.

**Other Comments Or Suggestions:**

1. The authors specify a parameter $E$ in their algorithm which trades off the convergence rate and communication cost, it would be interesting to see the experimental results on varying $E$.

**Other Strengths And Weaknesses:**

Strengths:
1. The presentation is very clear which makes the paper easy to follow.

Weaknesses:
1. The results seem build on the assumption that the reward function is identical across all domains including source domains and the target domain. It restricts the scenarios where the algorithm is applicable. Correct me if my understanding is wrong.
2. The authors comment in the main paper that their results can be directly extended to the case with different similarities parameter in source domains (which I agree), but it would be better to at least mention and supplement it in Appendix which I didn't find.
3. The results are built on the tabular settings because of the average step of Q-values from source domains. Then the claim in line 384 on the scalability should be more conservative, though the authors specify the local update step (not global step) can be implemented by any scalable robust RL algorithm.

**Questions For Authors:**

1. Is it true that the rewards of all source domains and the target domain are assumed to be same? If it is true, is there any way to get rid of it?
2. I can see the advantage of the proposed algorithms in the distributed setting since only Q-values are communicated. What if the communication of any raw data between these source domains is safe and possible, do the proposed algorithms still outweigh those estimators defined in line 220-224?
3. In Theorem 5.6 and 6.4, when $E=1$, on the right hand side, the similarity parameter $\Gamma$ does not affect the convergence rate, why communicating at each round can get rid of the $\Gamma$, it sounds unreasonable to me, is there any explanation?
4. According to Theorem 5.6, the final $Q_{AO}$ is approaching $\frac{\sum_{k=1}^KQ _k}{K}$, why doesn't one directly learn a good $Q_k$ in each source domain since we have lots of data in each domain, and then take the average instead of doing it in an iterative way?

**Relation To Broader Scientific Literature:**

The contributions in this paper are related to two main areas which are offline RL and robust RL.

First, the key contribution is the application of the principle of pessimism to the problem of transfer learning which is related to offline RL where pessimism is widely used to provide high-probability lower bound estimator of Q-functions. In offline RL, the estimator is constructed using some concentration inequality.

In this paper, the authors construct the lower bound estimator via robust RL. They assume an uncertainty model set for each source domain where the target domain lies in. By finding the optimal policy in this uncertainty, they can lower bound the performance in the target domain.

Another broad related area is Q-learning. The algorithm proposed in this paper is a modified version of Q-learning in the framework of distributed transfer learning.

**Theoretical Claims:**

I checked the correctness of Lemma 4.1, Lemma 5.1, Theorem 5.2 and Theorem 5.5. They seem correct to me.

---

> ### Author Rebuttal · Authors · 2025-04-01
>
> We sincerely thank the reviewer for the time and feedback. Please refer to the link https://anonymous.4open.science/r/ICML-2663/README.md for our newly conducted experiments.
>
> **Negative transfer**
> Our initial environments were relatively simple, so negative transfer was not evident. To support our claims, we ran a new Cartpole experiment (Fig 1). The source domains are with randomly generated parameters (length of pole), and the target is the default one. As shown, MDTL-Avg suffers from negative transfer, while MDTL-Max successfully mitigates it. We will elaborate further in the final version.
>
> **Experiments on E and robust baselines**
> We validate the trade-off in Fig. 2 and Table 1. As E increases, convergence slows (under fixed step size) and performance declines. However, computational time also decreases, indicating reduced communication.
>
> We also add two robust baselines (see Table 2 and reply to R rfmi).
>
> **Identical reward assumption**
> We made this assumption to simplify presentation and analysis, but our framework can extend to settings with differing rewards.
>
> 1. If the (unknown) target domain reward differs from the sources, additional knowledge on its relation to the source reward should be known, e.g., $|r_t(s,a)-r_s(s,a)|\leq R(s,a)$, otherwise the target may have different goals with source domains, making transfer impossible. With such knowledge, we additionally define a **reward** uncertainty set for source: $\mathcal{R}^k_{s,a}=\{r': |r'-r(s,a)|\leq R(s,a) \}$ and modify the robust Bellman operator to $\mathbf{T}^k(Q)=(r-R)+\sigma(V)$, preserving pessimism and effectiveness.
>
> 2. If source rewards also differ from each, we can still apply different local operator $\mathbf{T}^k(Q)=(r^k-R^k)+\sigma(V)$ to similarly ensure conservative.
>
> We will discuss more in the final version.
>
> **Extended to the case with different similarities parameter in source domains**
> We thank the reviewer for this insightful point. While most proofs are independent of the radius, some technical adjustments are needed. For example, in extending Thm. 6.4, we can adjust (89) to yield a bound $\gamma\lambda\frac{E-1}{1-\gamma}\max_i\{\Gamma_i\}$. We will discuss this further in the final version.
>
> **Claim on the scalability**
> Thank you for this valuable comment. We will revise the claim. By scalability, we refer to the local updates, which can leverage scalable, model-free methods. While global aggregation is necessary in tabular Q-learning, our approach is not limited to this setting. Our method can be paired with function approximation or policy gradient approaches to improve scalability. For instance:
> 1. With function approximation for robust RL [1], or
> 2. Robust policy gradient [2], the global step only requires parameter aggregation (e.g., Jin et al., 2022), making it scalable in practice.
> We further numerically verify this by develop experiments under Cartpole and DVRP (please see our response to #tyX8). We will revise the statement and leave further extensions for future work.
>
> [1] Zhou, R. et al., Natural actor-critic for robust reinforcement learning with function approximation, 2023
> [2] Wang, Y. & Zou, S., Policy gradient method for robust reinforcement learning, 2022
>
>
> **Proxies in Line 220**
> Our proxies outperform options (1)-(3) due to reduced conservativeness. Though proxy (4) is less conservative, it presents practical challenges:
> 1. Its uncertainty set lacks a ball structure, making standard robust RL methods inapplicable.
> 2. Intersection-based methods require accurate knowledge of each local model and set, which is not available via raw data communication alone.
> Thus, our method has two advantages over (4): lower communication requirements and easier implementation.
>
> **E=1 in Theorem 5.6/6.4**
> When E = 1, the global Q-table is updated every step using $Q \leftarrow \frac{1}{K} \sum_k \mathbf{T}^k(V)$. This operator is a $\gamma$-contraction, so convergence rate is independent of $\Gamma$. Even in the presence of unbiased noise, the expected update preserves this rate. Technically, Eq. (34) bounds $\|\sigma(V) - \sigma(V')\| \leq \|V - V'\| \leq \frac{1}{1 - \gamma}$, independent of $\Gamma$. However, $\Gamma$ still affects the pessimism level $\zeta$, which impacts the **effectiveness**, though not the **convergence rate**.
>
> When E > 1, agents perform multiple local updates with their own robust operators. The global update then reflects local differences, making it depend on $\Gamma$. A similar effect is observed in Wang et al., 2024 (Theorem 1), where heterogeneity does not affect convergence when  E = 1.
>
> **Learn locally and average**
> We apologize for the possible misunderstandings here. $Q_{AO}$ is the intended solution, and $Q_k$ is the algorithm’s output. The reviewer may have confused $Q_k$ with $Q^*\_k$, the robust value function for local domain k. Importantly, $Q_{AO}$ is different from $\frac{\sum Q^*\_k}{K}$, so simply learning locally and averaging does not yield the correct solution.

---

> > ### Comment · Reviewer_QZdX · 2025-04-04
> >
> > Thank the authors for their efforts on the rebuttal, especially for providing additional experimental results. The elaborations are detailed. Some of my concerns are solved including different reward functions, the situation of $E=1$. Some remaining concerns or questions are:
> >
> > 1. You explained that $Q_k$ in Theorem 5.6 is the output of the algorithm instead of $Q_k^*$. I am still confused here since Algorithm 1 outputs $Q_k$ in each round $t$, right? Then what value does $Q_{AO}$ converge to by $||E[Q_{AO}-\frac{\sum_{k=1}^K Q_k}{K}]||$ in Theorem 5.6. I may misunderstand something here. This is important for me to evaluate the results, especially the claim on negative learning, since Theorem 5.6 and 6.4 are almost same except the term within the expectation.
> >
> > 2. Thank the authors for providing additional results, however, the experiments of negative learning are not that evident if we look at the peak performance. Since the mitigation of the negative learning is one of the contributions, I am expecting more convincing results. Also can the authors explain more on Theorem 6.4 comparing with Theorem 5.6 that what value do MDTL-MAX and MDTL-AVG converge to and how MDTL-MAX converge to a better value? This may be closely related to my first question.
> >
> > 3. Please revise the claim of scalability to make it precise. It would not negatively affect my evaluation of this work.
> >
> > 4. Does the algorithm need any prior information about the uncertainty set?
> >
> > I am keeping the score at the current stage but I am staying neural and willing to increase the score based on the further explanation to my above questions and discussions with other reviewers.

---

> > > ### Author Response · Authors · 2025-04-04
> > >
> > > We sincerely appreciate your response.
> > >
> > > 1. We apologize for the notational confusion. In Theorem 5.6, $ Q_{AO} $ denotes the fixed point of the average-based operator defined in eq(6). It is a static value and is not updated during training. Our algorithm is designed to estimate it and derive policy from it. At time step $t$, each local agent produces a local Q-table $Q_t^k$ ($Q_k$ in our previous notation), and the algorithm will take their aggregation $\bar{Q}\_t := \frac{1}{K} \sum\_{k=1}^K Q\_t^k$ as the output. Our convergence result measures how well this aggregated Q-table approximates $Q_{AO}$:
> > > $|| \mathbb{E} [ \bar{Q}\_t- Q_{AO} ]|| \to 0.$ $Q^*_k$ is the robust value function of each local source domain, and is different from the above notations and does not occur in convergence.
> > >
> > > Similarly, in Theorem 6.4, $Q_{MP}$ is the fixed point of the max-based operator, which is the value our algorithm aims to estimate. The global output is then the max-aggregation $\max_k Q_t^k$,
> > > and the convergence is $|| \mathbb{E} [ \max_k Q_t^k - Q_{MP}] || \to 0.$
> > >
> > > Both Theorems 5.6 and 6.4 establish that the algorithm converges to its corresponding fixed point, enabling recovery of the respective transferred policy. These convergence guarantees are orthogonal to the study of negative transfer, which instead arises from comparing of $Q_{MP}$ and $Q_{AO}$ (as elaborated in the next point).
> > >
> > > 2. The mitigation of negative transfer through MDTL-Max is in fact captured by Theorem 6.1 (2).
> > >
> > > We first highlight the fundamental advantage of our pessimism principle TL framework. As established in Section 4.2, the performance in the target domain is monotonic with respect to the conservativeness of the proxy—less conservative proxies yield better transferred policies (see Lemma 4.1).
> > >
> > > We note that $V_{MP}\geq V_{AO}$ by Thm 6.1 (2), indicating that $V_{MP}$ serves as a **less conservative proxy** than $V_{AO}$. Therefore, while the MDTL-Max algorithm (which targets $V_{MP}$) may require additional effort due to max-aggregation, it ultimately leads to improved transfer performance compared to MDTL-Avg (which targets $V_{AO}$).
> > >
> > > Furthermore, Theorem 6.1 (2) shows that $V_{MP}$ also outperforms **any individual local proxy** $ V\_{\mathcal{P}\_k} $. No such guarantee exists for $V_{AO}$. This result is crucial: it implies that MDTL-Max inherently mitigates negative transfer. Because of the monotonicity, MDTL-Max will perform at least as well as the best local source policy, effectively **ignoring misleading information from poorly aligned source domains**.
> > >
> > > To better illustrate this, consider a setting where only one source domain is similar to the target, while the others are significantly different. An ideal transfer method would selectively leverage the similar domain and ignore the rest. However, in practice, we cannot identify the "good" source domain in advance. If we treat all source domains equally—as in MDTL-Avg—this can lead to **negative transfer**, where the poor-quality sources degrade the overall performance. In contrast, MDTL-Max is designed to be **robust to such variation**: it guarantees performance at least as good as the best source, effectively and automatically filtering out the harmful domains and preventing negative transfer.
> > >
> > > We further develop 2 experiments to verify this, please refer to https://anonymous.4open.science/r/ICML-2663/README.md.
> > >
> > > In the Robot Recycle environment (fig 5), we set the target domain to have parameters $\alpha=\beta=0.8$. The source domains are configured as $\alpha_k=\beta_k \approx 0.1$ for $ k = 1,...,K-1$, and $\alpha_K = \beta_K=0.7$. Clearly, domain $K$ is much more similar to the target, and an ideal transfer should primarily rely on it while ignoring the others. However, MDTL-Avg treats all source domains equally, resulting in an overly conservative policy due to influence from dissimilar sources. In contrast, MDTL-Max achieves significantly better performance than all source domains, including domain $K$, thereby avoiding negative transfer.
> > >
> > > We observe similar results in the Frozen-Lake environment under an analogous setup (fig 4). For each algorithm, we run experiments across 10 random seeds. For each seed, the policy is evaluated over 10 independent episodes to compute the average return. Result shows that MDTL-Max consistently outperforms MDTL-Avg and successfully mitigates negative transfer.
> > >
> > > 3. We sincerely appreciate your suggestion and will revise the claim to:
> > > Our algorithms can also be implemented in a model-free manner to enhance computational and memory efficiency, where any model-free algorithm for robust reinforcement learning can be integrated into the local update step.
> > >
> > > 4. The algorithms require knowledge of the uncertainty set radius, which represents our prior knowledge of task similarities. In general, effective transfer cannot be expected without such knowledge. Please also refer to our response to Reviewer rfmi for a more detailed discussion.

---

### Official Review · Reviewer_tyX8 · 2025-03-13

**Overall Recommendation:** 3

**Summary:**

The paper introduces a novel pessimism-based transfer learning framework to address critical challenges in zero-shot transfer RL. The authors propose constructing conservative proxies—via robust Bellman operators and novel aggregation schemes (both averaged and minimal pessimism operators)—that yield lower bounds on the target performance. In addition, they develop distributed algorithms with convergence guarantees.

**Update after rebuttal**--I appreciate the authors’ thorough revision and the additional experiments addressing my concerns. I hope those will be implemented in the revised draft. I have decided to keep my original score.

**Claims And Evidence:**

The paper claims that incorporating a pessimism principle can yield a conservative proxy and the proposed distributed algorithms are computationally efficient and scalable. These are supported by theoretical proofs of contraction properties. However, empirical evidence is deficient to some extent.

**Essential References Not Discussed:**

There are some papers discussing zero-shot transfer in the context of contextual reinforcement learning. I highly recommend the authors to include them in related works. Furthermore, a brief discussion of recent advances in safe RL and risk-sensitive RL (which also aim to avoid over-optimistic policies) could further contextualize the pessimism principle. However, the omission does not detract significantly from the paper’s contributions.

**Experimental Designs Or Analyses:**

The experimental design is sound; however, a discussion on hyperparameter sensitivity and empirical analysis in many different benchmarks could further strengthen the work.

**Methods And Evaluation Criteria:**

This paper will largely benefit from widely used benchmark datasets. One example could be contextual MDP benchmarks.

**Other Comments Or Suggestions:**

- I didn’t find any significant typos.
- Font size in figure is too small.

**Other Strengths And Weaknesses:**

- The paper offers a thorough theoretical justification with detailed proofs. However, some proofs and algorithmic descriptions could benefit from additional intuition or summarizing remarks.
- The distributed algorithm design and privacy-preserving updates are well aligned with modern large-scale and decentralized applications. While simulations are convincing, experiments on real-world tasks would further strengthen the empirical claims.

**Questions For Authors:**

- In Table 2, it’s hard to see difference between MDTL-Avg and MDTL-Max. Please provide other examples that could distinguish these two.
- Actually, more experiments should be done to support this.
- I’m curious how critical unbiased estimation assumption is and how robust is the method to the assumption.
- I’m curious about the comparison with other methods that authors mentioned throughout the papers. How does this approach perform better than others both in your theory and  experiments?

**Relation To Broader Scientific Literature:**

The paper is well situated within the broader literature:
- It builds on established ideas in robust RL, domain randomization, and multi-task learning.
- It addresses limitations in existing transfer RL methods (no guarantee in safety and performance when transferred) by providing performance guarantees and methods to avoid negative transfer.

**Theoretical Claims:**

I haven’t checked all the details in proof, but the theoretical contributions are both novel and rigorous. Some proofs could benefit from additional intuition or summary comments to aid understanding.

---

> ### Author Rebuttal · Authors · 2025-04-01
>
> We sincerely thank you for your time and feedback, and appreciate the reviewer identifying our contribution. Please refer to the link https://anonymous.4open.science/r/ICML-2663/README.md for our newly conducted experiments.
>
> **Hyperparameter sensitivity; Benchmark like contextual MDPs.**
> We appreciate the reviewer’s suggestion.
>
> Two hyperparameters are included: E and $\lambda$.
> 1. We implement algorithms with different E in Fig 2, Table 1. A larger E results in slower convergence but less computation, matching our theoretical results.
> 2. We run experiments with $\lambda=0.2$ in Table 2. Together with results in our paper for 0.1, it shows our methods are uniformly better and robust to it.
>
> We present 2 more experiments. (1). Cartpole Experiment (Fig. 3): We apply our methods to the Cartpole environment, a common example of a contextual MDP. Results show that even when updates introduce bias, our methods outperform the baseline, confirming robustness. (2). Dynamic Vehicle Routing Problem (DVRP): We design a real-world scenario—DVRP [Iklassov, Z, et al., Reinforcement Learning for Solving Stochastic Vehicle Routing Problem, 2024]—which extends the classic vehicle routing problem with real-time requests and environmental uncertainty. Three important objectives are considered: Minimizing overall routing cost, Enhancing route smoothness, and Improving route stability. As shown in Table 3, our methods consistently outperform baselines across all criteria, and are more effective.
>
> We will add more experiments in the final version.
>
> **Additional references.**
> We thank the reviewer for recognizing our contributions and providing constructive suggestions.
>
> We briefly discussed contextual RL (cRL) for zero-shot generalization in Appendix A. cRL seeks to optimize performance under a contextual distribution to enable transfer without fine-tuning. However, theoretical guarantees remain limited. For instance, [1] provides a suboptimality bound, but it requires samples from the contextual distribution, potentially including target domain data. Furthermore, it remains unclear whether cRL can mitigate negative transfer, especially when the context distribution is assumed to be uniform.
> We also appreciate the reviewer’s mention of safe and risk-sensitive RL, which aim to optimize reward while maintaining low cost under specific criteria [2]. However, most of these works are developed for single-environment settings, and do not address transfer scenarios. We will incorporate a more thorough discussion in the final version.
>
> [1] Wang, Z., et al., Towards Zero-Shot Generalization in Offline Reinforcement Learning, 2025
> [2] García, J., et al., A Comprehensive Survey on Safe Reinforcement Learning, 2015
>
> **Intuition or summarizing remarks.**
> Thank you for the suggestion. Our algorithm consists of two main parts: local updates and global aggregation. Each agent independently updates its local Q-table using its data, and after every E steps, a global aggregation step unifies the local Q-tables. Our convergence proof mirrors this structure: it is decomposed into a local part (governed by the local Bellman operator) and a global part (governed by the aggregation mechanism). We will include further discussion in the final version.
>
> **Unbiased Estimation Assumption.**
> The unbiased estimation assumption is introduced to simplify convergence analysis. However, our convergence results can be extended to settings with controlled bias, such as the bias introduced by max aggregation. This bias can be mitigated using techniques like threshold-MLMC [1].
> Importantly, even in the presence of bias, if the expected proxy remains pessimistic, our transfer learning framework and theoretical guarantees still hold. We thus argue that our method is robust to such biases.
> To support this, we include an experiment on Cartpole (Fig. 3) analyzing the impact of bias. Results show that our proxy remains conservative, and the pessimism-based framework continues to outperform the baseline, confirming robustness.
>
> [1] Wang, Y., et al., Model-Free Robust Reinforcement Learning with Sample Complexity Analysis, 2024
>
> **Comparisons with baselines.**
> As noted in the paper, our primary advantage lies in theoretical guarantees: we optimize a lower bound on the target domain’s performance, avoiding overly optimistic decisions that may fail under domain shift. In contrast, domain randomization (DR) lacks such guarantees and can be overly optimistic, especially when the randomized training environments fail to capture true uncertainty.
>
> We acknowledge that when the uncertainty set is constructed using inaccurate prior knowledge, our method may become overly conservative, potentially leading to suboptimal performance compared to DR. Nonetheless, the ability to guarantee robust performance across all scenarios is a central contribution—especially important for safety-critical or high-stakes applications, where conservativeness is not only expected but necessary.

---

### Official Review · Reviewer_QBMR · 2025-03-13

**Overall Recommendation:** 3

**Summary:**

This paper studies zero-shot transfer reinforcement learning. The authors incorporate a pessimism principle into transfer learning to serve as a lower bound to conservatively estimate the target domain’s performance. The authors propose and analyze two types of conservative estimates, rigorously characterizing their effectiveness, and develop distributed, convergent algorithms to optimize them.

**Claims And Evidence:**

Yes.

**Essential References Not Discussed:**

No.

**Experimental Designs Or Analyses:**

No.

**Methods And Evaluation Criteria:**

Yes.

**Other Comments Or Suggestions:**

See the weaknesses above.

**Other Strengths And Weaknesses:**

Strengths:

1. This paper proposes a new framework based on the pessimism principle, which constructs and optimizes a conservative estimation of the target domain’s performance.
2. The paper is well-written and easy-to-follow.
3. The authors provide some upper bounds for the performance of their proposed algorithms. They also provide experimental results to support the theoretical findings.

Weaknesses:
1. The main concern is that the suboptimality gap is indeed dependent on the $\max_{\pi}\zeta^\pi $. Though optimizing the lower bound is a good solution and the authors provide the convergence results to the lower bounds, the gap $\max_{\pi}\zeta^\pi $ seems not controllable, meaning that there is no theoretical upper bound on the actual suboptimality gap.
2. The section 2.1 can not formally define the problem setting. It would be better to specify the setting and the assumptions needed explicitly in this section.
3. Theorem 7.2. seems to be too abrupt to appear, the $\Delta_T$ is used without definition.
4. The proposed methods borrow some ideas from federated learning, it would be better to briefly introduce some related work in federated learning in the Related Work section (in appendix A. Additional Related Works).

**Questions For Authors:**

1. Regarding my concern in the weaknesses 1. above, could you please give some explanations?
2. In Proposition 5.4., you prove the results under total variation or Wasserstein distance, do similar results hold under other metrics?

**Relation To Broader Scientific Literature:**

This paper is related to transfer learning and federated learning. The authors propose to optimize a lower bound of the objective in the transfer reinforcement learning, and utilize some ideas from the federated learning literature.

**Theoretical Claims:**

No.

---

> ### Author Rebuttal · Authors · 2025-04-01
>
> We sincerely thank you for your time and feedback, and appreciate the reviewer identifying our contribution. Please refer to the link https://anonymous.4open.science/r/ICML-2663/README.md for our newly conducted experiments.
>
> **Upper bound on $\max_\pi\zeta^\pi$.**
>
> We first clarify that $\|\zeta\|$ has an upper bound and can be controlled via the construction of the uncertainty sets. With our proxies, it holds that $\|\zeta\|\leq \mathcal{O}((1-\gamma)^{-2}\Gamma)$. Moreover, when adapting our Max proxy with different radii $\Gamma_i$, it holds that $\|\zeta\|\leq \mathcal{O}((1-\gamma)^{-2}\min\{\Gamma_i\})$. Hence the suboptimality gap can be upper bounded by the domain similarities, which is reasonable and subject to the nature of the problem. This dependence also motivates one of the potential extensions of our framework -- a safe-to-improve principle: once a conservative proxy is constructed, any improvement to it directly improves target performance, without the loss of theoretical performance guarantees. For instance, if additional information is available (e.g., a small set of target domain data or allowance of explorations), we can further reduce radii $\Gamma$ to ensure a better transfer performance.
>
> **Formally define the problem setting. It would be better to specify the setting and the assumptions needed explicitly in this section.**
>
> Our formulation is as follows. Consider the target domain $\mathcal{M}\_0 = (\mathcal{S},\mathcal{A},P\_0,r,\gamma)$ with the unknown target kernel $P_0$. We have no data from it, but instead have access to multiple source domains $\mathcal{M}\_k = (\mathcal{S},\mathcal{A},P\_k,r,\gamma)$. Our goal in zero-shot multi-domain transfer learning is to optimize the performance $V_{P_0}^\pi$ under $\mathcal{M}_0$. For simplicity, we consider the case of identical reward (please see our response to Reviewer QZdx for extensions).
>
> The only assumption is that, there exists an upper bound $\Gamma\geq D(P_0||P_k)$, that is known by the learner. This assumption is reasonable, as such information is essential to obtain performance guarantees. Even if in the worst-case, we can set $\Gamma=1$ (in total variation) to construct an (overly) conservative proxy, which yet still avoids decisions with severe consequences in the target domain, preferred in transfer learning settings.
>
>
> **Theorem 7.2. seems to be too abrupt to appear, the $\Delta_T$ is used without definition.**
>
> We apologize for missing the definition of $\Delta_T$. It is the difference between the algorithm output and the fixed points of the aggregated operators: $\Delta_T := Q_{\rm AO} - Q^T$ for MDTL-Avg, and $\Delta_T := Q_{\rm MP} - Q^T$ for MDTL-Max. We will clarify these.
>
>
> **The proposed methods borrow some ideas from federated learning, it would be better to briefly introduce some related work in federated learning in the Related Work section (in appendix A. Additional Related Works).**
>
> We thank the reviewer for the helpful suggestion. We will include related work on federated reinforcement learning (FRL). We note, however, that directly extending FRL to our setting poses non-trivial challenges: (1) FRL has linear update rules (non-robust operators), while ours involve non-linear robust operators; (2) FRL focuses on optimizing average performance across local environments, whereas we address the more challenging max-based multi-domain transfer; and (3) more importantly, FRL is not designed for transfer learning, while our pessimism principle enables transfer with theoretical guarantees.
>
>
>
> **In Proposition 5.4., you prove the results under total variation or Wasserstein distance, do similar results hold under other metrics?**
>
> According to our proof, the result holds as long as $\sigma\_{\bar{\mathcal{P}}}(V) - \frac{1}{K}\sum_{k=1}^K\sigma\_{\mathcal{P}\_k}(V) \leq 0$. We note that additional for $l_p$-norm, the support function has a duality (Clavier et al., 2024. Near-Optimal Distributionally Robust Reinforcement Learning with General $L_p$ Norms.): $\sigma(V)=\max_{\alpha} ( PV\_{\alpha}+f(\Gamma,V\_{\alpha}))$ for some function $f$. Thus $\sigma\_{\bar{\mathcal{P}}}(V) - \frac{1}{K}\sum_{k=1}^K\sigma\_{\mathcal{P}\_k}(V)=\max\_{\alpha} ( \bar{P}V\_{\alpha}+f(\Gamma,V\_{\alpha}))-\frac{1}{K}\sum_i \max_{\alpha} ( P_iV\_{\alpha}+f(\Gamma,V\_{\alpha}) )\leq 0$. It is our future interest to extend to other uncertainty sets.

---

### Official Review · Reviewer_rfmi · 2025-03-14

**Overall Recommendation:** 4

**Summary:**

The paper introduces a novel framework for zero-shot transfer reinforcement learning (RL) based on the pessimism principle. The key idea is to construct a conservative proxy for the target domain's performance, ensuring that the transferred policy achieves a robust lower bound on performance while avoiding negative transfer. The authors propose two types of conservative proxies:
(1) an averaged operator-based proxy and (2) a minimal pessimism proxy. They develop distributed algorithms for optimizing these proxies, with convergence guarantees. The framework is shown to be robust against model uncertainties and scalable to large-scale problems. Experiments on the recycling robot and HPC cluster management problems demonstrate that the proposed methods outperform non-robust baselines, especially in scenarios with domain shifts and model uncertainty.

**Claims And Evidence:**

- **Claim 1**: The pessimism principle ensures a robust lower bound on target domain performance and avoids negative transfer.
  - **Evidence**: Theoretical analysis (**Lemma 4.1**) shows that the sub-optimality gap depends on the level of pessimism\( ||\zeta|| \), and experiments consistently outperform non-robust DR baselines (Figure 3).
- **Claim 2**: The proposed algorithms are efficient and scalable., enabling decentralized learning across multiple source domains (Partially supported).
  - Theorems 5.5 & 6.4: Provide convergence guarantees for both MDTL-Avg (Averaged Proxy) and MDTL-Max (Minimal Pessimism Proxy).
  - Algorithm 1 is designed for distributed training, ensuring privacy-preserving updates.
  - **Concern**: The multi-level Monte Carlo (MLMC) method is computationally expensive, which may limit scalability in high-dimensional RL tasks.
- **Claim 3**: The framework is robust to model uncertainty.
  - Proposition 7.1 suggests that pessimistic proxies provide robustness against perturbations in the target domain, implying distributional robustness.
  - **Concern**: No explicit robustness tests (e.g., varying noise levels, adversarial perturbations) are conducted.

**Essential References Not Discussed:**

1. **Adversarial Robust RL:**
   - Rajeswaran et al., 2017. "EPOpt: Learning Robust Neural Network Policies Using Model Ensembles."
   - Tessler et al., 2019. "Action Robust Reinforcement Learning."

2. **Distributionally Robust RL:**
   - Wiesemann et al., 2014. "Distributionally Robust Convex Optimization."
   - Panaganti & Kalathil, 2022. "Sample Complexity of Robust RL with a Generative Model."

**Experimental Designs Or Analyses:**

The experiments are conducted on two specific problems:
- **Recycling Robot Problem**: A classic RL problem where a robot with a rechargeable battery must decide whether to search for cans or wait for someone to bring them. This is a relatively simple environment with discrete states and actions.
- **HPC Cluster Management Problem**: A task where a cluster manager must decide whether to allocate incoming tasks immediately or enqueue them for later processing. This is also a discrete environment.

### **Strengths**
- Well-structured comparison against proximal domain randomization (DR) baselines.
- Clear ablation study on uncertainty levels (Table 1, Figure 2).

### **Concerns**
- No explicit negative transfer tests: Does not include cases where misleading source domains degrade performance.

**Methods And Evaluation Criteria:**

The authors propose two robust transfer proxies:
- Averaged Operator-Based Proxy (AO): Uses a weighted combination of source domains' robust value functions to construct a conservative but smooth estimate.
- Minimal Pessimism Proxy (MP): Uses a max-aggregation approach to prioritize similar domains while maintaining pessimistic guarantees.
The algorithms are federated-style, designed for distributed policy learning without direct data sharing, making them privacy-friendly and applicable in decentralized systems.

The experiments are conducted on two specific problems:
- **Recycling Robot Problem**: A classic RL problem where a robot with a rechargeable battery must decide whether to search for cans or wait for someone to bring them. This is a relatively simple environment with discrete states and actions.
- **HPC Cluster Management Problem**: A task where a cluster manager must decide whether to allocate incoming tasks immediately or enqueue them for later processing. This is also a discrete environment.

The evaluation focuses on performance in the target domain and robustness to model uncertainty.

The paper compares against domain randomization (DR) methods, which is a reasonable baseline for zero-shot transfer RL. However, it ignores other relevant robust RL approaches, such as:
- Adversarial robust RL (e.g., worst-case minimax RL), which directly optimizes for robustness against worst-case scenarios.
- Distributionally robust RL (DRO) approaches, which handle uncertainty by optimizing policies under distributional shifts.

**Other Comments Or Suggestions:**

The paper covers the most relevant experiment but could benefit from performing negative transfer experiments to validate the robustness of pessimistic aggregation.

**Other Strengths And Weaknesses:**

**Strengths:**
- Novel use of pessimism in zero-shot transfer RL, the pessimism-based approach is well-motivated
- Strong theoretical contributions with formal performance guarantees.
- Federated-style implementation ensures privacy in multi-domain learning.

**Weaknesses:**
- Computational cost of MLMC aggregation not addressed
- Assumes Prior Knowledge of Domain Similarity(which may not hold in real-world applications).
- No explicit negative transfer experiments.

**Questions For Authors:**

1. How does the MLMC-based pessimistic aggregation scale to large state-action spaces?
2. Can the pessimism level \( \zeta \) be adaptively tuned instead of being fixed?

**Relation To Broader Scientific Literature:**

This work connects to several research directions:

1. The framework extends **robust RL** (Iyengar, 2005; Nilim & El Ghaoui, 2004) and **transfer RL** (Chen et al., 2021) by incorporating pessimism-based transfer guarantees.
2. The **federated learning** approach aligns with distributed RL (Jin et al., 2022; Wang et al., 2024).

**Theoretical Claims:**

I verified the key theoretical claims, particularly:

**Lemma 4.1** (Pessimism Gap & Sub-Optimality Bound)
   - Correctly characterizes the effect of pessimism on target policy performance.
   - Uses standard value function contraction properties to establish robustness guarantees.

**Theorem 5.2** (Averaged Proxy is Conservative)
   - Proves that the AO proxy remains a valid lower bound to the target domain’s value function.

**Theorem 6.1** (Minimal Pessimism Proxy is More Effective)
   - Assumes prior knowledge of domain similarity, which is non-trivial in real-world applications.

---

> ### Author Rebuttal · Authors · 2025-04-01
>
> We sincerely thank you for your time and feedback, and appreciate the reviewer identifying our contribution and novelty. Please refer to the link https://anonymous.4open.science/r/ICML-2663/README.md for our newly conducted experiments.
>
> **Other relevant robust RL approaches**
> We thank the reviewer for pointing these methods out. We first want to highlight that our method is indeed based on distributionally robust RL, where our uncertainty set is constructed to account for the potential distributional shift. The adversarial robust RL, although commonly studied in experiments, generally lacks theoretical guarantees on, e.g., convergence, and tends to be overly conservative. Our methods, however, enjoy both convergence and performance guarantees. We will include more discussion in our final version.
>
> We then numerically verify that our method is effective, and present the results for the recycling robot problem in Table 2 in the above link. We use adversarial action-robust RL [1] and distributionally robust RL [2] as new baselines. As the results show, although adversarially robust and distributionally robust RL outperform
> non-robust methods, their performance remain significantly inferior to ours.
>
> [1] Tessler et al., 2019. "Action Robust Reinforcement Learning and Applications in Continuous
> Control."
>  [2] Panaganti \& Kalathil, 2022. "Sample Complexity of Robust RL with a Generative Model."
>
> **Cost of MLMC**
> MLMC is primarily designed for proof under the tabular settings, and is essential for obtaining an unbiased update. As we discussed, the computational cost of MLMC can be viewed as a trade-off of transfer performance, thus such a high cost may be acceptable in safety-critical applications such as robotics and autonomous driving.
>
> Its cost can be reduced along several potential directions: (1) The level number $N$ of MLMC can be controlled through techniques like threshold-MLMC [1], which applies an upper bound on $N$. Although it results in a biased estimation, the bias can be controlled and hence still implies convergence (numerically verified in Fig 3 in the link). (2) Techniques to reduce the computational cost of the robust Bellman operator can also be applied. One potential approach is to relax the uncertainty set constraint, which would result in a conservative and efficient solution [2]. (3) Reducing the aggregation frequency by increasing $E$ is also another way to reduce the computational cost, but introduces a trade-off in the convergence rate, as we have shown in our theoretical results (further numerically shown in Fig 2 of the link).
>
> [1] Wang, Y. et al. Model-Free Robust Reinforcement Learning with Sample Complexity Analysis, 2024
> [2] Kumar, N.et al. Policy gradient for rectangular robust Markov decision processes, 2023
>
> **No robustness tests** We developed some robustness testing experiments in Appendix, where our methods are shown to be more robust to target domain uncertainty. We also conduct additional experiments to verify robustness. Firstly, as the results in Table 2 in the link show, our methods are more robust to uncertainty in the target domain. Under different levels of uncertainty, our methods outperform baselines and is hence robust. Secondly, in Fig 3, we showed that even with noise or bias in the update, our algorithms still outperform the baseline and are also robust.
>
> We will include more robustness tests in the final version.
>
> **Assuming prior knowledge of domain similarity** Generally, such knowledge can be obtained through domain experts, or estimated from a small amount of target data.
>
> Moreover, our method remains applicable even without prior knowledge, by setting $\Gamma$ to a known upper bound on distributional distance (e.g., total variation between any two distributions is at most 1). While this yields an over-conservative proxy, it still helps prevent significant drops in transfer performance due to our built-in pessimism principle. We believe this is the most reliable approach in settings where no similarity information is available.
>
> We also note that this limitation applies to other methods as well. For instance, domain randomization (DR) assumes prior access to a set of environments that includes the target domain (Chen et al., 2021). Without such prior knowledge, our pessimism principle helps prevent undesirable outcomes, whereas DR provides no such guarantees.
>
> **Can ($\zeta$) be adaptively tuned?**
> We first note that $\zeta$ depends on $\Gamma$ in our algorithm. Since we consider a zero-shot setting, $\Gamma$ is pre-set and fixed, and it cannot be tuned without any additional information. However, if any additional knowledge is available, e.g., a small amount of target domain data, or we are allowed to fine-tune via exploration, we can use it to shrink the uncertainty radius and enhance the effectiveness.
>
> **Negative transfer** We sincerely refer the reviewer to our response to Reviewer QZdX. The results are in Fig 1.

---

### Decision · Program_Chairs · 2025-05-01

**Decision:**

Accept (poster)

**Comment:**

All reviewers appreciate the contribution of their paper. I also think it has nice theoretical contribution.